# Spatial transcriptomics maps host–gut microbiome biogeography at high resolution

Ioannis Ntekas [1,2], Lena Takayasu[1,2], David W. McKellar[1,2], Benjamin Grodner [1], Chase Holdener [1], Peter Schweitzer[1], Young Seo Park [1], Maya Sauthoff[1], Qiaojuan Shi [1], Ilana L. Brito [1] & Iwijn De Vlaminck [1]✉

Intermicrobial and host–microbial interactions are critical for the functioning of the gut microbiome, but few tools are available to measure these interactions in situ. Here we report a method for broad spatial sampling of microbiome–host interactions in the gut at high resolution (1 µm). This method combines enzymatic in situ polyadenylation of both bacterial and host RNA with spatial RNA sequencing to increase bacterial RNA recovery and enable transcriptomic analysis of low-abundance and spatially restricted microbial taxa. We benchmark the method against existing spatial transcriptomic workflows, demonstrating improved sensitivity and resolution. Application of this method in a mouse model of intestinal neoplasia revealed the biogeography of the mouse gut microbiome as function of location in the intestine, frequent strong intermicrobial interactions at short length scales and tumour-associated changes in the architecture of the host–microbiome interface. This method is compatible with widely available commercial platforms for spatial RNA sequencing and can therefore be readily adopted to study the role of short-range, bidirectional host–microbe interactions in microbiome health and disease.

It has long been speculated that the gut microbiome functions as an organ system with tissue-like properties defined by interactions between microbial and host cells[1,2]. Yet, investigating host–microbiome interactions has been difficult due to a lack of adequate measurement tools[3]. Advances in imaging have enabled the study of the localization of specific microorganisms in the gut and oral microbiomes[4–7], but these approaches are limited in multiplexity or fail to provide detailed information about host function or transcriptional responses[3–10].

Spatially resolved RNA sequencing (spatial RNA-seq) has recently emerged as a powerful approach to profile gene expression in intact tissues while preserving spatial context[11]. Commercial implementations are predominantly optimized for polyadenylated host transcripts and have been applied to examine the cellular architecture of intestinal tissues in health and disease[12–16]. More recently, adaptations of spatial RNA-seq have enabled profiling of the host–microbiome interface, for

example, by capturing A-rich microbial RNAs with poly(dT) primers or via custom capture arrays[17–20]. Although these studies have advanced our understanding of host–microbe organization, they remain limited in spatial resolution, sensitivity for microbial RNA and accessibility due to custom reagents.

Here, we address these hurdles by exploring the use of enzymatic polyadenylation of microbial RNA and host RNA in situ to map the microbiome and host via spatial RNA-seq (Fig. 1), building on our previous work on the use of this chemistry for spatial sequencing of total host RNA[21]. We demonstrate that in situ polyadenylation improves bacterial RNA recovery by oligo(dT) spatial transcriptomics arrays, by up to 100-fold, and we show that this chemistry is compatible with multiple commercial platforms. The enhanced recovery of bacterial RNAs enables spatial sampling of the microbiome at high resolution. In addition to bacterial RNAs, in situ polyadenylation enables capture

[1]Nancy E. and Peter C. Meinig School of Biomedical Engineering, Cornell University, Ithaca, NY, USA. [2]These authors contributed equally: Ioannis Ntekas, Lena Takayasu, David W. McKellar. ✉e-mail: vlaminck@cornell.edu

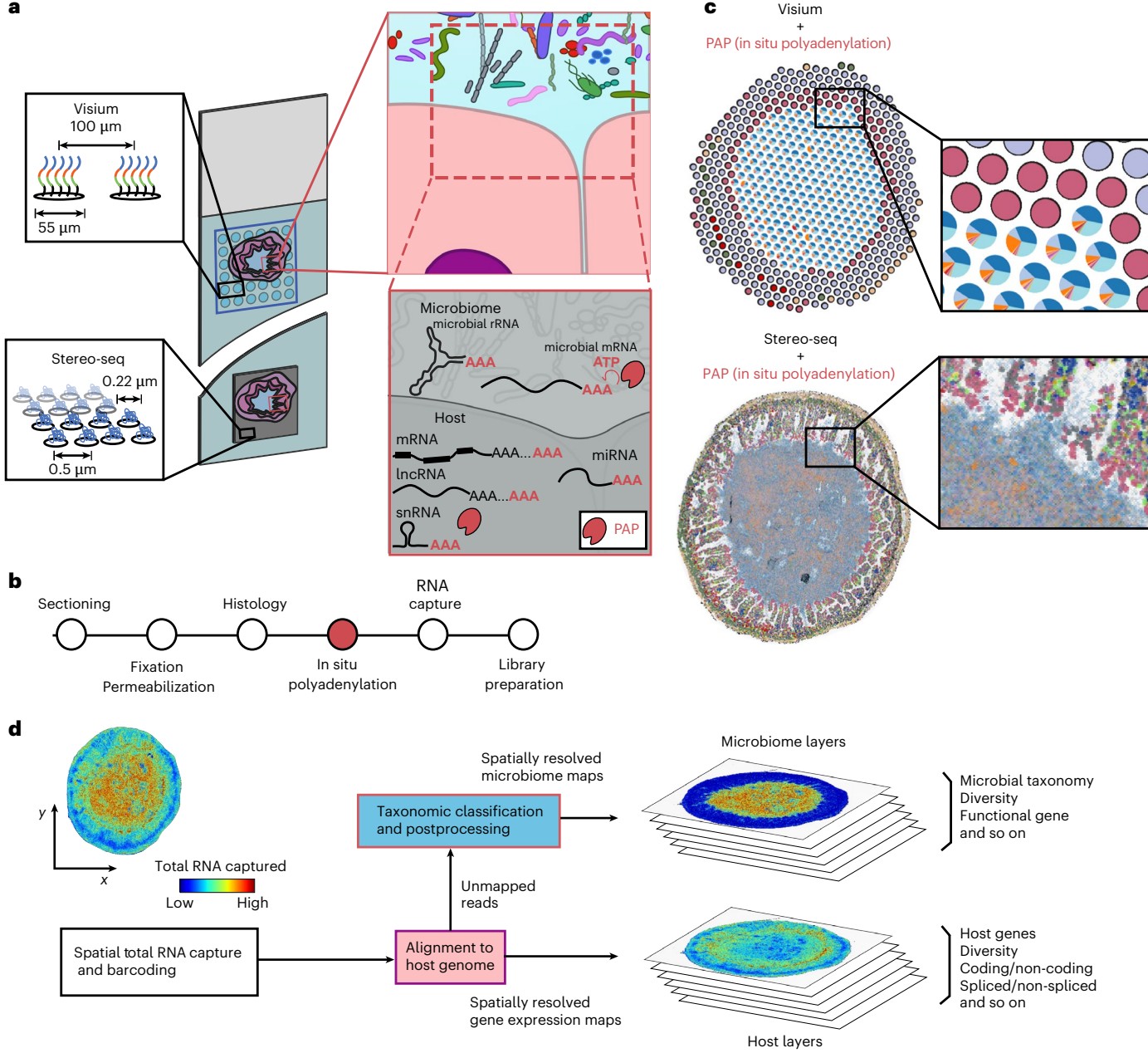

**Fig. 1 | In situ polyadenylation enables the capture of microbiome signals with sequencing-based spatial transcriptomics platforms. a**, Overview of the experimental design. Array-based spatial RNA sequencing (at low or high spatial resolution) is combined with in situ polyadenylation via PAP. **b**, A schematic of the protocol for microbiome and host spatial total RNA sequencing. The standard steps of cryosectioning, fixation and histology are followed by enzymatic in situ polyadenylation, total RNA capture and sequencing library preparation. **c**, Representative data for the low (top) and high (bottom) resolution platforms. **d**, A schematic of bioinformatics workflow.

and characterization of both A-tailed and non-A-tailed host transcripts in the intestine. By integrating these layers of information, the resulting method provides a view of microbiome–host interactions in the gut (Fig. 1). Application of this method revealed the location-dependence of the mouse gut microbiome organization, interactions within and between microbial taxa at short length scales and changes in microbiome and host cell architectures at microbiome–tumour interfaces.

## Results

### Spatial mapping of microbiome–host interaction via in situ polyadenylation

We tested whether in situ polyadenylation can enhance the recovery of microbial RNA, initially at low spatial resolution. We collected fresh-frozen tissue from a mouse model of colorectal cancer (APC deficient) at four locations—proximal small intestine, ileum, caecum and colon—and performed spatial transcriptomics on the Visium platform (Fig. 2a). Following fixation, sectioning, haematoxylin and eosin(H&E) staining and imaging, we implemented enzymatic in situ polyadenylation (Methods). To quantify the effects of in situ polyadenylation, we also performed conventional spatial RNA-seq, without in situ polyadenylation, on proximal tissue sections (Methods). After cDNA synthesis and sequencing, we obtained an average of 156 million reads per sample (156 million ± 43 million) corresponding to an average of 31,250 reads per spot (Supplementary Table 1).

To quantify microbial and host-specific sequences, we mapped the reads to the host reference genome and then performed

taxonomic classification on the unmapped reads[22] (Methods). To assess potential contamination and sequence misclassification, we assayed non-intestinal tissue (murine heart). We found very low levels of microbial signal in these tissues (0.002–0.04% of total reads, Extended Data Fig. 1). We next quantified the enrichment in microbial RNA enabled by in situ polyadenylation in intestinal tissue. In situ polyadenylation resulted in up to a 99-fold enrichment of bacterial RNA (Fig. 2b and Supplementary Fig. 1), with improved capture for most taxa, while maintaining high capture efficiency for host genes (Fig. 2c and Extended Data Fig. 2a). The enrichment of RNA from viruses and archaea was greatest in the proximal small intestine (10-fold and 6-fold increase, respectively; Extended Data Fig. 2b,c). Notably, in situ polyadenylation enhanced detection of both lowly abundant (for example, *Tannerellaceae* and *Eggerthellaceae* families) and highly abundant taxa (for example, *Lactobacillaceae* and *Lachnospiraceae*; Extended Data Fig. 3). To validate microbial composition profiles measured via spatial RNA-seq, we compared with bulk metatranscriptome sequencing from proximal tissue sections across all regions and observed strong agreement, in caecum and colon where the microbial biomass was high (Extended Data Fig. 4). In the small intestine, agreement was more modest, which may reflect greater susceptibility to false positives in low-biomass regions and substantial spatial variability at submillimetre scales (Extended Data Fig. 4).

In addition to microbial RNA, we found that in situ polyadenylation also improved the capture of several types of host-derived RNA (Fig. 2d, Extended Data Fig. 5 and Supplementary Fig. 3). For example, unspliced mRNAs were enriched after polyadenylation (15.6% of unique molecules versus 2.1%; Fig. 2d). These unspliced molecules may represent nascent transcripts, which could provide insight into dynamic cellular responses. Other biotypes were enriched, including ribosomal RNAs (rRNAs, 2.6% versus 0.16%), microRNAs (miRNAs, 0.62% versus 0.021%), small nucleolar RNAs (snoRNAs, 0.068% versus 0.009%), long non-coding RNAs (lncRNAs, 2.80% versus 1.42%), small nuclear RNAs (snRNAs, 0.043% versus 0.0012%) and miscellaneous RNAs (miscRNAs, 0.26% versus 0.001%). In situ polyadenylation enabled the identification of RNAs that are common to all assayed regions, including miscellaneous RNAs such as *Rny1* and *Rny3*, the vault RNA *Vaultrc5* and the small nuclear RNA *Rn7sk* (Extended Data Fig. 5b). In addition, we observed molecules with spatially patterned expression, including the lncRNA *Gm16759*, enriched specifically in the ileum. *Gm16759* has been shown to regulate *Smad3* expression, inhibiting the induction of intestinal regulatory T cells via the TGF-β pathway[23]. In the proximal intestine, we observed expression of the lncRNA *Gm31992*, while in the distal sections, including the caecum and large intestine, we detected expression of other non-coding RNAs including the lncRNAs *Gm56583* and *miR9-3hg*, which is implicated in human cancer[24,25].

We next examined microbiome composition as a function of gut region. Moving down the gastrointestinal tract from the proximal small intestine and ileum to the caecum and colon, we observed an increase in taxonomic richness per spot (average of 14.1 genera in the small intestine to 114.4 in the large intestine and lumen; Fig. 2e). *Lactobacillaceae* and *Muribaculaceae* were abundant in the proximal small intestine and ileum but not in caecum and colon. *Lachnospiraceae* and *Clostridiaceae* had the greatest abundance in the caecum while *Oscillospiraceae* had the highest abundance in the colon. *Flavobacteriaceae*, *Eggerthellaceae*, *Barnesiellaceae*, *Prevotellaceae* and *Tannerellaceae* had higher relative abundance in the small intestine (Fig. 2f), in line with the previous findings[26].

We measured changes in microbiome composition along the transverse axis, from the tissue to the lumen, where variations in pH, oxygen levels, nutrient accessibility and contact with the host's defence mechanisms, are expected to influence microbial composition[26]. In the small intestine, we observed greatest microbial diversity in the lumen, whereas, in the caecum and large intestine, we observed increased diversity near the mucosa. The limited resolution of the Visium platform did not allow us to resolve the mucosal layer or the interface between the lumen and mucosa (Fig. 2e). We therefore divided each map into five bins on the basis of distance to the lumen and measured the relative abundance in each bin for four representative phyla: Actinomycetota, Pseudomonadota, Bacteroidota and Bacillota (Fig. 2g). Doing so, we found that Actinomycetota and Pseudomonadota were generally more abundant near the mucosa and tissue layer, particularly in the small intestine. Bacteroidota, the most abundant phyla in the small intestine, were preferentially present away from the tissue and mucosa. Conversely, Bacillota were the dominant phyla in the caecum and large intestine across all bins, with higher levels observed away from the tissue, whereas Bacteroidota were enriched in the tissue layer (Fig. 2g and Extended Data Fig. 5c). These results demonstrate the effectiveness of in situ polyadenylation for mapping microbial and host RNA (Fig. 2h,i) and inspire experimentation at higher spatial resolution.

## Mapping microbiome–host interaction at higher spatial resolution

We next implemented in situ polyadenylation on a high-resolution spatial RNA-seq platform (Stereo-seq, STOmics), which yielded maps of host and microbiome at ~0.5 µm pixel resolution (Fig. 1a). This method was performed on tissue sections adjacent to those profiled by Visium. We found that in situ polyadenylation again improved the capture of non-coding and microbial RNA (Extended Data Fig. 6). We confirmed that the measurements performed at low resolution (Visium) and high resolution (Stereo-seq) for both host and microbial RNA were in agreement at the bulk level (Supplementary Fig. 4).

**Fig. 2 | Spatial total RNA sequencing of the murine GI tract with the Visium platform. a**, Sampling locations across the murine gastrointestinal (GI) tract measured with and without in situ polyadenylation (proximal small intestine (PS), ileum (IL), caecum (CE) and colon (CO)). **b**, Bar plots showing the percent of unique molecules classified as bacterial in the paired experiments with and without in situ polyadenylation. Murine heart tissue is included as a negative control (CTL). **c**, A scatter plot showing the genera total counts per million (CPM) UMI + 1(left) and the host genes total CPM UMI + 1(right) for the paired Visium experiment on CO with and without in situ polyadenylation. **d**, Box-and-whisker plots showing the RNA molecules percentage distribution per spatial spot for the paired experiments collected from the four different parts of the GI. Each data point corresponds to one spatial spot of the Visium array covered by host tissue. Numbers of spots were: PS (Visium, *n* = 1,405; Visium + PAP, *n* = 2,133), IL (Visium, *n* = 1,209; Visium + PAP, *n* = 1,204), CE (Visium, *n* = 273; Visium + PAP, *n* = 374) and CO (Visium, *n* = 581; Visium + PAP, *n* = 902). The boxes indicate the median and interquartile range (25th to 75th percentiles); whiskers extend to 1.5× the interquartile range; outliers are not shown. **e**, Spatial maps for the four profiled GI locations with Visium + in situ polyadenylation. The tissue portion is coloured on the basis of deconvolution results, where each spot is assigned to the most abundant cell type (see the legend). The lumen portion of the plot shows ln(Microbial Counts +1) (top) and the richness at the genus level (bottom). **f**, Box-and-whisker plots showing genus-level richness per spatial spot for four GI regions profiled with Visium + in situ polyadenylation (left) and stacked bar plots showing relative microbial family abundance for the same samples (right). Each data point in the box plots corresponds to one spatial spot of the Visium array covered by host tissue. The numbers of spots were: PS (*n* = 2,718), IL (*n* = 1,744), CE (*n* = 2,657) and CO (*n* = 3,042). The boxes indicate the median and interquartile range (25th to 75th percentiles); whiskers extend to 1.5× the interquartile range; outliers are not shown. **g**, Relative abundance of microbial taxa along the transverse axis (from tissue to lumen) at the phylum level for four GI locations. The dot plots for each location display the relative abundance of four major phyla, divided into five spot-distance bins from the tissue. The dot size represents the relative abundance percentage, and the colour indicates the relative abundance *z*-score across bins. **h**, Spatial maps of bacterial family transcript counts for *Eggerthellaceae* and *Tannerellaceae* profiled by standard Visium (left) and Visium with in situ polyadenylation (right). **i**, Spatial maps showing the log-normalized expression of select non-coding RNAs profiled by standard Visium (left) and Visium with in situ polyadenylation (right).

In a section of mouse ileum, we recovered a total of $6.1 × 10^6$ host RNAs (4.35 host unique molecular identifiers (UMIs) per square micrometre in the tissue) representing 29,386 genes and $13.1 × 10^6$ million microbial RNAs (11.38 molecules square micrometre in the lumen) representing 91 species with >0.01% abundance (Supplementary Fig. 5). We used paired imaging data to assign host RNAs to individual cells (Supplementary Fig. 6) and then predicted cell types via computational

deconvolution using single-cell RNA-seq data from the same mouse model as a refs. 27,28 (Supplementary Figs. 7–9). The relative abundance of host cell types measured with and without polyadenylation was in agreement (Extended Data Fig. 7).

We combined the host map with the microbial signal at 0.5 μm resolution to generate a detailed view of the host–microbiome interface (Fig. 3a). We observed distinct zonation patterns of coding and

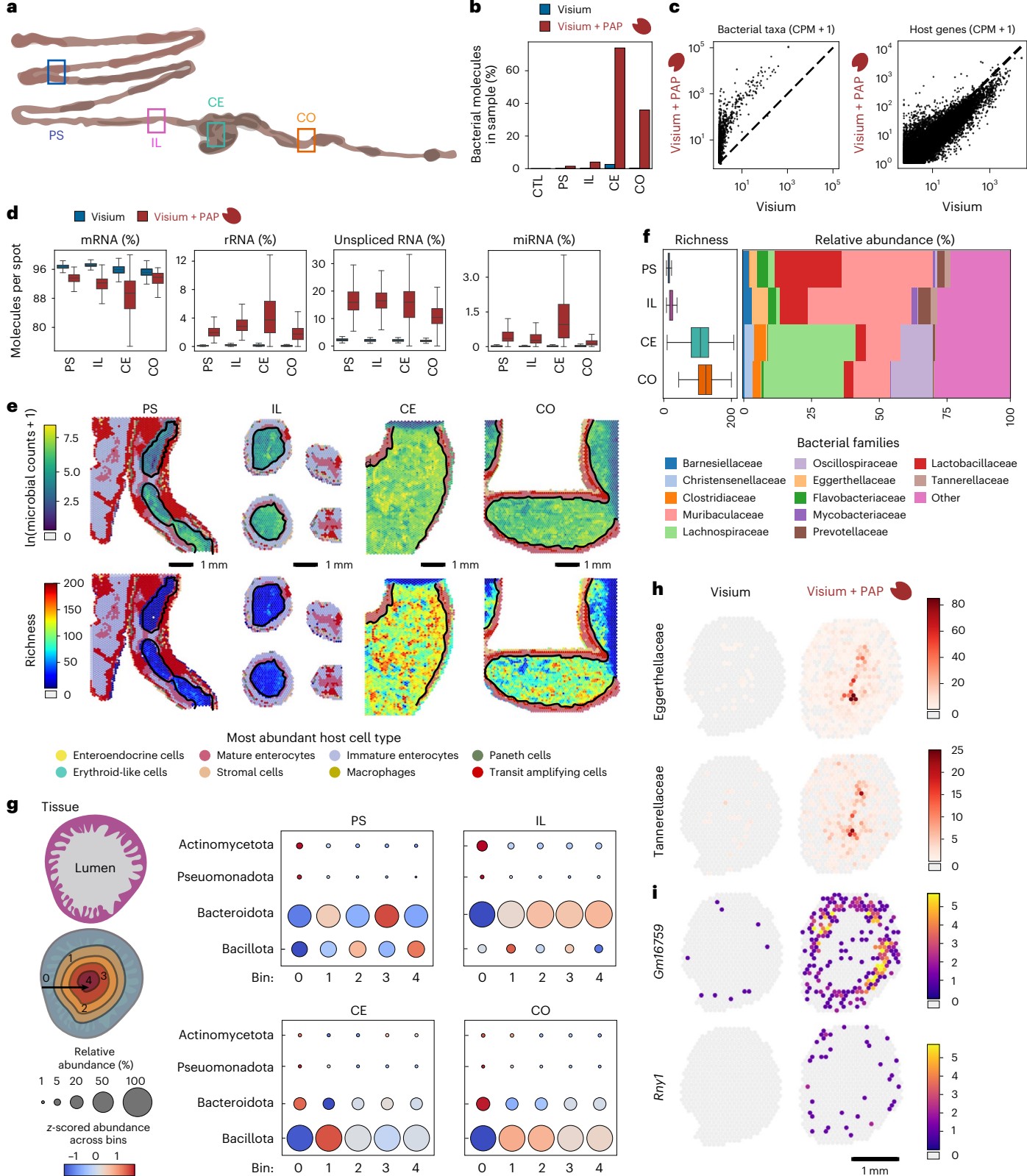

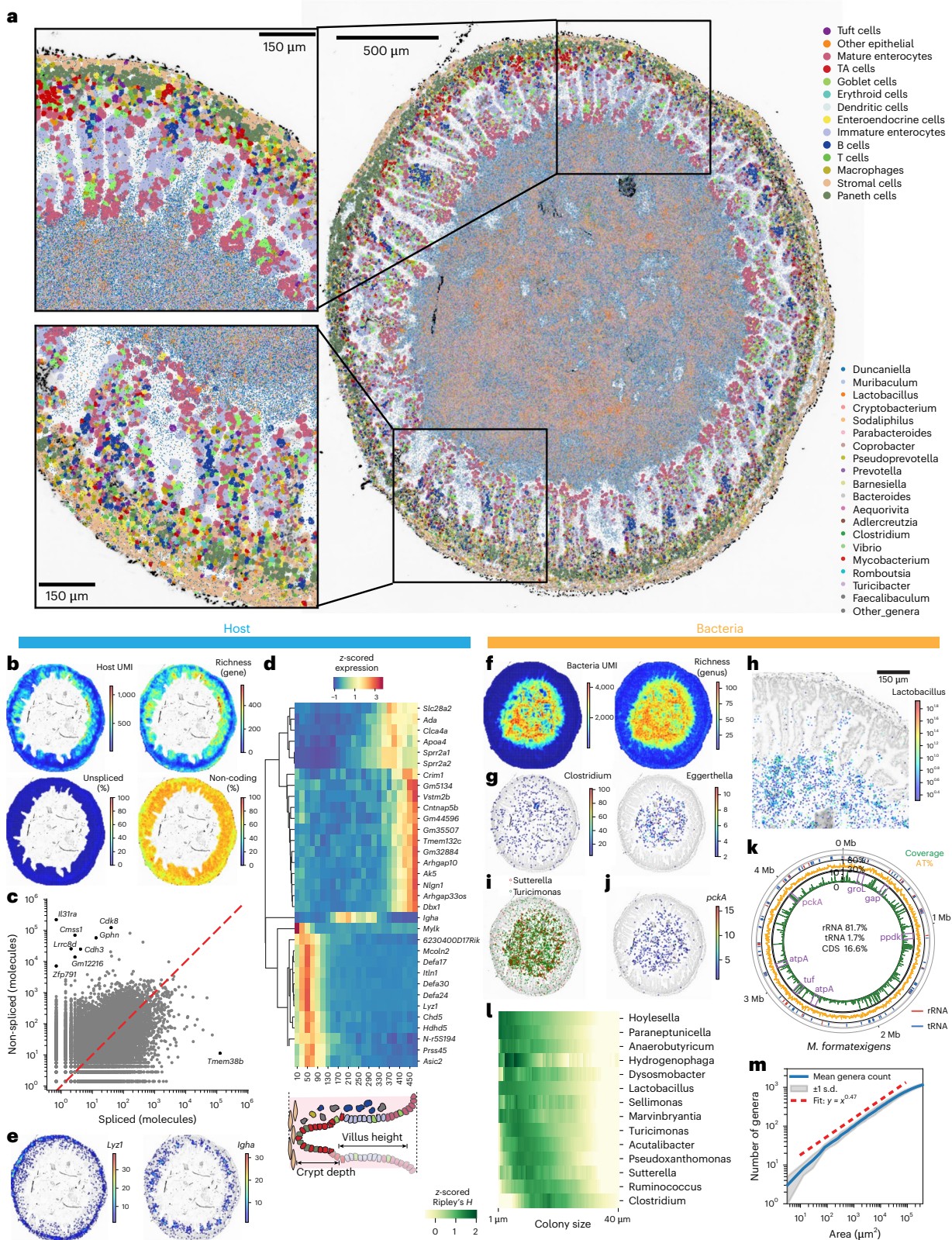

**Fig. 3 | High-resolution spatial mapping of host total gene expression and the microbiome. a**, Spatial maps of host gene expression and microbiome composition (spots with more than one microbial RNA detected are shown). **b**, Spatial maps of host UMIs, gene richness, unspliced RNA and non-coding gene ratio (20-μm square bins). **c**, Plots of spliced and unspliced molecules for each coding gene (outliers in black, points with distance from $y = x$ greater than five standard deviations). **d**, A heat map of the expression of select genes along the distance from the outer tissue edge. **e**, Spatial gene expression of select genes (20 μm² bins). **f**, Maps of measured unique bacterial molecules (UMI) and genus

richness (20 μm² bins). **g**, Spatial maps of abundance of the genera *Clostridium* and *Eggerthella*. Spots with more than one microbial count are shown. **h**, A zoom-in of abundance of *Lactobacillus*. **i**, Example of spatially correlated genera. **j**, Spatial maps of bacterial functional genes. **k**, Circular plot showing sequencing depth across the *M. formatexigens* genome. Genome coordinates are shown in the outer ring with select protein-coding genes denoted in purple, rRNA loci in red and tRNA in blue. The orange track indicated AT% and the green depicts sequencing coverage. **l**, *z*-scored Ripley's *H* score. **m**, Genera accumulation curve in the ecosystem of the gut. TA cell, transit amplifying cell.

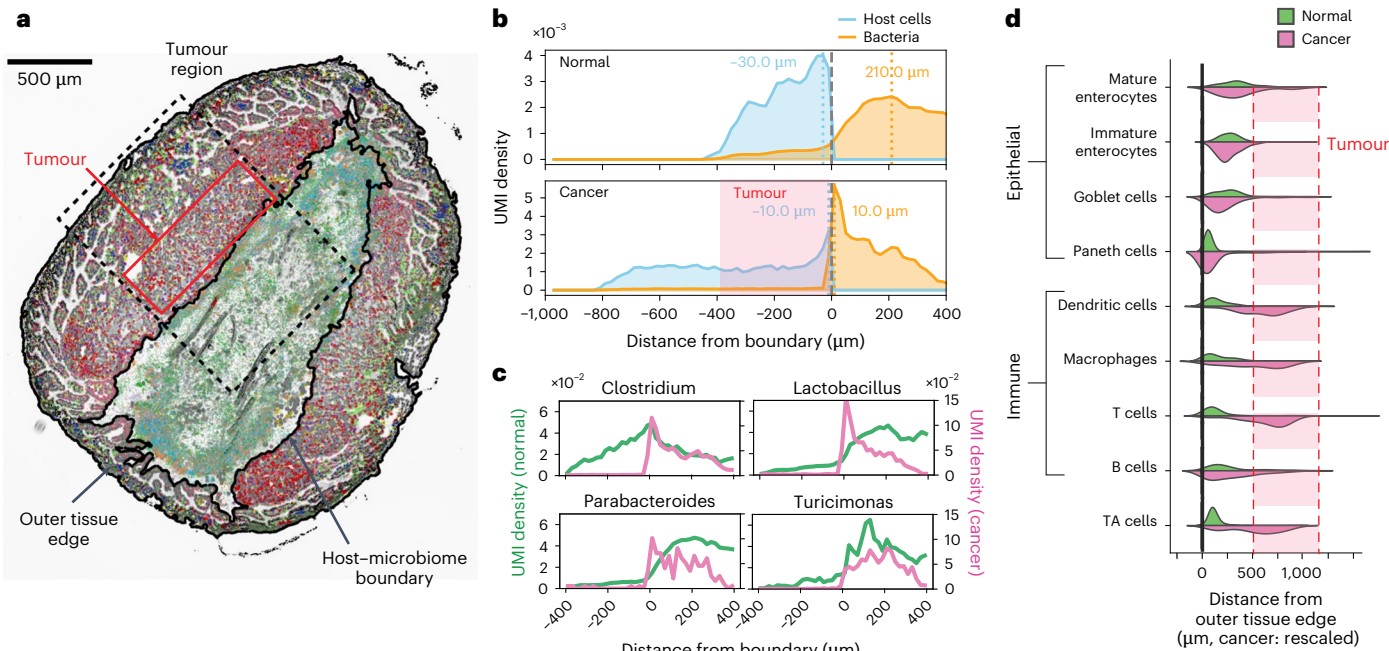

**Fig. 4 | High-resolution spatial mapping of a tumour–microbiome interface.**
**a**, Spatial map of host gene expression and microbiome in an ileum section with tumour (ApcMin/+ mouse, spots with more than one microbial count shown). The colour legend is the same as in Fig. 3a. **b**, Probability density plot of host and microbial cells as function of distance to the host–microbiome boundary, for normal (top) and tumour-laden (bottom) tissue. **c**, Genera abundance as a function of distance to the host–microbiome boundary for normal (green) and tumour-laden (pink) tissue. **d**, Violin plot of cell density as function of distance from the outer tissue edge. The distances of each cell type in the tumour were linearly rescaled to match the location of mature enterocytes in a section without tumour. TA cell, transit amplifying cell.

non-coding host gene expression. Host gene expression was spatially non-uniform, with higher levels and diversity of gene expression observed at the tips of the villi, probably due to the increased transcriptional activity of mature enterocytes (Fig. 3b, Extended Data Fig. 6 and Supplementary Fig. 8). Unspliced mRNAs accounted for 22.5% of the total host RNA in the tissue, with a higher proportion observed at the base of the crypts, possibly associated with the turnover of transit amplifying cells (Fig. 3b). Genes with a high proportion of unspliced molecules included *Cdk8* (cyclin-dependent kinase 8), a transcription regulatory protein and oncogene associated with human colorectal cancer (Fig. 3c). These observations are in line with previous studies which have shown that Apc-deficient CRC cells dysregulate RNA splicing machinery[29]. As expected, in situ polyadenylation improved the capture of non-coding gene transcripts (Extended Data Fig. 6b). Non-coding RNA expression was elevated in the zone closer to the gut wall (Fig. 3b). Some non-coding genes showed cell-type specific expression, such as *Hnf1aos1* (Supplementary Fig. 10). We detected several landmark genes, including the non-coding gene *6230400D17Rik*, which was enriched closer to the gut wall, and *Ada*, which was found at the tips of the villi (Fig. 3d). Finally, we identified transcripts of genes that are known to be involved in microbiome response, such as the lysozyme encoding gene *Lyz1* and other antimicrobial peptides including defensins expressed by Paneth cells at the base of crypts, as well as *Igha*, which encodes a segment of the IgA heavy chain, expressed by plasmablasts in the lamina propria (Fig. 3e).

Analysis of the local density and diversity of bacteria revealed micro-scale features of the spatial organization of the microbiome in the ileum (Fig. 3f). We observed lower bacterial burden and diversity close to the host boundary. *Clostridium* was evenly distributed, with the exception of one large cluster in the lumen. *Klebsiella* was abundant near the tip of the villi, and *Eggerthella* was abundant away from the tissue boundary (Fig. 3g). We observed clusters for several genera: 54 genera showed significant autocorrelation (Moran's I P values <0.05,

major genera with >0.01% total bacterial counts) in line with colony formation (Supplementary Fig. 11). For these genera, we calculated Ripley's H to infer cluster size (Fig. 3l). Some genera including *Lactobacillus* showed small colony size (radius <10 μm, Fig. 3h), whereas others including *Turicimonas* had medium-sized colonies (~10 μm) and taxa including *Clostridium* formed bigger colonies (>30 μm). Analysis of spatial correlation[30] between colony-forming genera revealed strong correlations between genera, including between *Turicimonas* and *Sutterella* (Fig. 3i).

To further explore the biogeography of the gut microbiome, we analysed additional sections from different locations in the gastrointestinal tract (proximal small intestine, ileum, caecum, proximal and distal colon). In the ileum, the distribution of phyla along the transverse axis closely matched the measurements on the low-resolution platform, with Bacillota and Pseudomonadota enriched near the tissue and Bacteroidota in the lumen (Supplementary Fig. 12). In the large intestine, Bacillota remained the most abundant phylum with greater representation in the lumen, whereas Bacteroidota were enriched closer to the mucosa. At the family level, in the proximal colon, *Lachnospiraceae* and *Oscillospiraceae* were enriched near the tissue boundary, whereas *Lactobacillaceae* and *Bacteroidaceae* were more prominent in the lumen, consistent with the known biogeography of this region[31] (Supplementary Fig. 13). We again examined for the presence of colonies (Methods) and observed clear evidence of colony formation for *Lactobacillus*, *Turicibacter* and *Clostridium* in the ileum and more distal regions and smaller colonies in the proximal small intestine (Supplementary Fig. 14), with smaller colonies for *Lactobacillus* and larger colonies for *Clostridium* in ileum, consistent with our earlier observations. The size of bacterial colonies may be influenced by factors such as bacterial reproductive capacity, the abundance of available resources and the level of inter-taxa competition. Further investigation is needed to elucidate how these colonies form and how they contribute to microbial community structure.

We aligned microbial reads to the genomes of the six most abundant species (Methods), which enabled the identification of bacterial gene expression. Transcripts of *groL*, a marker gene of rapid growth, *ppdk*, *pckA* and *gap*, which are related to sugar metabolism and *atpA*, a housekeeping gene, were detected, with signal predominantly originating from the lumen (Fig. 3j,k and Extended Data Fig. 8). The relative contribution of rRNA versus bacterial mRNA and tRNA was dependent on location in the gut. rRNA abundance was highest in the caecum, while in the colon we observed an increase in tRNA reads (>5% of aligned reads). In the small intestine, rRNA was lowest, and coding sequences (CDS) were comparatively more abundant (for *Marvinbryantia formatexigens* 16.6%). These spatial trends, together with colony information and bacterial gene expression, can provide insight into bacterial growth and function within the mouse gut.

Last, we measured the relationship between habitat area size and the number of unique species identified in the ecosystem (Fig. 3m). The relationship between the number of unique genera observed and the area sampled followed a power law over three orders of magnitude ($16 \ \mu m^2$ to $0.16 \ mm^2$), in line with observations of species–area relationships in a wide range of systems, including plant and animal ecosystems. The observed power exponent of 0.47 (genus level) indicated relatively high spatial dispersion for the microbiome in the ileum of mice relative to exponents reported for plant, animal and environmental ecosystems[32].

Microorganisms are an inherent part of the microenvironment of cancers that develop at epithelial barrier surfaces[33]. To study the spatial organization of host cells and gut microorganisms associated with tumours, we assayed a section of ileum tissue with notable tumours. We compared the microbiome at the edges of tumour and normal tissue (Fig. 4a and Extended Data Fig. 9). To this end, we first defined the boundary between host and microbiome based on microscopy images (Supplementary Fig. 15) and then measured the spatial organization of host cell types and key taxa as a function of distance to this boundary. This analysis showed that in normal tissue, microorganisms are most dense 100–200 μm from the host villi (Fig. 4b, top), whereas in the tumour tissue, microorganisms are most dense directly at the boundary with the tumour (Fig. 4b, bottom). *Clostridium*, the most abundant genus, and *Lactobacillus* and *Parabacteroides* were closely associated with the tumour edge (Fig. 4c), whereas for normal tissue, these taxa were found away from the tissue boundary. Both *Lactobacillus* and *Turicimonas* again showed evidence of colony formation (radius 10–20 μm; Supplementary Fig. 16). In normal tissue, mature enterocytes were located closest to the host–microbe boundary, followed by immature enterocytes and other intestinal epithelial cells. Paneth cells were located at the basal region, in line with the known architecture of ileum tissue. By contrast, tumour-associated transit amplifying cells, and immune cells including dendritic cells, macrophages and CD8+ T cells were enriched in the tumour (Fig. 4d and Supplementary Fig. 17). At the gene level, expression of some genes moved more towards the lumen in the tumour subregion, including cancer-related genes such as *Pkm* (Supplementary Fig. 18). Mucin-producing goblet cells were located away from the tissue-microbiome boundary due to the presence of the tumour mass. Additional data from an adjacent ileum section and data from an independent mouse validated these observations and indicate a relation between tumour–microbiome proximity and tumour size (Supplementary Figs. 17 and 19).

## Discussion

In this study, we show that combining in situ polyadenylation with spatial RNA-seq enables spatial mapping of the gut microbiome and the host A-tailed and non-A-tailed transcriptomes. By adding a simple enzymatic step in the workflow of commercially available platforms, this method provides an accessible and scalable way to measure the host–microbe interactome across spatial scales. We applied this methodology to profile intestinal tissue in wild-type mice and a mouse model of intestinal neoplasia. We characterized changes in microbiome composition as a function of location in the mouse intestine, which corroborated many known features of the organization of the gut microbiome in mice. Analyses at high spatial resolution revealed interactions within and between microbial taxa, including evidence of colony formation for certain taxa, with colony sizes ranging from a few micrometres to >30 μm in radius. Colony formation may indicate active growth, and if so, colony size may be a proxy for growth rate. At the boundary between the microbiome and tumour-laden tissue, we observed a shift of key microbial taxa towards the boundary with the host, suggesting the potential for increased interactions in this region. These localized changes in microbiome structure are probably in part explained by the altered local host tissue architecture, including the displacement of mucin-producing cells away from the boundary. As we have shown previously, in situ polyadenylation also enabled mapping of both the A-tailed and non-A-tailed host transcriptomes[21]. Analysis of the 'total' transcriptome revealed spatially restricted expression of several classes of non-coding RNAs. We identified landmark coding and non-coding molecules along the crypt–villus axis, with increased expression in mature enterocytes at the villus lining and a higher fraction of unspliced, newly transcribed RNA near the crypt.

Although this study lays the groundwork for considering spatial structure in microbiome research, limitations remain. The primary barrier lies in the high cost of commercial spatial RNA-seq platforms, which can hinder broader adoption and applications such as drug screening. Notably, the added cost of in situ polyadenylation is relatively small (Supplementary Tables 2–4). Because polyadenylation leads to a redistribution of reads across coding, non-coding and microbial RNAs, host mRNA genes are recovered at lower depth; however, this can be mitigated by deeper sequencing (Supplementary Fig. 20). For studies focused on predefined gene programs, imaging-based methods that target specific host or microbial transcripts offer higher sensitivity and spatial resolution[4,34–38], whereas sequencing approaches have inherently higher sequence resolution. The lower sensitivity of sequencing platforms is important to consider given that capture efficiency may constrain identification of specialized cell types and states[34,35,37]. The lower resolution achieved with sequencing assays also limits more detailed analysis of microbiome organization, for microorganisms with low local abundance, as compared with imaging[36]. We note for example that some microbial reads appear in regions where bacteria would not be expected, which is probably explained by projection of luminal RNA into deeper regions during capture, diffusion during enzymatic reactions or small positional shifts during tissue sectioning or mounting, as well as misclassification of host or reagent-derived sequences by current database and classification approaches. Rigorous controls and orthogonal validation therefore remain essential for these methods to guard against false positives[39]. Additional technical refinements in library preparation chemistry, fixation protocols, tissue sectioning, incorporation of long-read sequencing[40,41] and improvements to the library preparation including host and microbial rRNA depletion, may further increase capture efficiency for non-coding RNAs and microbial mRNA transcripts[42–49]. Further, making the methodology compatible with formalin-fixed paraffin-embedded tissue would open application of these techniques in pathology[50]. Last, there is a pressing need for new computational tools that can integrate host coding, non-coding, and microbial signals.

Despite these limitations, this study shows that spatial transcriptomics provides a window into microbiome ecology and host–microbiome interaction. Going forward, spatial transcriptomics will be a powerful approach to explore questions in gut immunology[51,52], to explore microbial colonization of mucus and intestinal tissue, to study microbiomes in small niches such as crypts and to investigate the concept of the cancer-associated microbiome. Spatial transcriptomics can

further be applied to explore the role of specific taxa in diseases with known microbiome involvement, such as inflammatory bowel disease and other autoimmune disorders.

## Methods

### Animal models and experimental procedures
All animal protocols were approved by the Cornell University Institutional Animal Care and Use Committee (IACUC), and experiments were performed in compliance with institutional guidelines (protocol number IACUC 2016-0088). C57BL/6-ApcMin/+/J and C57BL/6 wild-type mice were used for the spatial transcriptomics experiments. All mice were maintained at the mouse facility at Weill Hall of Cornell University under standard housing conditions, including a 12 h light–12 h dark cycle, controlled ambient temperature (approximately 20–25 °C) and controlled humidity (approximately 30–70%), with ad libitum access to food (autoclavable Teklad global 14% protein rodent maintenance diet #2014-S; Envigo) and water. ApcMin/+ and wild-type mice were initially ordered from Jackson Laboratory and then bred in the barrier facility. The ApcMin/+ mice used in these experiments have a chemically induced transversion point mutation at nucleotide 2549, resulting in a stop codon at codon 850, truncating the APC protein.

Five mice were used for the spatial transcriptomics experiments: three C57BL/6-ApcMin/+/J (two female, one male) and two C57BL/6 wild-type mice (one female, one male), all approximately 13 weeks old at the time of collection. Overall health, food intake and weight of the mice were monitored to ensure tumour burden did not violate ethical standards. After approximately 100 days, mice were killed using 5 min of $CO_2$ asphyxiation followed by tissue collection. Intestines from the mice were inspected for tumour localization, excess fat was removed, and cut into individual sections. Tissues were embedded in cryomolds with OCT compound (Tissue-Tek) and frozen in an isopentane-liquid nitrogen as previously described. The small intestine was cut into four to six approximately equal-sized segments and the large intestine into two to three segments, and the caecum was processed separately.

### In situ polyadenylation reaction optimization
To determine an appropriate incubation time for in situ polyadenylation, an in vitro time-course assay was performed. A uniform 1.8 kb RNA library (Promega, L4561) was incubated with yeast poly(A) polymerase (yPAP, Thermo Scientific) under identical reaction conditions as the tissue sections. Duplicate reactions were prepared for a series of incubation timepoints (0, 1, 5, 15 and 25 min). Following polyadenylation, RNA was purified using a silica-membrane-based cleanup kit (RNA Clean and Concentrate, Zymo Research), and fragment size distributions were analysed with an Agilent Tapestation 4200. In bulk reactions, over 90% of input RNAs were polyadenylated within 1 min of incubation, and poly(A) tails exceeding 10 kb in length were detected after 15 min at 37 °C. On the basis of these results, a 25-min incubation was selected as adequate time for the enzymatic reaction in situ (Extended Data Fig. 10).

### In situ polyadenylation for the gastrointestinal tract profiling with the Visium platform
Cryosections were obtained from the proximal small intestine, ileum, caecum and colon of a single individual (male, 13 weeks). Sections were processed using a polyadenylation protocol or the standard Visium protocol. For the polyadenylation protocol, 10-µm-thick tissue sections were mounted onto Visium Spatial Gene Expression v1 slides. Sections were fixed in freshly prepared methacarn solution (60% methanol, 30% glacial acetic acid and 10% chloroform) at room temperature for 15 min. H&E staining was performed according to the Visium protocol, and tissue sections were imaged using a Zeiss Axio Observer Z1 microscope equipped with a Zeiss Axiocam 305 colour camera. The resulting H&E images were corrected for shading, stitched, rotated, thresholded

and exported as TIFF files using Zen 3.1 software (Blue edition). After imaging, slides were transferred into the Visium Slide Cassette. The methacarn fixation step serves as an initial permeabilization for the following enzymatic incubation.

In situ polyadenylation was conducted using yPAP (Thermo Scientific, catalogue number 74225Z25KU). Each capture area was equilibrated by adding 100 µl of 1× yPAP Reaction buffer (20 µl 5× yPAP reaction buffer, 2 µl 40 U µl$^{-1}$ Protector RNase Inhibitor, 78 µl nuclease-free $H_2O$), for 30 s at room temperature, after which the buffer was removed. Subsequently, 75 µl of yPAP enzyme mix (15 µl 5× yPAP reaction buffer, 3 µl 600 U µl$^{-1}$ yPAP enzyme, 1.5 µl 25 mM ATP, 5 µl murine RNase inhibitor, 50.5 µl nuclease-free $H_2O$) was added to each reaction chamber. The chambers were sealed and incubated at 37 °C for 25 min, and the enzyme mix was removed. Following polyadenylation, enzymatic permeabilization was performed for 30 min, followed by the standard Visium library preparation protocol to generate cDNA and final sequencing libraries. For the standard Visium experiment, polyadenylation was skipped and immediately proceeded to permeabilization.

### In situ polyadenylation with the Stereo-seq platform
Adjacent ileal cross-sections to those profiled with Visium were profiled by STOmics platform. The 10-µm-thick sections were placed onto STOmics mini chips (product number 211ST004). For the polyadenylation protocol, sections were fixed in methacarn for 15 min, followed by a DNA staining step according to the STOmics protocol. Imaging was performed on a Zeiss Axio Observer Z1 Microscope using a Hamamatsu ORCA Fusion Gen III Scientific CMOS camera. Images were stitched, rotated, thresholded, processed and exported as TIFF files using Zen v.3.1 software (Blue edition), then registered using the STOmics software. After imaging, in situ polyadenylation was performed followed by 12 min permeabilization and library preparation. For the standard experiment, polyadenylation was skipped and immediately proceeded to permeabilization.

Additional samples profiled included ileal sections from two mice, male and female, with a tumour adjacent to the lumen, four sections from a healthy C57BL/6 wild-type male (proximal small intestine, ileum, caecum and distal colon) and one proximal colon section from a wild-type female.

### Sequencing of the spatial transcriptomics libraries
Sequencing of the Visium libraries was performed on a NextSeq 2K (Illumina) platform using a P3 200 bp kit, with reads allocated as follows: 28 bp for read 1, 10 bp for index 1, 10 bp for index 2 and 190 bp for read 2. For the libraries prepared using the STOmics platform, sequencing was carried out on a Complete Genomics DNBSEQ-T7 Sequencer using the DNBSEQ-T7 High-throughput Sequencing Set (FCL PE100) and the associated STOmics primer set. The sequencing run consisted of a 50-bp read 1 (with dark cycles from bases 26 to 40), a 100-bp read 2 and a 10-bp index read.

### Metatranscriptome library preparation of adjacent sections and sequencing
For each intestinal region, adjacent cryosections corresponding to Visium and Visium + PAP samples were processed for bulk metatranscriptomic analysis. Total RNA was extracted using the NucleoSpin RNA Stool Kit (Takara Bio, 740130.50). To remove contaminating genomic DNA, DNase treatment was performed using a two-step protocol: (1) initial treatment with Baseline-ZERO DNase (1 µl; LGC Biosciences, DB0715K) and (2) treatment with TURBO DNase (3 µl; Thermo Fisher Scientific, AM2238) in 10 µl of the supplied buffer. A 1-µl aliquot of RNA was diluted in 99 µl of nuclease-free water, incubated at 37 °C for 30 min, and then placed on ice. RNA cleanup was performed using the RNA Clean and Concentrator kit (Zymo Research, R1013), and eluted in 10 µl of nuclease-free water.

Stranded total RNA-seq libraries were prepared using the SMARTer Stranded Total RNA-Seq Kit v3−Pico Input Mammalian (Takara Bio, 634485), following the manufacturer's protocol. The four libraries were pooled and sequenced with MiniSeq sequencer (Miniseq High Output Reagent Cartridge, 150 Cycles, Ilumina).

## Preprocessing and alignment of spatial transcriptomics data

To ensure similar alignment and quantification across platforms and methodologies we used the 'slide_snake' pipeline that utilizes Snakemake[53] (6.1.0), which can be found on GitHub[54]. For the Visium and STRS (Visium) libraries, the pipeline first trims poly(A) and poly(G) sequences, and primer sequences using cutadapt[55]. The reads were aligned using STAR v2.7.10a[56] and STARSolo[57] (specified parameters: --outFilterMultimapNmax 50, --soloMultiMappers EM, --clipAdapterType CellRanger4) to generate expression matrices for every sample. The GeneFull matrices were used for downstream analyses. Barcode whitelists and the associated spot spatial locations for Visium data were copied from the Space Ranger software ('Visium-v1_coordinates.txt'). For the Stereo-seq and STRS (Stereo-seq) libraries, barcode maps were provided by the manufacturer as .h5 files and converted to text format using ST_BarcodeMap[58]. Alignment references were generated from the GRCm39 reference sequence using GENCODE M32 annotations.

## Unmapped reads classification and construction of microbiome Anndata objects

In this study, Kraken2 (version 2.09)[22] was used to classify microbial reads. We used the standard Kraken2 database supplemented with the mouse genome. Unmapped reads flagged in the BAM file were processed to retain the correct cell barcode and UMI information as identified by STARsolo. This permitted the demultiplexing of Kraken2 output by cell barcode and UMI. For data integration, we employed Pandas, Scanpy, NumPy, Scipy and regular expressions to create an AnnData object with cell barcodes as observations and NCBI taxonomy IDs as features. Only classified reads were retained for subsequent analysis.

## Image registration and cell segmentation

For Visium and Visium + PAP samples, image registration was performed using the 10x Genomics Loupe Browser. H&E-stained tissue images were aligned to the spatial capture array, and tissue and lumen regions were manually annotated. For Stereo-seq samples, nuclear-stained fluorescence images were acquired during sample processing. Image registration and cell segmentation were carried out using the Stereo-seq Analysis Workflow (SAW) provided by STOmics. Fluorescence images were aligned to the chip layout based on the ChipID metadata. Following registration, automated cell segmentation was performed using SAW's built-in algorithms. Segmentation masks were used to define cell boundaries, and barcodes within each segmented region were aggregated to construct single-cell transcriptomes. These cell-level datasets were then used for downstream spatial analysis and deconvolution.

## Processing and alignment of metatranscriptomic libraries

Metatranscriptomic sequencing data were processed using a custom computational workflow. Adaptor trimming and quality filtering were performed using BBDuk (v38.90). Filtered reads were aligned to the mouse genome (GRCm39) with STAR (v2.7.10a) using GENCODE M32 annotations. Gene-level quantification was carried out with featureCounts (v2.0.0). Duplicate reads were identified and marked using Picard MarkDuplicates (v2.19.2). Unmapped reads after host genome alignment were extracted and taxonomically classified using Kraken2 (v2.0.9).

## Sterile control preprocessing and identification of taxa to filter

To assess the Kraken2-classified microbial counts occurring in non-intestinal tissues for the low-resolution platform we realigned previously published Visium and STRS libraries of mock-infected C57BL/6J 11-day-old mice with and without polyadenylation as described in the corresponding studies[15,21]. A total of 85 taxa occurring at 1 ppm (UMI) or greater were excluded from downstream analysis as potential misclassification. For the Stereo-seq libraries, a sterile control experiment was conducted. In brief, fresh-frozen heart from an 11-day-old mouse was sectioned on a Stereo-seq 1 cm × 1 cm tile (STOmics, BGI). The sample was fixed in methanol at −20 °C for 20 min followed by the in situ polyadenylation and the Stereo-seq library preparation protocol as described above. Taxa occurring at frequencies higher than 1 ppm UMI were excluded from downstream analyses.

## Preprocessing of the Visium and Visium + PAP data

Spatial coordinates were assigned to the Visium and Visium + PAP library spots based on the barcode map provided by the Space Ranger software ('Visium-v1_coordinates.txt'). The accompanying H&E histology images of each experiment were used to manually mark the spots that correspond to tissue and lumen. Scanpy[59], mudata[60,61] and muon[60] were used to construct multimodal objects separately for the microbial maps (in the taxonomic levels of phylum, family, genus and species). This was done for each one of the accounted microbial superkingdoms of Archaea, Bacteria and Viruses. For downstream analyses, only the spots covered by tissue or corresponding to lumen were accounted for.

## Microbial percentage and enrichment calculation for the paired Visium and Visium + PAP experiments

Microbial percentages were calculated as the fraction of Kraken2-classified unique molecules assigned to each superkingdom, normalized by the total library size (host-aligned plus Kraken2-classified unique molecules). Enrichment was defined as the ratio of those percentages between paired Visium + PAP and standard Visium experiments.

## Relative abundance and bacterial richness calculations for the low-resolution datasets

To calculate the relative abundance for each sample, at the family level, the corresponding family reads were collapsed and divided by the total molecules originating from bacteria as classified by Kraken2. The microbial richness per spot was calculated as the number of unique taxa occurring per spot after the exclusion of taxa accounting for 0.01% or less of microbial molecules in the whole sample. For the transverse axis relative abundance analysis, spots were spatially binned from the tissue to the lumen based on their minimum distance to the lumen-associated region. Phyla relative abundance data were then aggregated within each bin to quantify relative abundances across the tissue–lumen axis.

## Rarefaction and sequencing saturation analysis

Rarefaction analysis was performed on paired Visium and Visium + PAP libraries collected from the four intestinal regions. Sequencing data were subsampled at defined fractions (10% to 100% of total reads) and aligned to the mouse genome using the Snakemake-based pipeline. The unmapped reads were extracted and taxonomically classified using Kraken2 and AnnData objects were generated as described for the full dataset. For classified microbial reads in each condition, rarefaction curves were generated by plotting the number of unique bacterial molecules against sequencing depth. Michaelis–Menten models were fitted using nonlinear least squares to estimate theoretical saturation behaviour, and model fit was evaluated using the coefficient of determination ($R^2$). To calculate saturation for the total library, and for host- and microbiome-derived reads, the following general formula was used:

$$\text{Saturation} = 1 - \frac{\text{Modality unique molecules}}{\text{Modality reads with valid barcode}}.$$

Here, modality refers to the source of molecules: host-aligned reads, microbial reads classified by Kraken2 or all reads combined. This metric captures diminishing returns in the recovery of unique molecules with increasing sequencing depth. Saturation was computed at each subsampling level and modelled using Michaelis–Menten kinetics constrained to a maximum of 1. The resulting curves were used to estimate the sequencing depth required to achieve saturation (for example, saturation of -0.9).

## Gram stain comparison of Visium + PAP to the bulk RNA measurement

Adjacent samples profiled by Visium + PAP and by bulk metatranscriptomics were compared at the genus level. Reads were taxonomically classified with Kraken2 and counts were aggregated by genus. The same taxa excluded in spatial control analyses were removed before comparison. Gram-stain labels for genera were retrieved from BacDive[62] and were curated to fill missing entries for highly abundant genera, including the Gram-negative: *Pseudoprevotella, Hoylesella, Segatella, Allomuricauda, Marvinbryantia, Lachnoclostridium, Vescimonas, Mediterraneibacter, Coprococcus, Caproicibacterium, Massilistercora, Tellurirhabdus, Blattabacterium, Koleobacter* and *Faecalitalea*. The labels were used to calculate percent Gram-positive, Gram-negative, unknown or unclassified for the two types of measurement.

## Cell type deconvolution

We employed the cell2location[28] model (version 0.1.3) to deconvolve spatial transcriptomics generated with the Visium and Stereo-seq platforms. The scRNA-seq reference, from a previous study on Apc Min/+ mice[27] was filtered to include only genes that are highly expressed and informative for identifying rare cell types, with thresholds set at cell_count_cutoff of 5, cell_percent_cutoff of 0.01 and nonz_mean_cutoff of 1.12. Cell-type-specific expression signatures were generated using negative binomial regression from these selected genes and applied to the spatial transcriptomics data to determine cell-type identities, with the highest prediction scores used for assignment. The detection_alpha parameter was set to 20 and N_cells_per_location was set to 30 for Visium and 1 for Stereo-seq.

## Bacterial gene function analysis

Bacterial reference resources comprised selected abundant bacterial genomes (accession IDs GCF_037113525.1 and GCF_025148285.1) and bacterial genes downloaded from NCBI (downloaded 18 September 2025); for the genes, we retrieved and used all sequences annotated under the names atpA, enolase, tufA, eno, gap, groL, msmX, pckA, ppdK and spoIVCA. Whole-genome FASTA files were annotated with Prokka (v[1.14.5]) using default parameters to produce GFF3 feature annotations (genes, rRNA/tRNA, CDS and product fields). Unmapped reads from the alignment to the host genome were mapped to the bacterial references using Bowtie2 (v[2.5.1]) with default parameters after building indexes via bowtie2-build. Genome-wide mapping summaries were visualized with pycirclize (v[1.6.0]) as Circos-style plots. Tracks included (1) annotated ribosomal RNA/protein genes from the Prokka GFF, (2) optional GC/AT content and (3) 1-kb binned coverage. Values used for plotting were clipped to the predefined display range to avoid axis boundary artefacts. The per-base depth was computed from sorted BAMs with samtools depth -aa, then aggregated into non-overlapping 1-kb windows (mean per window) and written as bedGraph (chrom, start, end, mean). Using these bins (midpoints as genomic positions), we optionally smoothed coverage (window of 1 bin; none) and called peaks with scipy.signal.find_peaks (prominence of 0.2 × 10th percentile of positive bins; minimum distance of 5 bins is -5 kb). Prokka GFF3 annotations (gene/CDS/rRNA/tRNA) were parsed, and multisegment features with the same name were merged (minimum start, maximum end). Each peak was labelled overlap if its bin intersected any feature (all overlaps recorded, one

representative chosen) or nearest otherwise (nearest feature and distance reported)[63,64].

## Spatial autocorrelation analysis

Moran's I was calculated for the major genera (abundance >0.01%) using the Moran function from the Python library pysal. Spatial weights were generated using the $k$-nearest neighbours matrix ($k = 4$) from the weights module in pysal. For genera with a Moran's I $P$ value <0.05, Ripley's $H$ was subsequently derived using the following formula:

$$K(r) = \frac{2A}{N(N-1)} \sum_{i=1}^{N} \sum_{j=i+1}^{N} I(d_{ij} \leq r)$$

$$L(r) = \sqrt{\frac{K(r)}{\pi}}$$

$$H(r) = L(r) - r$$

where $d_{ij}$ is the Euclidean distance between points $i$ and $j$. $I(d_{ij} \leq r)$ is an indicator function that is 1 if the distance $d_{ij}$ is less than or equal to $r$ and 0 otherwise. $A$ is the area of the observation window. $N$ is the number of points in the dataset.

## Spatial colocalization analysis

To assess the spatial colocalization of bacterial genera in the Stereo-seq datasets, we applied the Smoothie package using default parameters for the Stereo-seq platform[30]. In brief, bacterial spatial data classified at the genus level by Kraken2 were spatially smoothed (20-µm Gaussian kernel), and pairwise correlations were computed among all resulting bacterial signal surfaces to quantify colocalization patterns across the tissue area.

## Spatial clustering of microbial signal (HDBSCAN[65]) and contouring

To assess the spatial clustering characteristics of a given bacterial taxa, we subset spots with nonzero counts and used their spatial coordinates (obsm['spatial']) as input to HDBSCAN (v[0.8.38]). To choose hyperparameters, we performed a grid search over min_cluster_size and min_samples. For each setting we fit HDBSCAN, removed noise labels (−1) and clusters with size <min_cluster_size and computed the silhouette score on the remaining points. The best setting was selected by the maximum silhouette score. HDBSCAN was called with Euclidean distance and default options unless stated; noise points (label of −1) were excluded from silhouette computation. Silhouette scores were computed with scikit-learn (v[1.3.1]).

## Boundary detection

The microscope data were saved in greyscale and then averaged using the OpenCV blur function with a kernel size 100 µm. After that, the data were binarized with a threshold of 80 for normal tissue and 100 for cancer tissue. Finally, boundaries were extracted using the OpenCV findContours function.

## Reporting summary

Further information on research design is available in the Nature Portfolio Reporting Summary linked to this article.

## Data availability

Data will be made available upon publication under GEO accession numbers; GSE276866 for the low-resolution datasets, GSE277196, GSE277197, GSE308507, GSE316608 and GSE316962 for the high-resolution datasets, and GSE316869 for the bulk RNA-seq datasets.

## Code availability

Code associated with this work can be found via GitHub at https://github.com/ntekasi/microSTRS.

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

## Acknowledgements

We thank the Cornell Biotechnology Resource Center and Logan Schiller for their help with sequencing the libraries. We thank the Cornell Center for Animal Resources and Education for animal housing and care. We thank M. Mantri, S. Jiang, M. Wang, R. Agarwal and the other members of the De Vlaminck lab for helpful discussions and feedback. We also thank J. Jones, S. Asami and W. Suda for the helpful discussion. This work was supported by the WE&ME foundation and NIH grants R01AI176681, R01AR081449 and R37AI189855 to I.D.V.

## Author contributions

I.N., L.T., D.W.M. and I.D.V. conceived of the study. I.N., L.T., P.S., B.G. and Q.S. performed the experiments. I.N., L.T., D.W.M., C.H., Y.S.P. and M.S. analysed the data. I.N., L.T. and I.D.V. wrote the manuscript. All authors provided input and comments.

## Competing interests

D.W.M., I.N. and I.D.V. have filed a patent on technology described in this work. The other authors declare no competing interests.

## Additional information

**Extended data** is available for this paper at https://doi.org/10.1038/s41564-026-02286-7.

**Correspondence and requests for materials** should be addressed to Iwijn De Vlaminck.

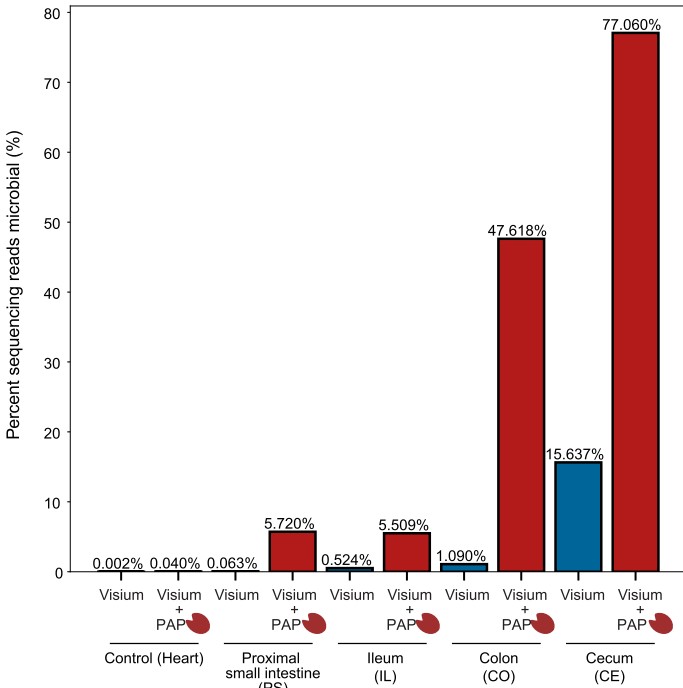

**Extended Data Fig. 1 | Percentage of sequencing reads classified as Microbial.** The bar plots show the percentage of reads that did not map to the mouse genome and were classified as microbial (belonging to Bacteria, Archaea, or Viruses superkingdoms) for the non-intestine control/ Murine Heart samples Visium and Visium with *in situ* Polyadenylation, and the paired experiments for the 4 examined sites of the gastrointestinal tract. The samples with *in situ* polyadenylation are marked with a red enzyme cartoon.

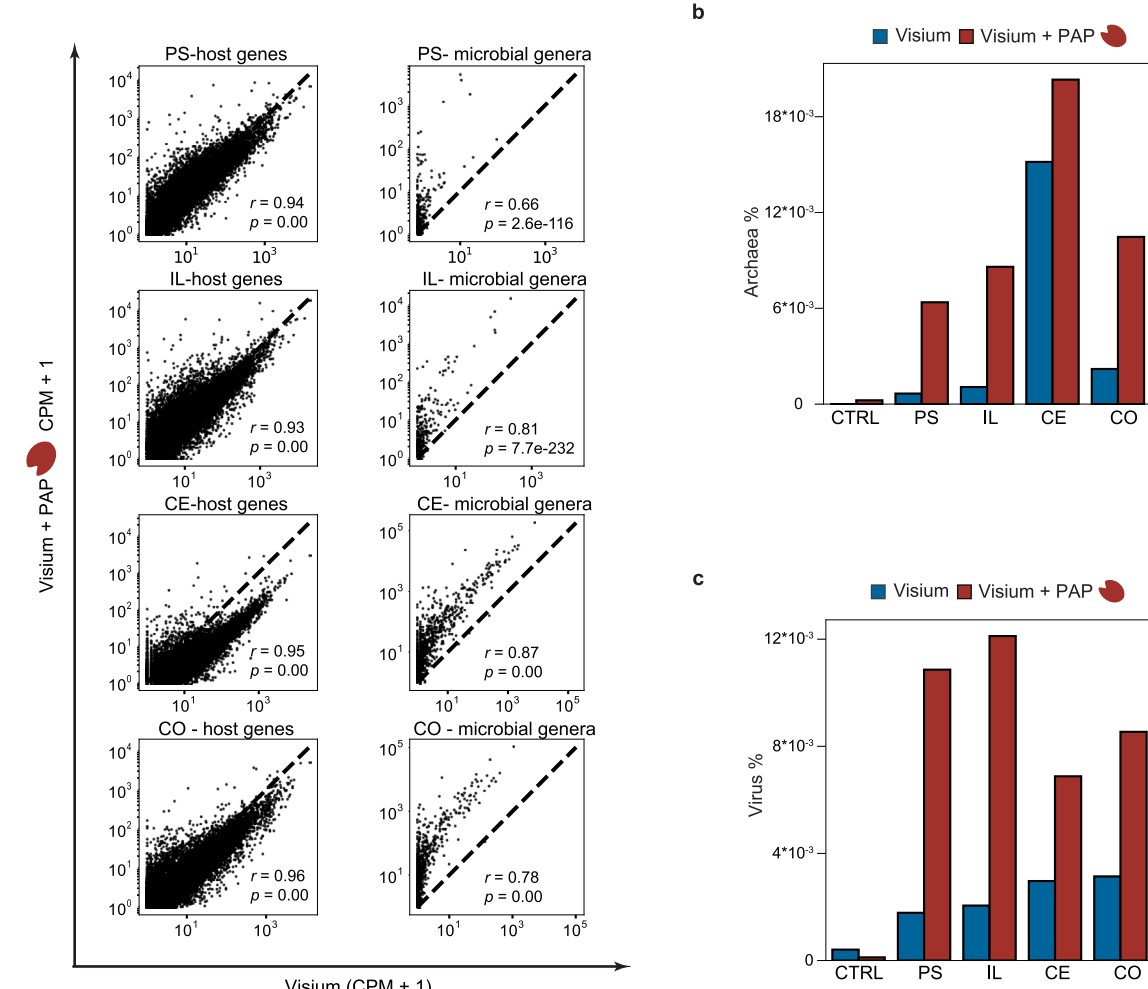

**Extended Data Fig. 2 | Enrichment of microbial signal capture via *in situ* polyadenylation.** (**a**) Scatter plots showing the Aggregated unique molecules per million for host genes (left) and bacterial taxa (right) for the tissue pairs. Pearson correlation coefficient *r* and *p*-value are noted on each scatterplot (**b**) Percent of Unique molecules captured classified as archaea for the paired experiments with (red) and without (blue) *in situ* polyadenylation for the Murine heart and the 4 sampling tissue sites. (**c**) Percent of Unique molecules captured classified by Kraken2 as viruses for the paired experiments with (red) and without (blue) *in situ* polyadenylation for the Murine heart and the 4 sampling tissue sites.

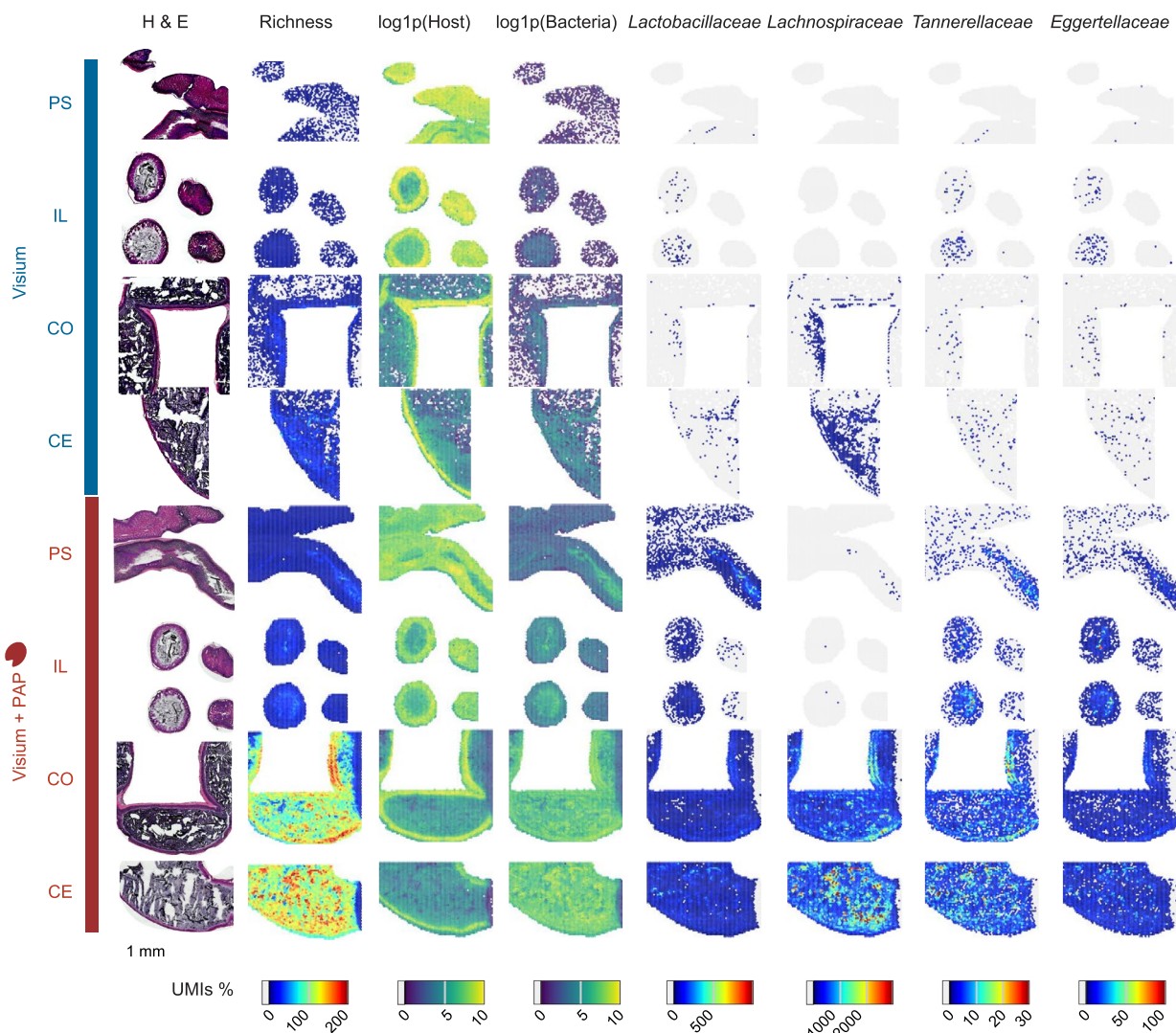

**Extended Data Fig. 3 | Spatial maps for the paired Visium and STRS assays.** Spatial feature plots and histological data for the tissues processed with the low-resolution platform (with and without polyadenylation).

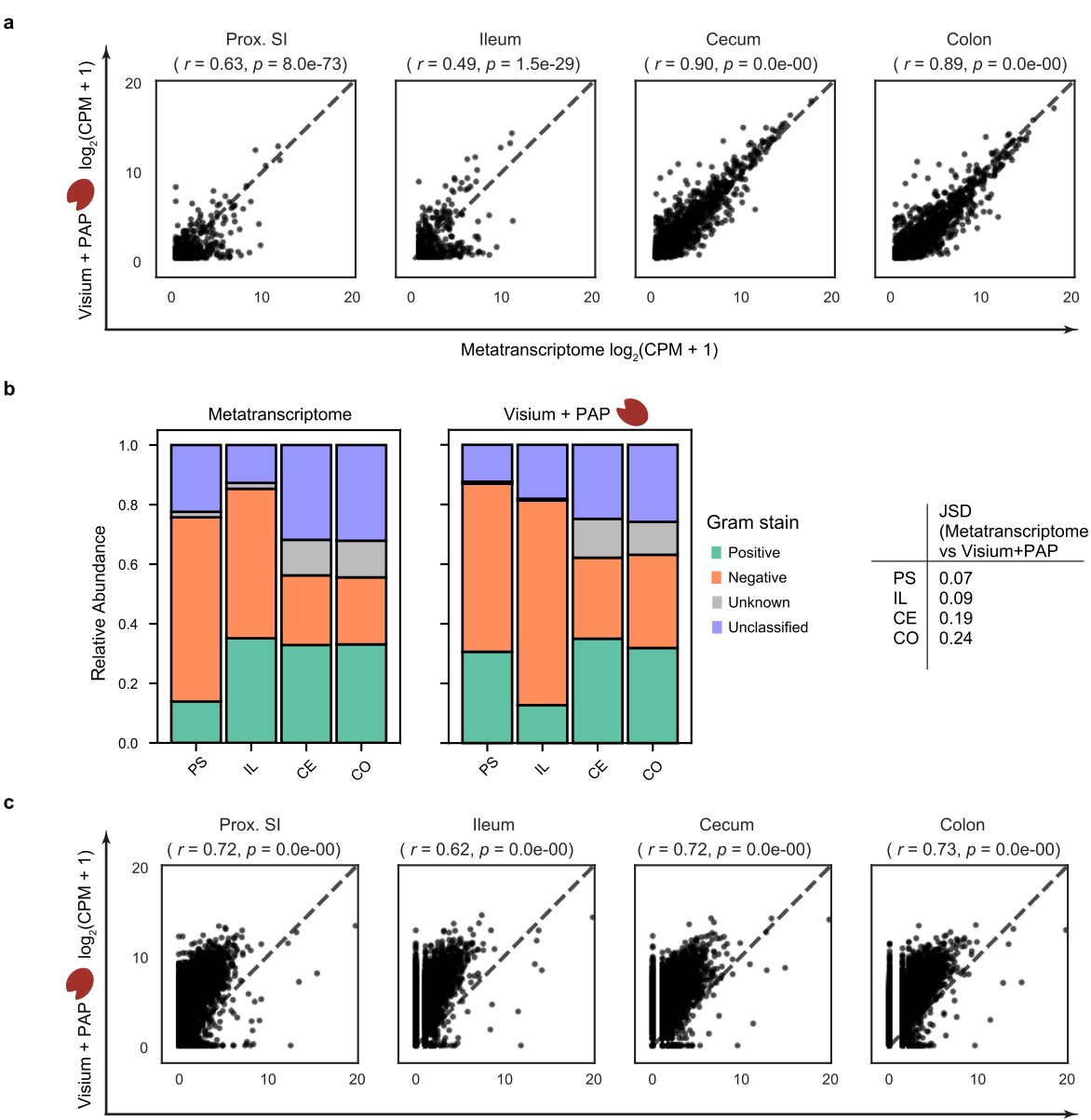

**Extended Data Fig. 4 | Orthogonal validation of the Visium with *in situ* Polyadenylation sequencing libraries.** (**a**) Scatter plot of genus-level concordance between bulk metatranscriptomes and spatial libraries (Visium + *in situ* polyadenylation, PAP) from proximal sister sections of each gut region (PS, IL, CE, CO). Each point is a Kraken-classified genus; axes show log2(CPM + 1). Dashed line, $x = y$; Pearson's r and two-sided p-values are shown. (**b**) Relative Gram-stain composition across regions for the paired bulk metatranscriptome (left) and spatial (middle) datasets. (right) Jensen–Shannon divergence (JSD) between metatranscriptomics and Visium+PAP spatial assays. (**c**) Scatter plot comparing host gene expression in bulk RNA-seq and Visium+PAP from proximal sister sections (PS, IL, CE, CO). Each point is a gene; axes are log$_2$(CPM + 1), computed per sample from host gene counts. Dashed line denotes $x = y$; Pearson's r and two-sided p-values are shown.

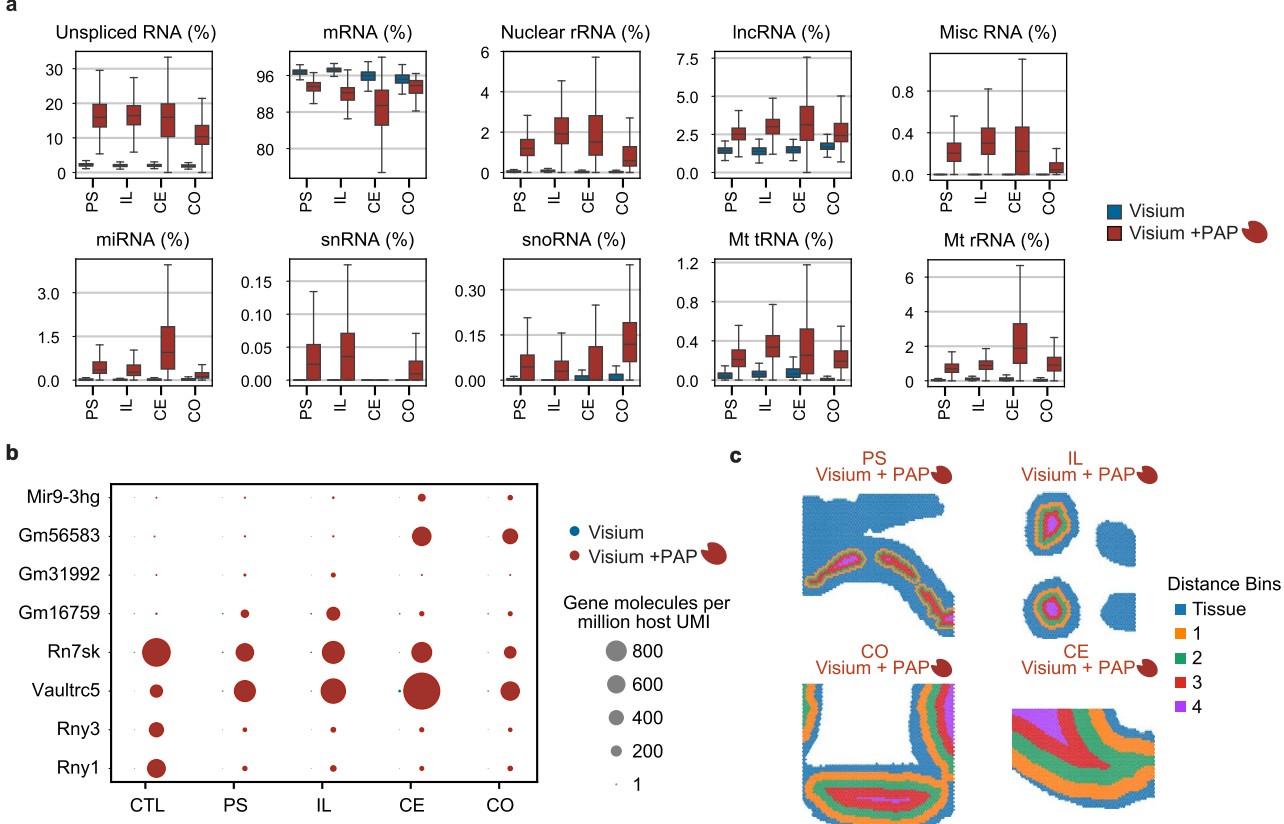

**Extended Data Fig. 5 | RNA biotype distributions for the paired Visium and STRS experiments.** (**a**) Boxplots representing the distributions of select RNA biotypes across the 4 sampling sites. (**b**) Dotplot of selected genes that are common to all four GI tract regions (PS, IL, CE,CO) and murine heart tissue (CTL). (**c**) Maps showing the spatial bins used for the relative abundance calculation.

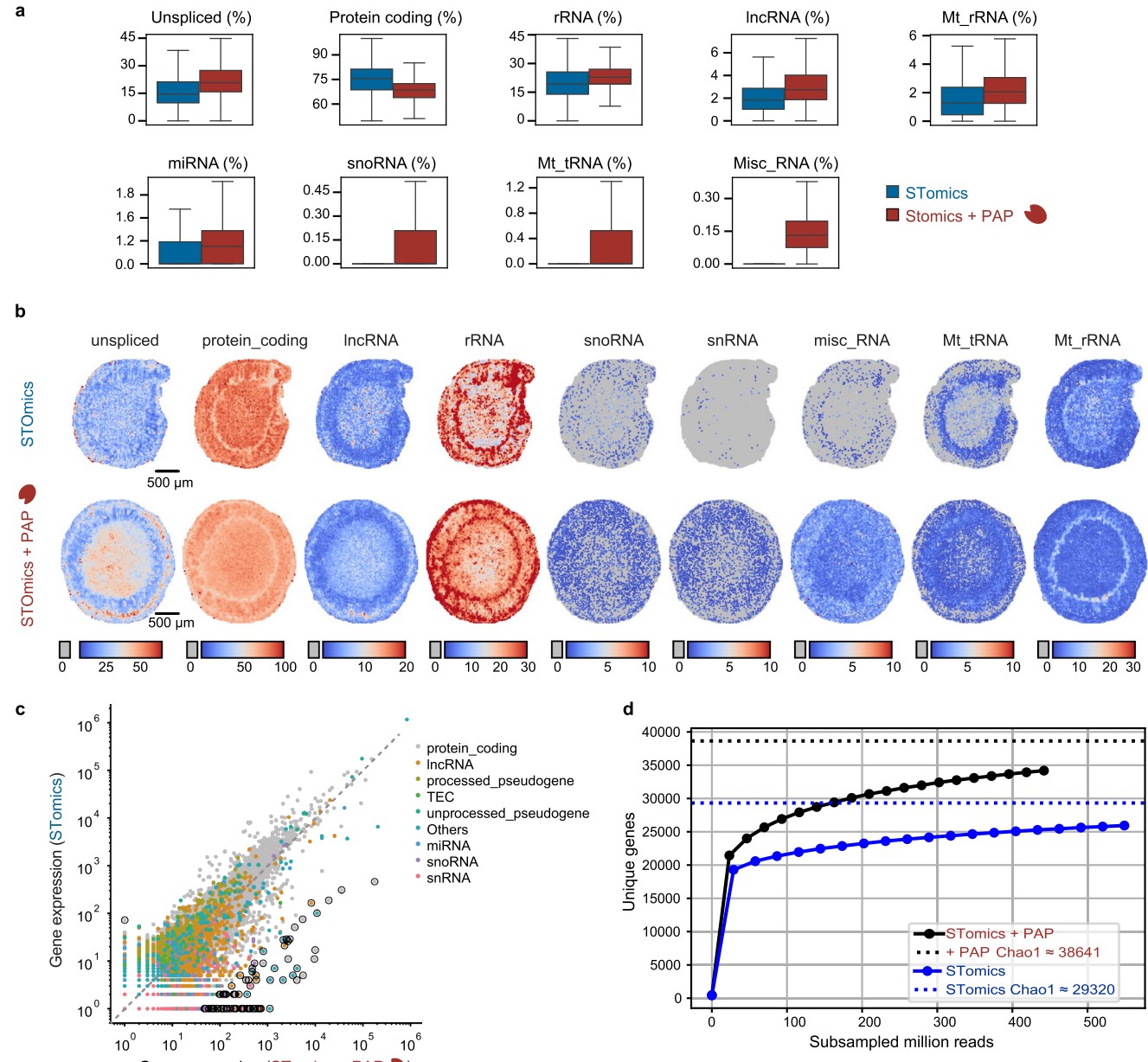

**Extended Data Fig. 6 | RNA biotype distributions for the paired Stereoseq and STRS (Stereoseq + PAP) experiments.** (**a**) Boxplots representing the amount of select RNA biotypes (**b**) Spatial maps showing the patterns of the biotypes across the paired experiments. (**c**) Scatter plot showing the rescaled sum of unique molecules per million for host genes. The gene expressions were linearly rescaled so that the sum of each expression matched the mean of the sums. (**d**) Rarefaction curves showing Unique genes as a function of Subsampled million reads for the STOmics and STOmics with in situ polyadenylation experiments (STOmics + PAP; red) Chao1 estimates are denoted with dashed lines.

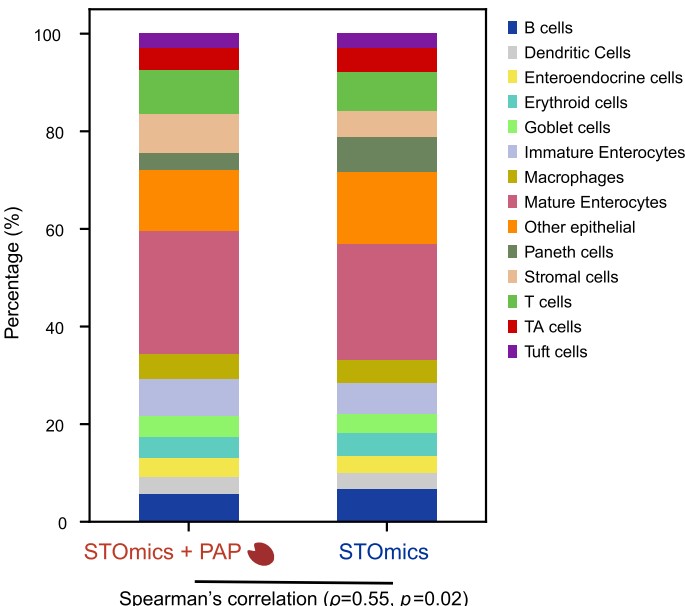

Spearman's correlation ($\rho$=0.55, $p$=0.02)
Pearson's correlation ($r$=0.97, $p$=2.33x10$^{-10}$)

**Extended Data Fig. 7 | Cell type composition for Stereoseq and STRS (Stereoseq + PAP).** Cell type composition for the paired Stereo-seq + PAP (red) and Stereo-seq (blue) spatial experiments as predicted with deconvolution.

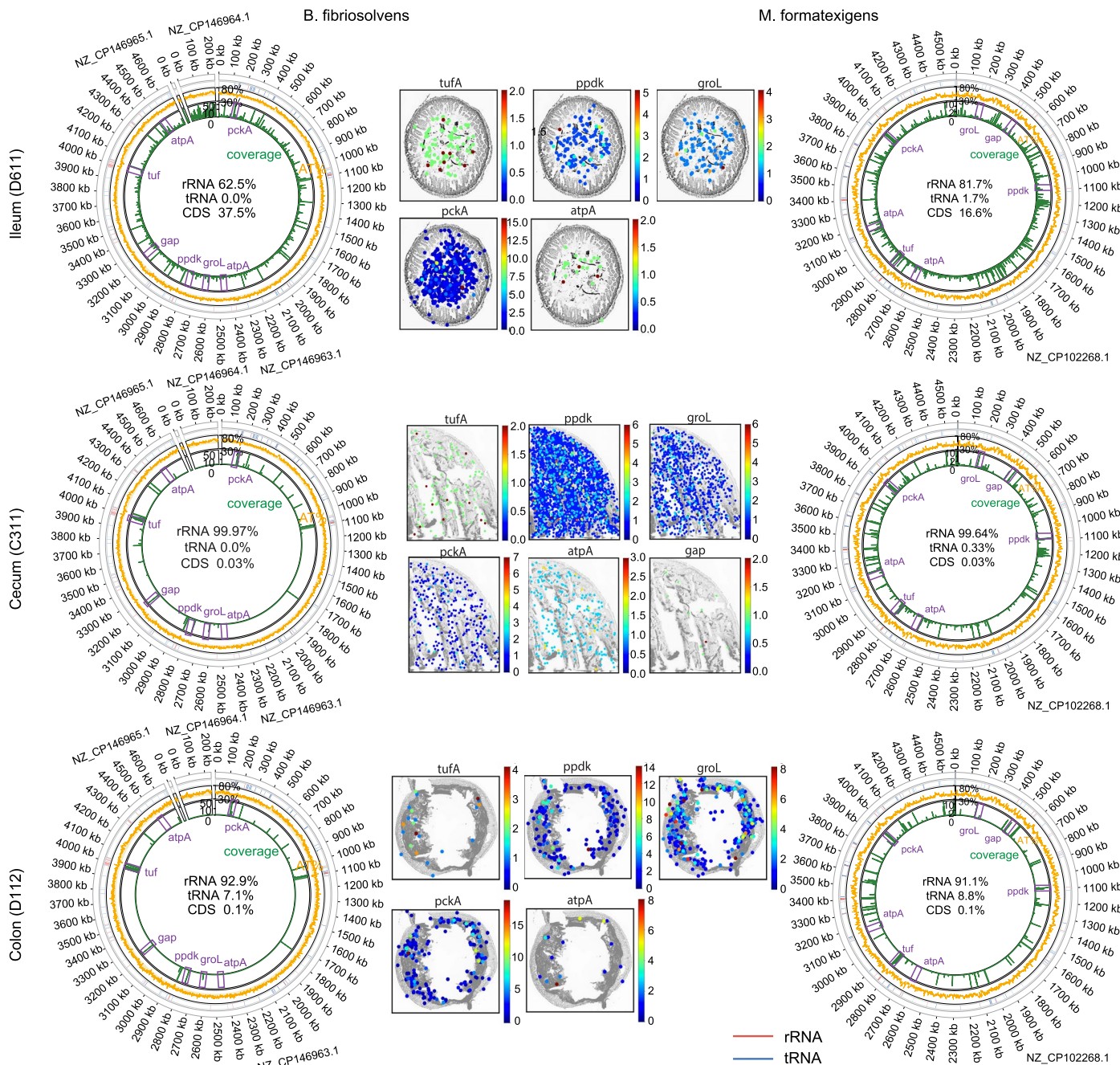

**Extended Data Fig. 8 | Circular data visualization of bacterial gene expression.** Circular plots showing sequencing depth across the B. fibriosolvens and M. formatexigens genomes and sample spatial maps of identified bacterial genes for three intestinal locations. For the circular plots, genome coordinates are shown in the outer ring with select protein-coding genes denoted in purple, rRNA loci in red and tRNA in blue. The orange track indicated AT% and the green depicts sequencing coverage.

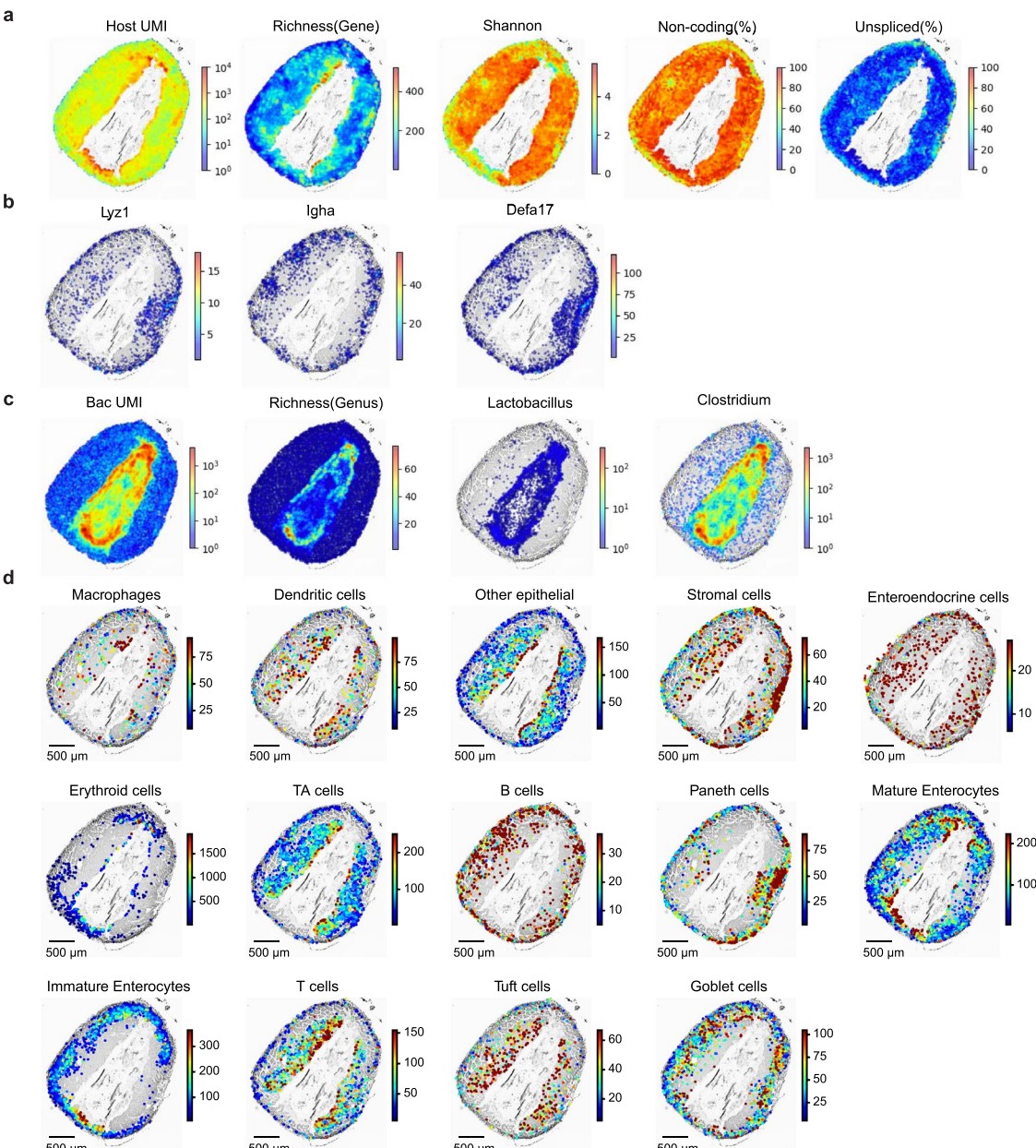

**Extended Data Fig. 9 | Spatial mapping of cancer section. (a)** Spatial mapping of host UMIs, gene richness, unspliced molecule and non-coding gene ratio in 20 μm square bin, **(b)** Spatial mapping of ratio of selected gene expressions in 20 μm square bin. **(c)** Spatial mapping of bacterial UMIs, gene richness in 20 μm square bin. **(d)** The spatial mapping of each cell type. Color scale represents total gene expression for each cell within the plotted cell type.

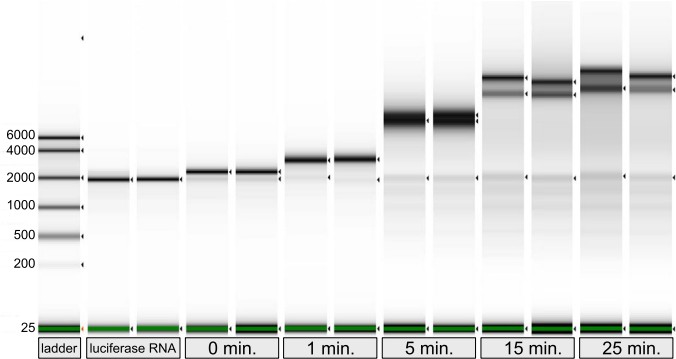

**Extended Data Fig. 10 | Poly(A) tailing time-course assay.** A 1.8 kb in vitro transcribed luciferase RNA was incubated with yeast Poly(A) Polymerase (yPAP) for the indicated times (0–25 min). Reaction products were purified and analyzed on an Agilent Tapestation. The 0 min condition represents a control where the reactions were kept on ice.

# Reporting Summary

## Statistics

For all statistical analyses, confirm that the following items are present in the figure legend, table legend, main text, or Methods section.

| n/a | Confirmed | |
|---|---|---|
| ☐ | ☒ | The exact sample size (*n*) for each experimental group/condition, given as a discrete number and unit of measurement |
| ☒ | ☐ | A statement on whether measurements were taken from distinct samples or whether the same sample was measured repeatedly |
| ☒ | ☐ | The statistical test(s) used AND whether they are one- or two-sided *Only common tests should be described solely by name; describe more complex techniques in the Methods section.* |
| ☒ | ☐ | A description of all covariates tested |
| ☒ | ☐ | A description of any assumptions or corrections, such as tests of normality and adjustment for multiple comparisons |
| ☒ | ☐ | A full description of the statistical parameters including central tendency (e.g. means) or other basic estimates (e.g. regression coefficient) AND variation (e.g. standard deviation) or associated estimates of uncertainty (e.g. confidence intervals) |
| ☒ | ☐ | For null hypothesis testing, the test statistic (e.g. *F*, *t*, *r*) with confidence intervals, effect sizes, degrees of freedom and *P* value noted *Give P values as exact values whenever suitable.* |
| ☒ | ☐ | For Bayesian analysis, information on the choice of priors and Markov chain Monte Carlo settings |
| ☒ | ☐ | For hierarchical and complex designs, identification of the appropriate level for tests and full reporting of outcomes |
| ☒ | ☐ | Estimates of effect sizes (e.g. Cohen's *d*, Pearson's *r*), indicating how they were calculated |

*Our web collection on statistics for biologists contains articles on many of the points above.*

## Software and code

Policy information about availability of computer code

| Data collection | No software was used to collect data in this study. |
|---|---|
| Data analysis | Preprocessing and alignment of spatial transcriptomics data<br>To ensure similar alignment and quantification across platforms and methodologies we used the "slide_snake" pipeline that utilizes Snakemake53 (6.1.0), which can be found on github (https://github.com/mckellardw/slide_snake). For the Visium and STRS (Visium) libraries, the pipeline first trims poly(A) and poly(G) sequences, as well as primer sequences using cutadapt54. The reads were aligned using STAR v2.7.10a55 and STARSolo56 (specified parameters: --outFilterMultimapNmax 50, --soloMultiMappers EM, --clipAdapterType CellRanger4) to generate expression matrices for every sample. For downstream analyses the GeneFull matrices were used. Barcode whitelists and the associated spot spatial locations for Visium data were copied from the Space Ranger software ("Visium-v1_coordinates.txt"). For the StereoSeq and STRS (StereoSeq) libraries, barcode maps were provided by the manufacturer as .h5 files and converted to text format using ST_BarcodeMap (https://github.com/STOmics/ST_BarcodeMap). Alignment references were generated from the GRCm39 reference sequence using GENCODE M32 annotations.<br><br>Unmapped reads classification and construction of microbiome Anndata objects<br>In this study, to classify reads of microbial origin out of the unmapped reads we utilized Kraken2 (version 2.09)21. We used the standard Kraken2 database supplemented with the mouse genome. Unmapped reads flagged in the BAM file were processed to retain the correct cell barcode and unique molecular identifier (UMI) information as identified by STARsolo. This allowed for the demultiplexing of Kraken2 output by cell barcode and UMI. For data integration, we employed Pandas, Scanpy, NumPy, Scipy, and regular expressions to create an AnnData object with cell barcodes as observations and NCBI taxonomy IDs as features. Only classified reads were retained for subsequent analysis.<br><br>Image Registration and Cell Segmentation |

For Visium and Visium+PAP samples, image registration was performed using the 10x Genomics Loupe Browser. Hematoxylin and eosin (H&E)-stained tissue images were aligned to the spatial capture array, and regions corresponding to tissue and lumen were manually annotated. For Stereo-seq samples, nuclear-stained fluorescence images were acquired during sample processing. Image registration and cell segmentation were carried out using the Stereo-seq Analysis Workflow (SAW) provided by STOmics. Fluorescence images were aligned to the chip layout based on the ChipID metadata. Following registration, automated cell segmentation was performed using SAW's built-in algorithms. Segmentation masks were used to define cell boundaries, and barcodes within each segmented region were aggregated to construct single-cell transcriptomes. These cell-level datasets were then used for downstream spatial analysis and deconvolution.

Processing and Alignment of Metatranscriptomic Libraries
Metatranscriptomic sequencing data were processed using a custom computational workflow. Adapter trimming and quality filtering were performed using BBDuk (v38.90). Filtered reads were aligned to the mouse genome (GRCm39) with STAR (v2.7.10a) using GENCODE M32 annotations. Gene-level quantification was carried out with featureCounts (v2.0.0). Duplicate reads were identified and marked using Picard MarkDuplicates (v2.19.2). Reads that did not align to the host genome were extracted and taxonomically classified using Kraken2 (v2.0.9).

Sterile control pre-processing and identification of taxa to filter
To assess the Kraken2 classified microbial counts occurring in non-intestinal tissues for the low-resolution platform we re-aligned previously published Visium and STRS libraries of mock-infected C57BL/6J 11 days year old mice with and without polyadenylation as described in the corresponding studies14,20. 85 taxa occurring at 1ppm (UMI) or greater were excluded from downstream analysis as potential misclassification. For the Stereo-seq libraries, a sterile control experiment was conducted. Briefly, fresh-frozen heart from a eleven day old mouse were sectioned on a Stereo-seq 1cm x 1cm tile (STOmics, BGI). The sample was fixed in methanol at -20°C for 20 minutes followed by the in situ polyadenylation and the Stereo-seq library preparation protocol as described above. Taxa occurring at frequencies higher than 1 ppm UMI were excluded from downstream analyses.

Pre-processing of the Visium and Visium + PAP data
Spatial coordinates were assigned to the Visium and Visium + PAP library spots based on the barcode map provided by the Space Ranger software ("Visium-v1_coordinates.txt"). The accompanying hematoxylin and eosin histology images of each experiment were used to manually mark the spots that correspond to tissue and lumen. Scanpy57, mudata58,59, and muon58 were used to construct multimodal objects separately for the microbial maps (in the taxonomic levels of phylum, family, genus, and species). This was done for each one of the accounted microbial superkingdoms of Archaea, Bacteria and Viruses. For downstream analyses, only the spots covered by tissue or corresponding to lumen were accounted for.

Microbial percentage and enrichment calculation for the paired Visium and Visium + PAP experiments
For the three discussed superkingdoms, the percentage of reads falling under to a superkingdom classification was calculated as the percentage of Kraken-classified reads that belong to the superkingdom over the total counts of the library defined as the sum of unique molecules aligned to the host and unique molecules classified by Kraken2. The enrichment for each paired experiment was defined as the ratio of those percentages.

Relative abundance and bacterial richness calculations for the low-resolution datasets
To calculate the relative abundance for each examined sample, at family level, the corresponding family reads were collapsed and divided by the total molecules originating from bacteria as classified by Kraken2. The microbial richness per spot was calculated as the number of unique taxa occurring per spot after the exclusion of taxa accounting for 0.01% or less of microbial molecules in the whole sample. For the transverse axis relative abundance analysis, cells were spatially binned from the tissue to the lumen based on their minimum distance to the lumen-associated region. Phyla relative abundance data were then aggregated within each bin to quantify relative abundances across the tissue-lumen axis.

Rarefaction and Sequencing Saturation Analysis
Rarefaction analysis was performed on paired Visium and Visium+PAP libraries collected from four intestinal regions (proximal small intestine, ileum, cecum, and colon). Sequencing data were subsampled at defined fractions (10% to 100% of total reads). After alignment to the mouse genome using the Snakemake-based pipeline, unmapped reads were extracted and taxonomically classified using Kraken2. AnnData objects were generated at multiple taxonomic levels as described for the full dataset. For classified microbial reads in each condition, rarefaction curves were generated by plotting the number of unique bacterial molecules against sequencing depth. Michaelis-Menten models were fitted using non-linear least squares to estimate theoretical saturation behavior, and model fit was evaluated using the coefficient of determination ($R^2$). To calculate saturation for the total library, as well as for host- and microbiome-derived reads, the following general formula was used:

Here, modality refers to the source of the molecules: host-aligned reads, microbial reads classified by Kraken2, or all reads combined. This metric captures diminishing returns in the recovery of unique molecules with increasing sequencing depth. Saturation was computed at each subsampling level and modeled using Michaelis-Menten kinetics constrained to a maximum of 1. The resulting curves were used to estimate the sequencing depth required to achieve saturation (e.g., Saturation ≈ 0.9).

Gram stain comparison of Visium + PAP to the bulk RNA measurement
Adjacent samples profiled by Visium+PAP and by bulk metatranscriptomics were compared at the genus level. Reads were taxonomically classified with Kraken2 and counts were aggregated by genus. The same taxa excluded in spatial control analyses were removed prior to comparison. Gram-stain labels for genera were retrieved from BacDive60 and were curated to fill missing entries for highly abundant genera. The curated set included the Gram-negative: Pseudoprevotella, Hoylesella, Segatella, Allomuricauda, Marvinbryantia, Lachnoclostridium, Vescimonas, Mediterraneibacter, Coprococcus, Caproicibacterium, Massilistercora, Tellurirhabdus, Blattabacterium, Koleobacter, and Faecalitalea. The labels were used to calculate percent Gram-positive, Gram-negative, unknown or unclassified for the two types of measurement.

Cell type deconvolution
We employed the cell2location27 model (version 0.1.3) to deconvolve spatial transcriptomics data for the experiments conducted with both Visium and Stereo-seq technologies. The scRNA-seq reference, derived from a previous study on Apc Min/+ mice26 was filtered to include only genes that are highly expressed and informative for identifying rare cell types, with thresholds set at cell_count_cutoff = 5, cell_percent_cutoff = 0.01, and nonz_mean_cutoff = 1.12. Cell-type-specific expression signatures were generated using negative binomial regression from these selected genes. These signatures were applied to the spatial transcriptomics data to determine cell-type identities, with

the highest prediction scores used for assignment. For Visium, we set N_cells_per_location to 30, and for Stereo-seq, we set it to 1, with the detection_alpha parameter set to 20 in both cases.

Bacterial gene function analysis
Bacterial reference resources comprised selected abundant bacterial genomes (accession IDs: GCF_037113525.1, GCF_025148285.1) and bacterial genes downloaded from NCBI (downloaded 2025-09-18); for the genes, we retrieved and used all sequences annotated under the names atpA, enolase, tufA, eno, gap, groL, msmX, pckA, ppdK, and spoIVCA. Whole-genome FASTA files were annotated with Prokka (v[1.14.5]) using default parameters to produce GFF3 feature annotations (genes, rRNA/tRNA, CDS, and product fields). Unmapped reads from the alignment to the host genome were mapped to the bacterial references using Bowtie2 (v[2.5.1]) with default parameters after building indexes via bowtie2-build. Genome-wide mapping summaries were visualized with pycirclize (v[1.6.0]) as Circos-style plots. Tracks included (i) annotated ribosomal RNA/protein genes from the Prokka GFF, (ii) optional GC/AT content, and (iii) 1 kb binned coverage. Values used for plotting were clipped to the predefined display range to avoid axis boundary artefacts. The per-base depth was computed from sorted BAMs with samtools depth -aa, then aggregated into non-overlapping 1 kb windows (mean per window) and written as bedGraph (chrom, start, end, mean). Using these bins (midpoints as genomic positions), we optionally smoothed coverage (window = 1 bin; none) and called peaks with scipy.signal.find_peaks (prominence = 0.2 × 10th percentile of positive bins; minimum distance = 5 bins ≈ 5 kb). Prokka GFF3 annotations (gene/CDS/rRNA/tRNA) were parsed, and multi-segment features with the same name were merged (min start, max end). Each peak was labeled overlap if its bin intersected any feature (all overlaps recorded, one representative chosen) or nearest otherwise (nearest feature and distance reported).

Spatial autocorrelation analysis
Moran's I was calculated for the major genera (abundance > 0.01%) using the Moran function from the Python library pysal. Spatial weights were generated using the -nearest neighbors (KNN) matrix ( =4) from the weights module in pysal. For genera with a Moran's I p-value < 0.05, Ripley's H was subsequently derived using the formula written on the manuscript.

Spatial co-localization analysis
To assess the spatial co-localization of bacterial genera in the Stereo-seq datasets, we applied the Smoothie package using default parameters for the Stereo-seq platform29. Briefly, bacterial spatial data classified at the genus level by Kraken2 were spatially smoothed (20 µm Gaussian kernel), and pairwise correlations were computed among all resulting bacterial signal surfaces to quantify co-localization patterns across the tissue area.

Spatial clustering of microbial signal (HDBSCAN31) and contouring
To assess the spatial clustering characteristics of a given bacterial taxa, we subset spots with nonzero counts and used their spatial coordinates (obsm['spatial']) as input to HDBSCAN (v[0.8.38]). To choose hyperparameters, we performed a grid search over min_cluster_size and min_samples. For each setting we fit HDBSCAN, removed noise labels (-1) and clusters with size < min_cluster_size, and computed the silhouette score on the remaining points. The best setting was selected by the maximum silhouette score.HDBSCAN was called with Euclidean distance and default options unless stated; noise points (label = −1) were excluded from silhouette computation. Silhouette scores were computed with scikit-learn (v[1.3.1]).

Boundary detection
The microscope data was saved in grayscale and then averaged using the OpenCV blur function with a kernel size 100 µm. After that, the data was binarized with a threshold of 80 for normal tissue and 100 for cancer tissue. Finally, boundaries were extracted using the OpenCV findContours function.

Code associated with this work can be found at https://github.com/ntekasi/microSTRS.

For manuscripts utilizing custom algorithms or software that are central to the research but not yet described in published literature, software must be made available to editors and reviewers. We strongly encourage code deposition in a community repository (e.g. GitHub). See the Nature Portfolio guidelines for submitting code & software for further information.

# Data

Policy information about availability of data

All manuscripts must include a data availability statement. This statement should provide the following information, where applicable:
- Accession codes, unique identifiers, or web links for publicly available datasets
- A description of any restrictions on data availability
- For clinical datasets or third party data, please ensure that the statement adheres to our policy

Data will be made available upon publication under GEO accession numbers; GSE276866 for the low-resolution datasets, GSE277196, GSE277197, GSE308507, GSE316608 and GSE316962 for the high-resolution datasets, and GSE316869 for the bulk RNA-seq datasets.

# Research involving human participants, their data, or biological material

Policy information about studies with human participants or human data. See also policy information about sex, gender (identity/presentation), and sexual orientation and race, ethnicity and racism.

| Reporting on sex and gender | *Use the terms sex (biological attribute) and gender (shaped by social and cultural circumstances) carefully in order to avoid confusing both terms. Indicate if findings apply to only one sex or gender; describe whether sex and gender were considered in study design; whether sex and/or gender was determined based on self-reporting or assigned and methods used.* |
|---|---|
| | *Provide in the source data disaggregated sex and gender data, where this information has been collected, and if consent has been obtained for sharing of individual-level data; provide overall numbers in this Reporting Summary. Please state if this information has not been collected.* |
| | *Report sex- and gender-based analyses where performed, justify reasons for lack of sex- and gender-based analysis.* |
| Reporting on race, ethnicity, or | *Please specify the socially constructed or socially relevant categorization variable(s) used in your manuscript and explain why* |

| Reporting on race, ethnicity, or other socially relevant groupings | *they were used. Please note that such variables should not be used as proxies for other socially constructed/relevant variables (for example, race or ethnicity should not be used as a proxy for socioeconomic status). Provide clear definitions of the relevant terms used, how they were provided (by the participants/respondents, the researchers, or third parties), and the method(s) used to classify people into the different categories (e.g. self-report, census or administrative data, social media data, etc.) Please provide details about how you controlled for confounding variables in your analyses.* |
|---|---|
| Population characteristics | *Describe the covariate-relevant population characteristics of the human research participants (e.g. age, genotypic information, past and current diagnosis and treatment categories). If you filled out the behavioural & social sciences study design questions and have nothing to add here, write "See above."* |
| Recruitment | *Describe how participants were recruited. Outline any potential self-selection bias or other biases that may be present and how these are likely to impact results.* |
| Ethics oversight | *Identify the organization(s) that approved the study protocol.* |

Note that full information on the approval of the study protocol must also be provided in the manuscript.

# Field-specific reporting

Please select the one below that is the best fit for your research. If you are not sure, read the appropriate sections before making your selection.

☒ Life sciences ☐ Behavioural & social sciences ☐ Ecological, evolutionary & environmental sciences

For a reference copy of the document with all sections, see nature.com/documents/nr-reporting-summary-flat.pdf

# Life sciences study design

All studies must disclose on these points even when the disclosure is negative.

| Sample size | One sample per condition was collected using the protocol in this manuscript, but eight total samples were collected for low-resolution analysis and three more samples were collected for high-resolution analysis. |
|---|---|
| Data exclusions | For downstream analysis, data from outside the tissues areas were excluded when noted. |
| Replication | We processed 18 samples in total using the method described in this manuscript, and assessed quality of each sample to ensure the repeatability of Spatial Total RNA-Sequencing. |
| Randomization | NA |
| Blinding | NA |

# Reporting for specific materials, systems and methods

We require information from authors about some types of materials, experimental systems and methods used in many studies. Here, indicate whether each material, system or method listed is relevant to your study. If you are not sure if a list item applies to your research, read the appropriate section before selecting a response.

## Materials & experimental systems

| n/a | Involved in the study |
|---|---|
| ☒ | ☐ Antibodies |
| ☒ | ☐ Eukaryotic cell lines |
| ☒ | ☐ Palaeontology and archaeology |
| ☐ | ☒ Animals and other organisms |
| ☒ | ☐ Clinical data |
| ☒ | ☐ Dual use research of concern |
| ☒ | ☐ Plants |

## Methods

| n/a | Involved in the study |
|---|---|
| ☒ | ☐ ChIP-seq |
| ☒ | ☐ Flow cytometry |
| ☒ | ☐ MRI-based neuroimaging |

## Animals and other research organisms

Policy information about studies involving animals; ARRIVE guidelines recommended for reporting animal research, and Sex and Gender in Research

| Laboratory animals | All animal protocols were approved by the Cornell University Institutional Animal Care and Use Committee (IACUC), and experiments were performed in compliance with institutional guidelines. C57BL/6-ApcMin/+/J mice were used for the spatial transcriptomics experiments. All mice (C57BL/6-ApcMin/+/J and C57BL/6-Wild type) were maintained at the barrier mouse facility at Weill Hall of |
|---|---|

Cornell University. ApcMin/+ and wild-type mice were initially ordered from Jackson Laboratory and then bred in the barrier facility. The ApcMin/+ mice used in these experiments have a chemically induced transversion point mutation at nucleotide 2549, resulting in a stop codon at codon 850, truncating the APC protein.

Wild animals | No wild animals were used in this study.

Reporting on sex | Both male and female mice were used, and their precise age was noted.

Field-collected samples | NA

Ethics oversight | The Cornell University Institutional Animal Care and Use Committee (IACUC) approved all animal protocols, and experiments were performed in compliance with its institutional guidelines (protocol number : IACUC 2016-0088).

Note that full information on the approval of the study protocol must also be provided in the manuscript.

# Plants

Seed stocks | NA

Novel plant genotypes | NA

Authentication | NA

