## [Peer Review File · Nature Microbiology]

Spatial Transcriptomics Maps Host-Gut Microbiome Biogeography at High Resolution

Corresponding Author: Dr Iwijn De Vlaminck

Version 0:

Reviewer comments:

Reviewer #1

(Remarks to the Author)

Ntekas, Takayasu, McKellar et al. present an ambitious and timely advance in spatial microbiome profiling, extending the group's prior STRS methodology by combining enzymatic in situ polyadenylation using Poly(A) Polymerase (PAP) with two spatial transcriptomics platforms: Visium and Stereo-seq. The approach enables enhanced capture of both host and microbial RNA, including non-polyadenylated species (which technically also include the virome as previous demonstrated by this group), at high spatial resolution. The study includes method validation in murine intestinal tissue and application to tumor-associated settings, offering detailed microbiome-host maps.

While the methodological framework is compelling and clearly addresses a key technical gap in spatial microbiome profiling, several critical aspects require clarification, benchmarking, and further contextualization to support broader adoption. This is a technically sound and innovative study that bridges spatial transcriptomics and microbiome science in a powerful way. Addressing the comments above will significantly improve the rigor, transparency, and impact of the work.

Outlined below are major and minor points that will help for a robust revision.

Major points:

1. Characterization of In Situ PAP Efficiency and Enzymatic Uniformity

The manuscript introduces in situ polyadenylation as a foundational innovation but omits an important discussion of optimization of conditions for PAP, including the enzymatic activity across tissue. Buffer diffusion (e.g., Mg^{2+}) is likely to be spatially heterogeneous, especially in thicker tissue sections or across mucus layers, and within the same tissue across various architecture.

The authors should discuss and ideally quantify the spatial uniformity of PAP activity across the tissue section. Include potential implications of non-uniform polyadenylation on spatial transcript detection and microbial capture bias.

2. Optimization Parameters: Time and Concentration of PAP Treatment

Details of how PAP conditions (e.g., enzyme concentration, ATP levels, incubation time) were selected are buried in the methods without clear rationalization for the readers who may wish to adopt this approach.

Given that over- or under-treatment may impact RNA integrity or enzymatic efficiency, the authors should provide a brief optimization study (even $n=1$ technical) or rationale for the selected parameters. Discussions on how further optimization could enhance capture of host non-coding RNAs or low-abundance microbial transcripts will be helpful.

3. Sequencing Depth and Data Saturation Metrics (Fig. 2C)

Fig. 2C demonstrates increased microbial capture, but lacks statistical analysis and implications for sequencing depth. The authors should

1. Perform and display a correlation test (e.g., Pearson's R) comparing microbial vs host UMI counts per spot in Fig. 2C.
2. Show the same scales for the X and Y axes
3. Add a sequencing saturation plot or species accumulation curve with and without PAP to contextualize the increase in sensitivity.
4. Discuss what sequencing depth is required to reach saturation for the host side and the microbiome side, especially given lower correlations in host transcript recovery.

4. Impact of Polyadenylation on Downstream Deconvolution

The manuscript uses reference-based deconvolution methods from scRNA-seq data not directly from the paired tissues here, but

from prior publications on the same mouse model. However, reference datasets typically represent polyA-enriched libraries, while the spatial data here include polyadenylated total RNA. This mismatch could skew predictions.

1. The authors should discuss or quantify how the addition of non-coding RNA features affect deconvolution accuracy
2. [Optional]: Consider presenting a comparison or benchmarking analysis with and without non-polyA species for cell type prediction.

5. Biological Interpretation of Low Host UMI Density

In the Stereo-seq dataset, the reported average of 3.77 host UMIs/ μm^2 seems rather low compared to typical single-cell or subcellular spatial datasets. Can the authors help

1. Provide a benchmark or comparative value from prior Stereo-seq or Visium studies without PAP.
2. Discuss whether this capture rate supports reliable downstream analysis (e.g., differential expression, clustering).
3. Discuss or assess if non-host RNA competition for capture sites contribute to lower host UMI recovery

6. Bias in Microbial Detection Efficiency Across Species

The authors present aggregate improvements in microbial RNA capture, but do not evaluate PAP efficiency across different microbial taxa. Will the authors be able to:

1. Test or discuss whether certain bacterial phyla or cell wall types (e.g., Gram-negative vs Gram-positive) are more amenable to PAP
2. Ideally, provide 16S rRNA profiles from adjacent tissue sections or bulk metagenomics as a baseline for microbial composition.

Minor Comments

1. Clarify Background Signal in Control (Fig. 2H)

The detection of microbial signal in control (non-PAP) samples suggests some level of background or non-specific capture. Can the authors discuss the likely sources of this signal (e.g., A-rich bacterial transcripts, mispriming), and estimate the percentage of background signal attributable to such artifacts.

2. Species Near Host-Microbiome Boundary

Given the spatial resolution and descriptive nature of genus-level enrichment at the host boundary, it would strengthen the interpretation to:

1. Validate taxa localization using FISH or adjacent mucus staining (e.g., Alcian Blue/PAS).
2. Highlight taxa known to inhabit the mucus layer (e.g., Akkermansia, Bacteroides fragilis) and compare observed results to expectations.

3. While the authors use Moran's I and Ripley's H to analyze spatial clustering, these are first-order analyses. The authors should consider adding higher-order spatial statistics such as KNN-based clustering, bacterial-bacterial co-occurrence networks, or host-microbe proximity scores to showcase the richness of the data obtained here, and demonstrate future directions of advance analysis of host-microbiome data.

4. The biological inference from the tumor dataset (e.g., Fig. 4B–D) seems overextended given the apparent $n=1$. Can the authors help clarify the sample size and biological replicates in the figure legends, main text and material and methods. The authors should additionally temper down the language around biological conclusions, given the limited power (both number of animals and spatial levels), and the relatively descriptive nature of the data.

5. To support potential adoption of the method, it will be of high interest to the readers for a cost estimation. Can the authors perform a tabulation of the reagent and enzyme costs per sample, time and labor requirements for in situ polyadenylation and library preparation steps. Such a estimation will be very helpful for broad adoption of this approach!

(Remarks on code availability)

I took a look briefly at the github, and commend the authors for making the code publically available. I would suggest a better cleaning up of the github for readability, including a cleaner figure-specific markdown for each figure/ST method type to allow readers to reproduce the data.

Reviewer #2

(Remarks to the Author)

In this manuscript, Ntekas, Takayasu and colleagues extend a previous published spatial RNA capture method introduced by the De Vlamincq laboratory (Nature Biotechnology 41, 2023) to a sample type that they had not demonstrated previously: the mammalian gastrointestinal tract. In this approach, previously termed spatial total RNA-sequencing (STRS), the authors treat slices of the mouse GI tract from an APC tumor model with poly-A-polymerase. On the host side, this leads to polyadenylation of a variety of non-mRNA species. On the microbiome side, this leads to the polyadenylation of both ribosomal RNAs and mRNAs. Critically, by marking all RNAs with this sequence feature, the authors are then able to use commercially available spatial transcriptomics platforms that leverage polyA capture (Visium and StereoSeq) to spatially barcode these RNAs, such that sequencing can reveal RNA identity and location.

The authors first demonstrate the ability of this method to characterize the abundance and distribution simultaneously with host side RNA expression at multiple locations along the length of the mouse gut using the visium platform. They highlight an interesting set of observations produced by these measurements, including established variations in the abundance and density

of the microbiota along the length of the GI tract, local patches of increased microbial richness within the lumen, and differential enrichment of different microbial taxa along the transverse axis.

They then extend their measurements to the stereo-seq platform which, unlike the ~55-micron pitch between capture features, has a ~0.5-micron pitch between 0.2-micron capture spots. This produces a noticeable increase in the resolution with which host and microbial RNAs can be assigned to locations in space. As validation of this method, the authors demonstrate a general agreement between the stereo-seq measurements of host and microbe and matched slices in measured with Visium. They then highlight a variety of interesting observations made on this singular slice. For example, on the microbial side, these differential distributions of different bacteria can now be resolved with respect to host features such as villi, revealing, as expected, differential abundance of different bacteria in proximity to the host. They also reveal an interesting spatial patchiness to the distribution of bacteria, which may represent local and differential outgrowth of specific bacteria. On the host side, they segment their measurements into cells and then transfer labels to these cells based on published scRNA-seq. They generally show cell types and marker genes thereof are generally found where one would expect in the gut.

Finally, in a single slice that contains a tumor (from a different mouse), they show that there are variations in both the organization of the host tissue, which is unsurprising, and in the distribution of specific bacteria with respect to the tumor-lumen boundary.

Overall, the authors have identified a particularly exciting application space, in my opinion, for their STRS method. Especially when coupled with the stereo-seq platform, I anticipate that this method will be of immediate interest not just to researchers that study the gut microbiome but microbial communities, both commensal and environmental, in general.

However, despite this enthusiasm, I have several major concerns that lower my enthusiasm for the work in its current form. I provide these below; however, if the authors can address these concerns, I would be supportive of publication of this method.

- 1) My first major concern revolves around reproducibility, benchmarking, and quantification of key performance metrics. Given that the central deliverable of this paper is a new method, I feel the authors have not provided sufficient description of these key quantities.
 - a. The comparison of host-side and microbial-side gene expression between Visium and stereo-seq on matching slices (Figure S6) and of the non-tumor and tumor-containing slice (Figure S14) are the only clear measure of technical reproducibility. It is reassuring that the authors see correlation between these different modalities and two slices. However, so much of the manuscript is dedicated to describing elements of the spatial patterns seen with stereo-seq, particularly in microbial distribution. Yet there is no cross validation of the spatial patterns between these Visium and stereo-seq. Could the authors demonstrate that they see similar patterns (at least coarsely along the transverse axis) between the two measurement types?
 - b. Similarly, there are only two stereo-seq slice measurements presented in the paper (that of Figures 3 and 4) and they represent a non-tumor and tumor environment. Thus, it is not at all clear if the patterns that are discussed in these figures are reproducible. Could the authors repeat these measurements in a second mouse and show that they see the same distribution of bacteria and the same statistical patterns in bacterial distributions that they highlight in these slices? If the biological variability is such that these patterns do not reproduce, then it would be good to see that the patterns observed in adjacent slices are reproduced.
 - c. In parallel, while it is great to see reproducibility between stereo-seq and visium. Both methods use the same protocols to prepare samples and generate polyA tails. Thus, it is not clear if there are artifacts shared by both methods. For example, given the diverse cell wall architecture of the microbiota, one might imagine that there is a differential ability of polyA polymerase to tag RNAs from different bacteria or for these tagged RNAs to be released from the sample such that they are equally efficiently captured on the stereo-seq or Visium surface. For these reasons, I feel that it is important that the authors benchmark their measurements against independent techniques. For example, two seemingly straightforward approaches would be 1) to compare host-side RNA expression in stereo-seq to bulk RNA sequencing of the colon and 2) to compare bacterial abundance in the colon to metagenomic or 16S bulk sequencing of either fecal pellets or colonic contents. Without such comparisons, it is not clear, particularly on the microbial side, if the patterns captured accurately reflect what is actually present in these samples.
 - d. The authors provide no discussion of false-positive or false counts in their measurements. Yet, there may be some evidence that these exist. For example, Fig. 3g appears to show *Clostridium* in the lumen as expected but also, surprisingly, inside the host mucosa, submucosa, and muscle layers and lining the outside of the muscle, likely in the serosa. Fig. 3i shows similar features for *Lactobacillus*. Do the authors feel that these intra-host bacteria are real (which would be a provocative conclusion) or do they represent false positives, or perhaps contamination during processing? Are these distributions reproducible between animals?
 - e. The authors make a provocative claim that they can also profile bacterial mRNAs, yet they provide little evidence supporting this claim. First, the methods are not sufficiently detailed to understand how they have handled the ~50% of non-ribosomal RNA reads from microbes using metagenomic sequencing on sister slices. For example, how was the metagenomic sequencing done? These methods should be clarified. Second, of the non-ribosomal reads, they only discuss a single example, EF-Tu, which represents 4% of these reads. What happened to the other 96%? Finally, they provide no evidence that these reads were properly mapped to EF-Tu. Could the authors provide additional evidence that they have properly measured bacterial mRNAs?
 - f. A critical performance metric is the fraction of mRNA that is captured. There are multiple lines of evidence in the paper that suggest that it might be low for both host and microbe. For example, the only two host genes that are highlighted (*Igha* and *Lyz1*) are known to be expressed at massive levels in the cells that make them, plasma cells and Paneth cells. Could the authors provide some estimate of the efficiency of RNA capture? I suspect it is quite modest, which is not necessarily problematic, but potential users need to have a realistic sense of this value so that they can reasonably anticipate what questions this method could be used to answer. In lieu of quantitative estimates, could the authors at least provide examples of more host genes? Examples of the measured expression of more modestly expressed but critical marker genes such as *Lgr5* (stem cells), *Pdgfra* (telocytes), *Pecam* (endothelial cells), *Cd3/Cd4/Cd8* (T cells) would be exceedingly helpful guides for readers as to what they might expect from their own measurements with this technique.
 - g. Finally, the authors repeatedly suggest that the spatial separation of capture features of the stereo-seq array is the actual spatial resolution of their measurements. Yet, they have not provided any evidence of the actual accuracy with which the original

distribution of RNAs in the sample is reconstructed. In fact, it would seem highly likely that there is at least some degree of diffusion that spreads RNAs out across multiple stereo-seq capture features. Could the authors provide some quantitative evidence to support their assertion that they reconstruct RNA distributions with 1 micron accuracy (as claimed throughout)?

2) My second major concern revolves around the accuracy of the cell type identification that has been performed on the host side
a. The authors claim to segment cells using companion images. Yet I can find no description of how this was done in the methods section. The authors should describe clearly how they have defined cell boundaries and parsed stereo-seq features into cells.

b. There are a wide variety of cell types that are missing in the cell populations the authors list. A particularly glaring omission is essentially all cell types of the stroma. The authors appear to find no fibroblasts, despite ample fibroblasts in the lamina propria and sub-mucosa. Similarly, they do not identify the smooth muscle cells found in the lamina propria, in the sub-mucosa, or that of the inner and outer muscle layers. Finally, they do not identify any endothelial cells, despite multiple such cell types in the sub-mucosa. Outside of the stroma, they also do not see any of the cells that one might expect for the enteric nervous system. Yet, given the size of their slice one would anticipate at least a few patches of the myenteric plexus visible between the inner and outer muscle layers. The authors should address this apparent issue with their ability to define well established and, in some cases, abundant cell populations.

c. Similarly, some of the cell type labels used are a bit atypical and, at the very least, not clearly defined in the manuscript. For example, what is the meaning of the epithelial population that is not mapped to one of the identified epithelial subtypes. What are glandular epithelial cells? Perhaps the authors could provide a short description of these cell populations in the methods.

d. Along these lines, the authors have a clear problem with their mesothelial population. These cells are known to line the outer muscle layer on the very outside of the gut, yet the cluster identified as mesothelial would appear to fill the entire muscle layer and extend into the lamina propria (e.g. Figure S8). I suspect the authors have misidentified smooth muscle cells and myofibroblasts as mesothelial cells. At the very least they should clarify their interpretation of this distribution.

e. Finally, the authors provide little to no validation that their cells have been segmented properly, that they have been properly identified, and that there is a high degree of certainty in their assignment. Could the authors provide examples of the images they used to drive segmentation of the stereo-seq data along with annotations of the cell types assigned within those images? Similarly, could they provide a UMAP visualization of the expression profiles they see for cells on the host side with the projected labels colored on this UMAP? This visualization would allow readers to understand the degree to which these cells can be resolved with these measurements. Finally, could the authors provide expression profiles for a variety of canonical markers for the cells that have been assigned to each of their labels? These expression profiles would serve three roles. First, they would allow users to better understand some of the cell type labels. Second, they would provide an independent validation of the quality of the segmentation by revealing if there is any bleed through in expression of canonical markers of one cell type in others. Finally, they would give readers a sense of the actual expression level observed for individual cells for these canonical markers. Again, a sense of RNA capture efficiency and sensitivity of the technique is a critical element for readers to think about how they would use this technique for their own questions.

Finally, I have a series of more minor comments that I think the authors should consider in a potential revision. I provide these below in no specific order.

1) The authors may wish to remove 'unbiased' and 'one micron resolution' from their abstract. Given that they have not benchmarked their method against any independent technique, I think they have not shown that it is unbiased. Moreover, as I mention above, I do not believe that they have shown that the spatial distributions they reconstruct accurately represent the original spatial distribution in the sample with 1 micron resolution. Similar comments in the abstract should also be removed. The work has the potential to be impressive in its own right without making claims like these that I do not feel are backed up by the experiments presented.

2) The authors cite only their previous image-based microbial mapping method, HiPR-FISH in the introduction and ignore the pioneering work of others. Gary Borisy's lab is an obvious omission. His CLASI-FISH method was the first to demonstrate the ability for multiplexed mapping of bacteria in microbial communities, and it feels ungenerous to not recognize this work, especially since their method also uses colorimetric barcodes extend via spectral demixing, exactly as CLASI-FISH introduced. This also omits the recent work of Lei Dai. His SEER-FISH method, while not demonstrated in the mammalian gut, clearly provides the ability to map the cellular distribution of microbial communities via imaging.

3) Similarly, the authors criticize the work of other laboratories that have shown that they can use spatial capture platforms, such as Visium, to profile the spatial distribution of bacteria. Specifically, they state that the capture methods used by these approaches will lead to 'measurement biases and a limited scope of discovery'. Yet, these authors have not shown these limitations with previous methods, nor am I familiar with any published data to support this assertion. Perhaps it would be more accurate (and generous) to the existing methods to highlight what might be benefits of the techniques introduced here.

4) I feel the authors could do a better job of acknowledging the similarities and differences to their previous STRS method in the introduction. If there are new technical features added to this method to make it compatible with microbes (and not just viruses, as shown before) and with stereo-seq (as opposed to visium, as shown before), it would be very useful for clear statements of what these features are. If not, I don't think it diminishes the work. However, the average reader would benefit from a clear statement that here they demonstrate that STRS works in the mammalian gut and in microbial communities.

5) Throughout the manuscript the authors show spatial maps with variable color scales, e.g. Fig 2h, 3e, 3i, 3j, 3k. These color scales are not clearly defined. For these examples, one might guess that these are UMIs for the listed genes/features. However, similar scales are seen when the authors discuss the distribution of cell types, e.g., Fig. S8, in which case it is not at all clear what these color scales mean.

(Remarks on code availability)

(Remarks to the Author)

The manuscript by Ntekas et al. presents the application of their already-published enzymatic in situ polyadenylation approach (STRS; McKellar, D.W., Mantri, M., Hinchman, M.M. et al. Spatial mapping of the total transcriptome by in situ polyadenylation. *Nat Biotechnol* 41, 513–520 (2023). <https://doi.org/10.1038/s41587-022-01517-6>) to the capture of both host and microbial transcripts to perform spatial analysis of the different RNA molecules (host, microbial, ncRNA, lncRNA, microRNA, etc.) in mouse gut. The authors apply their protocol to both the Visium 55-um platform and the Stereo-seq one. By having both host and microbial spatial information, the authors show the distribution of the host spatial gene expression information and of the microbial information in four different portions of the gut as well as in the lumen. By detecting both host and microbial information, the authors investigate what type of microbes are more present closer to the host tissue and vice versa as well as what microbes are more recurrent. Finally, the authors analyze a portion of ileum tissue with tumors.

The manuscript is well written and presents some potential interesting results related to the spatial architecture and interaction of host and microbial information in the mouse gut.

Major concerns:

1. The authors state that they analyzed multiple mice (line 327). How many? Also, the authors say that they used both male and female mice (line 333) but their numbers are not stated. Please report this information since information on biological replicates and gender effect on the presented biological findings is crucial.
2. Fig. 2c-d: what are the correlation values? It looks like Visium can capture a higher number of host RNA molecules than Visium+PAP as shown in Fig. 2d and also Supp. Fig. 2a. Can the authors explain why?
3. In Supp. Fig. 2a, it almost seems like the microbial reads are competing with host RNA on the surface probes. More analyses are needed to address this aspect. The same trend is observed in STOmics in Supp. Fig. 5a (protein coding). Are the authors sure that Visium+PAP and STOmics(stereo-seq)+PAP can capture comparable host information to Visium and STOmics without PAP? Can the authors use synthetic microbial communities of which they know their microbial proportions and spiked-in RNA to calculate the respective capture proportions?
4. Related to points 2 and 3, how is the enzymatic polyadenylation of naturally polyadenylated transcripts (i.e. double polyadenylation) affect the spatial their capture on the slides? Can the authors quantify how many transcripts are double polyadenylated and if there is a biased in such a double polyadenylation?
5. The authors used a mouse heart to show lack of contamination in the microbial signal. Lines 75-77: were the mouse heart tissue sections analyzed on the same Visium slide to check for contamination? That would be an adequate experimental design even though using a germ-free mouse colon is a more appropriate control.
6. Can the authors expand on why the enrichment of RNA from viruses and archea was greatest in the small intestine (Lines 81-82)?
7. Fig. 2f: profiles between CE and CO are fairly similar but CE captures so much more rRNA. How come?
8. Fig. 3b: non-coding panel has more non-coding RNA molecules on the left side of the tissue section but that pattern is not preserved in the other three panels where the bottom part is more different compared to the rest. How come?
9. Lines 362 and 375: how was the permeabilization conducted?
10. Line 72: 156 M reads/samples. How many reads per spot? It's a more meaningful statistic to understand the sensitivity of the approach.

Minor comments:

- Line 93: missing "0" at ".0683%"
- Fig. 1b: the permeabilization seems done together or right after fixation before histology. In the text is different.

(Remarks on code availability)

Decision Letter:

29th April 2025

Dear Iwijn,

Thank you for your patience while your manuscript "High Resolution Spatial Mapping of Microbiome-Host Interactions via in situ Polyadenylation and Spatial RNA Sequencing" was under peer-review at Nature Microbiology. My apologies for the delay as one of the three referees asked for some additional time to review and we had to oblige.

The manuscript has now been seen by 3 referees, whose expertise and comments you will find at the end of this email. They find your work of potential interest, but they have raised a number of concerns that will need to be addressed before we can consider publication of the work in Nature Microbiology.

In particular, you will see that the referees have asked for more clarification on the methods and the experimental validation. Thus, we'd highly recommend that you perform additional benchmarking, validation, and quantitation as suggested by R#2. We would also suggest that you modify the description of methods to include all information regarding the in vivo systems used,

optimisations made, how the cell types were identified, and perform high-order analysis, as suggested by the referees. However, for us to consider a revised version of this manuscript, all the concerns, not just the suggested ones, will need to be addressed. We'd also require a point-by-point response to all the comments made by the referees. Should further experimental data allow you to address these criticisms, we would be happy to look at a revised manuscript.

Please include a data availability statement as a separate section after Methods but before references, under the heading "Data Availability". This section should inform readers about the availability of the data used to support the conclusions of your study. This information includes accession codes to public repositories (data banks for protein, DNA or RNA sequences, microarray, proteomics data etc...), references to source data published alongside the paper, unique identifiers such as URLs to data repository entries, or data set DOIs, and any other statement about data availability. At a minimum, you should include the following statement: "The data that support the findings of this study are available from the corresponding author upon request", mentioning any restrictions on availability. If DOIs are provided, we also strongly encourage including these in the Reference list (authors, title, publisher (repository name), identifier, year). For more guidance on how to write this section please see: <http://www.nature.com/authors/policies/data/data-availability-statements-data-citations.pdf>

* If you have not done so already we suggest that you begin to revise your manuscript so that it conforms to our Article format instructions at <http://www.nature.com/nmicrobiol/info/final-submission>. Refer also to any guidelines provided in this letter.

When submitting the revised version of your manuscript, please pay close attention to our [href="https://www.nature.com/nature-portfolio/editorial-policies/image-integrity">Digital Image Integrity Guidelines](https://www.nature.com/nature-portfolio/editorial-policies/image-integrity) and to the following points below:

EXTENDED DATA FIGURES

Link Redacted

Note: This url links to your confidential homepage and associated information about manuscripts you may have submitted or be reviewing for us. If you wish to forward this e-mail to co-authors, please delete this link to your homepage first.

Nature Microbiology is committed to improving transparency in authorship. As part of our efforts in this direction, we are now requesting that all authors identified as 'corresponding author' on published papers create and link their Open Researcher and Contributor Identifier (ORCID) with their account on the Manuscript Tracking System (MTS), prior to acceptance. This applies to primary research papers only. ORCID helps the scientific community achieve unambiguous attribution of all scholarly contributions. You can create and link your ORCID from the home page of the MTS by clicking on 'Modify my Springer Nature account'. For more information please visit please visit [a href="http://www.springernature.com/orcid">www.springernature.com/orcid](http://www.springernature.com/orcid).

If you wish to submit a suitably revised manuscript we would hope to receive it within 6 months. If you cannot send it within this time, please let us know. We will be happy to consider your revision, even if a similar study has been accepted for publication at Nature Microbiology or published elsewhere (up to a maximum of 6 months).

All the best for the revision!

Reviewer Expertise:

Referee #1: imaging, computational, host-disease interactions

Referee #2: spatial transcriptomics, microbiome

Referee #3: spatial transcriptomics

Reviewer Comments:

Reviewer #1 (Remarks to the Author):

Ntekas, Takayasu, McKellar et al. present an ambitious and timely advance in spatial microbiome profiling, extending the group's prior STRS methodology by combining enzymatic in situ polyadenylation using Poly(A) Polymerase (PAP) with two spatial transcriptomics platforms: Visium and Stereo-seq. The approach enables enhanced capture of both host and microbial RNA, including non-polyadenylated species (which technically also include the virome as previous demonstrated by this group), at high spatial resolution. The study includes method validation in murine intestinal tissue and application to tumor-associated settings, offering detailed microbiome-host maps.

While the methodological framework is compelling and clearly addresses a key technical gap in spatial microbiome profiling, several critical aspects require clarification, benchmarking, and further contextualization to support broader adoption. This is a technically sound and innovative study that bridges spatial transcriptomics and microbiome science in a powerful way. Addressing the comments above will significantly improve the rigor, transparency, and impact of the work.

Outlined below are major and minor points that will help for a robust revision.

Major points:

1. Characterization of In Situ PAP Efficiency and Enzymatic Uniformity

The manuscript introduces in situ polyadenylation as a foundational innovation but omits an important discussion of optimization of conditions for PAP, including the enzymatic activity across tissue. Buffer diffusion (e.g., Mg^{2+}) is likely to be spatially heterogeneous, especially in thicker tissue sections or across mucus layers, and within the same tissue across various architecture.

The authors should discuss and ideally quantify the spatial uniformity of PAP activity across the tissue section. Include potential implications of non-uniform polyadenylation on spatial transcript detection and microbial capture bias.

2. Optimization Parameters: Time and Concentration of PAP Treatment

Details of how PAP conditions (e.g., enzyme concentration, ATP levels, incubation time) were selected are buried in the methods without clear rationalization for the readers who may wish to adopt this approach.

Given that over- or under-treatment may impact RNA integrity or enzymatic efficiency, the authors should provide a brief optimization study (even $n=1$ technical) or rationale for the selected parameters. Discussions on how further optimization could enhance capture of host non-coding RNAs or low-abundance microbial transcripts will be helpful.

3. Sequencing Depth and Data Saturation Metrics (Fig. 2C)

Fig. 2C demonstrates increased microbial capture, but lacks statistical analysis and implications for sequencing depth. The authors should

1. Perform and display a correlation test (e.g., Pearson's R) comparing microbial vs host UMI counts per spot in Fig. 2C.
2. Show the same scales for the X and Y axes
3. Add a sequencing saturation plot or species accumulation curve with and without PAP to contextualize the increase in sensitivity.
4. Discuss what sequencing depth is required to reach saturation for the host side and the microbiome side, especially given lower correlations in host transcript recovery.

4. Impact of Polyadenylation on Downstream Deconvolution

The manuscript uses reference-based deconvolution methods from scRNA-seq data not directly from the paired tissues here, but from prior publications on the same mouse model. However, reference datasets typically represent polyA-enriched libraries, while the spatial data here include polyadenylated total RNA. This mismatch could skew predictions.

1. The authors should discuss or quantify how the addition of non-coding RNA features affect deconvolution accuracy
2. [Optional]: Consider presenting a comparison or benchmarking analysis with and without non-polyA species for cell type prediction.

5. Biological Interpretation of Low Host UMI Density

In the Stereo-seq dataset, the reported average of 3.77 host UMIs/ μm^2 seems rather low compared to typical single-cell or subcellular spatial datasets. Can the authors help

1. Provide a benchmark or comparative value from prior Stereo-seq or Visium studies without PAP.
2. Discuss whether this capture rate supports reliable downstream analysis (e.g., differential expression, clustering).
3. Discuss or assess if non-host RNA competition for capture sites contribute to lower host UMI recovery

6. Bias in Microbial Detection Efficiency Across Species

The authors present aggregate improvements in microbial RNA capture, but do not evaluate PAP efficiency across different microbial taxa. Will be authors be able to:

1. Test or discuss whether certain bacterial phyla or cell wall types (e.g., Gram-negative vs Gram-positive) are more amenable to PAP
2. Ideally, provide 16S rRNA profiles from adjacent tissue sections or bulk metagenomics as a baseline for microbial composition.

Minor Comments

1. Clarify Background Signal in Control (Fig. 2H)

The detection of microbial signal in control (non-PAP) samples suggests some level of background or non-specific capture. Can the authors discuss the likely sources of this signal (e.g., A-rich bacterial transcripts, mispriming), and estimate the percentage of background signal attributable to such artifacts.

2. Species Near Host-Microbiome Boundary

Given the spatial resolution and descriptive nature of genus-level enrichment at the host boundary, it would strengthen the interpretation to:

1. Validate taxa localization using FISH or adjacent mucus staining (e.g., Alcian Blue/PAS).
2. Highlight taxa known to inhabit the mucus layer (e.g., Akkermansia, Bacteroides fragilis) and compare observed results to expectations.

3. While the authors use Moran's I and Ripley's H to analyze spatial clustering, these are first-order analyses. The authors should consider adding higher-order spatial statistics such as KNN-based clustering, bacterial-bacterial co-occurrence networks, or host-microbe proximity scores to showcase the richness of the data obtained here, and demonstrate future directions of advance analysis of host-microbiome data.

4. The biological inference from the tumor dataset (e.g., Fig. 4B–D) seems overextended given the apparent $n=1$. Can the authors help clarify the sample size and biological replicates in the figure legends, main text and material and methods. The authors should additionally temper down the language around biological conclusions, given the limited power (both number of animals and spatial levels), and the relatively descriptive nature of the data.

5. To support potential adoption of the method, it will be of high interest to the readers for a cost estimation. Can the authors perform a tabulation of the reagent and enzyme costs per sample, time and labor requirements for in situ polyadenylation and library preparation steps. Such a estimation will be very helpful for broad adoption of this approach!

Reviewer #1 (Remarks on code availability):

I took a look briefly at the github, and commend the authors for making the code publically available. I would suggest a better cleaning up of the github for readability, including a cleaner figure-specific markdown for each figure/ST method type to allow readers to reproduce the data.

Reviewer #2 (Remarks to the Author):

In this manuscript, Ntekas, Takayasu and colleagues extend a previous published spatial RNA capture method introduced by the De Vlaminck laboratory (Nature Biotechnology 41, 2023) to a sample type that they had not demonstrated previously: the mammalian gastrointestinal tract. In this approach, previously termed spatial total RNA-sequencing (STRS), the authors treat slices of the mouse GI tract from an APC tumor model with poly-A-polymerase. On the host side, this leads to polyadenylation of a variety of non-mRNA species. On the microbiome side, this leads to the polyadenylation of both ribosomal RNAs and mRNAs. Critically, by marking all RNAs with this sequence feature, the authors are then able to use commercially available spatial transcriptomics platforms that leverage polyA capture (Visium and StereoSeq) to spatially barcode these RNAs, such that sequencing can reveal RNA identity and location.

The authors first demonstrate the ability of this method to characterize the abundance and distribution simultaneously with host side RNA expression at multiple locations along the length of the mouse gut using the visium platform. They highlight an interesting set of observations produced by these measurements, including established variations in the abundance and density of the microbiota along the length of the GI tract, local patches of increased microbial richness within the lumen, and differential enrichment of different microbial taxa along the transverse axis.

They then extend their measurements to the stereo-seq platform which, unlike the ~55-micron pitch between capture features, has a ~0.5-micron pitch between 0.2-micron capture spots. This produces a noticeable increase in the resolution with which host

and microbial RNAs can be assigned to locations in space. As validation of this method, the authors demonstrate a general agreement between the stereo-seq measurements of host and microbe and matched slices in measured with Visium. They then highlight a variety of interesting observations made on this singular slice. For example, on the microbial side, these differential distributions of different bacteria can now be resolved with respect to host features such as villi, revealing, as expected, differential abundance of different bacteria in proximity to the host. They also reveal an interesting spatial patchiness to the distribution of bacteria, which may represent local and differential outgrowth of specific bacteria. On the host side, they segment their measurements into cells and then transfer labels to these cells based on published scRNA-seq. They generally show cell types and marker genes thereof are generally found where one would expect in the gut.

Finally, in a single slice that contains a tumor (from a different mouse), they show that there are variations in both the organization of the host tissue, which is unsurprising, and in the distribution of specific bacteria with respect to the tumor-lumen boundary.

Overall, the authors have identified a particularly exciting application space, in my opinion, for their STRS method. Especially when coupled with the stereo-seq platform, I anticipate that this method will be of immediate interest not just to researchers that study the gut microbiome but microbial communities, both commensal and environmental, in general.

However, despite this enthusiasm, I have several major concerns that lower my enthusiasm for the work in its current form. I provide these below; however, if the authors can address these concerns, I would be supportive of publication of this method.

- 1) My first major concern revolves around reproducibility, benchmarking, and quantification of key performance metrics. Given that the central deliverable of this paper is a new method, I feel the authors have not provided sufficient description of these key quantities.
 - a. The comparison of host-side and microbial-side gene expression between Visium and stereo-seq on matching slices (Figure S6) and of the non-tumor and tumor-containing slice (Figure S14) are the only clear measure of technical reproducibility. It is reassuring that the authors see correlation between these different modalities and two slices. However, so much of the manuscript is dedicated to describing elements of the spatial patterns seen with stereo-seq, particularly in microbial distribution. Yet there is no cross validation of the spatial patterns between these Visium and stereo-seq. Could the authors demonstrate that they see similar patterns (at least coarsely along the transverse axis) between the two measurement types?
 - b. Similarly, there are only two stereo-seq slice measurements presented in the paper (that of Figures 3 and 4) and they represent a non-tumor and tumor environment. Thus, it is not at all clear if the patterns that are discussed in these figures are reproducible. Could the authors repeat these measurements in a second mouse and show that they see the same distribution of bacteria and the same statistical patterns in bacterial distributions that they highlight in these slices? If the biological variability is such that these patterns do not reproduce, then it would be good to see that the patterns observed in adjacent slices are reproduced.
 - c. In parallel, while it is great to see reproducibility between stereo-seq and visium. Both methods use the same protocols to prepare samples and generate polyA tails. Thus, it is not clear if there are artifacts shared by both methods. For example, given the diverse cell wall architecture of the microbiota, one might imagine that there is a differential ability of polyA polymerase to tag RNAs from different bacteria or for these tagged RNAs to be released from the sample such that they are equally efficiently captured on the stereo-seq or Visium surface. For these reasons, I feel that it is important that the authors benchmark their measurements against independent techniques. For example, two seemingly straightforward approaches would be 1) to compare host-side RNA expression in stereo-seq to bulk RNA sequencing of the colon and 2) to compare bacterial abundance in the colon to metagenomic or 16S bulk sequencing of either fecal pellets or colonic contents. Without such comparisons, it is not clear, particularly on the microbial side, if the patterns captured accurately reflect what is actually present in these samples.
 - d. The authors provide no discussion of false-positive or false counts in their measurements. Yet, there may be some evidence that these exist. For example, Fig. 3g appears to show *Clostridium* in the lumen as expected but also, surprisingly, inside the host mucosa, submucosa, and muscle layers and lining the outside of the muscle, likely in the serosa. Fig. 3i shows similar features for *Lactobacillus*. Do the authors feel that these intra-host bacteria are real (which would be a provocative conclusion) or do they represent false positives, or perhaps contamination during processing? Are these distributions reproducible between animals?
 - e. The authors make a provocative claim that they can also profile bacterial mRNAs, yet they provide little evidence supporting this claim. First, the methods are not sufficiently detailed to understand how they have handled the ~50% of non-ribosomal RNA reads from microbes using metagenomic sequencing on sister slices. For example, how was the metagenomic sequencing done? These methods should be clarified. Second, of the non-ribosomal reads, they only discuss a single example, EF-Tu, which represents 4% of these reads. What happened to the other 96%? Finally, they provide no evidence that these reads were properly mapped to EF-Tu. Could the authors provide additional evidence that they have properly measured bacterial mRNAs?
 - f. A critical performance metric is the fraction of mRNA that is captured. There are multiple lines of evidence in the paper that suggest that it might be low for both host and microbe. For example, the only two host genes that are highlighted (*Igha* and *Lyz1*) are known to be expressed at massive levels in the cells that make them, plasma cells and Paneth cells. Could the authors provide some estimate of the efficiency of RNA capture? I suspect it is quite modest, which is not necessarily problematic, but potential users need to have a realistic sense of this value so that they can reasonably anticipate what questions this method could be used to answer. In lieu of quantitative estimates, could the authors at least provide examples of more host genes? Examples of the measured expression of more modestly expressed but critical marker genes such as *Lgr5* (stem cells), *Pdgfra* (telocytes), *Pecam* (endothelial cells), *Cd3/Cd4/Cd8* (T cells) would be exceedingly helpful guides for readers as to what they might expect from their own measurements with this technique.
 - g. Finally, the authors repeatedly suggest that the spatial separation of capture features of the stereo-seq array is the actual spatial resolution of their measurements. Yet, they have not provided any evidence of the actual accuracy with which the original distribution of RNAs in the sample is reconstructed. In fact, it would seem highly likely that there is at least some degree of diffusion that spreads RNAs out across multiple stereo-seq capture features. Could the authors provide some quantitative evidence to support their assertion that they reconstruct RNA distributions with 1 micron accuracy (as claimed throughout)?

2) My second major concern revolves around the accuracy of the cell type identification that has been performed on the host side

- The authors claim to segment cells using companion images. Yet I can find no description of how this was done in the methods section. The authors should describe clearly how they have defined cell boundaries and parsed stereo-seq features into cells.

- There are a wide variety of cell types that are missing in the cell populations the authors list. A particularly glaring omission is essentially all cell types of the stroma. The authors appear to find no fibroblasts, despite ample fibroblasts in the lamina propria and sub-mucosa. Similarly, they do not identify the smooth muscle cells found in the lamina propria, in the sub-mucosa, or that of the inner and outer muscle layers. Finally, they do not identify any endothelial cells, despite multiple such cell types in the sub-mucosa. Outside of the stroma, they also do not see any of the cells that one might expect for the enteric nervous system. Yet, given the size of their slice one would anticipate at least a few patches of the myenteric plexus visible between the inner and outer muscle layers. The authors should address this apparent issue with their ability to define well established and, in some cases, abundant cell populations.

- Similarly, some of the cell type labels used are a bit atypical and, at the very least, not clearly defined in the manuscript. For example, what is the meaning of the epithelial population that is not mapped to one of the identified epithelial subtypes. What are glandular epithelial cells? Perhaps the authors could provide a short description of these cell populations in the methods.

- Along these lines, the authors have a clear problem with their mesothelial population. These cells are known to line the outer muscle layer on the very outside of the gut, yet the cluster identified as mesothelial would appear to fill the entire muscle layer and extend into the lamina propria (e.g. Figure S8). I suspect the authors have misidentified smooth muscle cells and myofibroblasts as mesothelial cells. At the very least they should clarify their interpretation of this distribution.

- Finally, the authors provide little to no validation that their cells have been segmented properly, that they have been properly identified, and that there is a high degree of certainty in their assignment. Could the authors provide examples of the images they used to drive segmentation of the stereo-seq data along with annotations of the cell types assigned within those images?

Similarly, could they provide a UMAP visualization of the expression profiles they see for cells on the host side with the projected labels colored on this UMAP? This visualization would allow readers to understand the degree to which these cells can be resolved with these measurements. Finally, could the authors provide expression profiles for a variety of canonical markers for the cells that have been assigned to each of their labels? These expression profiles would serve three roles. First, they would allow users to better understand some of the cell type labels. Second, they would provide an independent validation of the quality of the segmentation by revealing if there is any bleed through in expression of canonical markers of one cell type in others. Finally, they would give readers a sense of the actual expression level observed for individual cells for these canonical markers. Again, a sense of RNA capture efficiency and sensitivity of the technique is a critical element for readers to think about how they would use this technique for their own questions.

Finally, I have a series of more minor comments that I think the authors should consider in a potential revision. I provide these below in no specific order.

- The authors may wish to remove 'unbiased' and 'one micron resolution' from their abstract. Given that they have not benchmarked their method against any independent technique, I think they have not shown that it is unbiased. Moreover, as I mention above, I do not believe that they have shown that the spatial distributions they reconstruct accurately represent the original spatial distribution in the sample with 1 micron resolution. Similar comments in the abstract should also be removed. The work has the potential to be impressive in its own right without making claims like these that I do not feel are backed up by the experiments presented.

- The authors cite only their previous image-based microbial mapping method, HiPR-FISH in the introduction and ignore the pioneering work of others. Gary Borisy's lab is an obvious omission. His CLASI-FISH method was the first to demonstrate the ability for multiplexed mapping of bacteria in microbial communities, and it feels ungenerous to not recognize this work, especially since their method also uses colorimetric barcodes extend via spectral demixing, exactly as CLASI-FISH introduced. This also omits the recent work of Lei Dai. His SEER-FISH method, while not demonstrated in the mammalian gut, clearly provides the ability to map the cellular distribution of microbial communities via imaging.

- Similarly, the authors criticize the work of other laboratories that have shown that they can use spatial capture platforms, such as Visium, to profile the spatial distribution of bacteria. Specifically, they state that the capture methods used by these approaches will lead to 'measurement biases and a limited scope of discovery'. Yet, these authors have not shown these limitations with previous methods, nor am I familiar with any published data to support this assertion. Perhaps it would be more accurate (and generous) to the existing methods to highlight what might be benefits of the techniques introduced here.

- I feel the authors could do a better job of acknowledging the similarities and differences to their previous STRS method in the introduction. If there are new technical features added to this method to make it compatible with microbes (and not just viruses, as shown before) and with stereo-seq (as opposed to visium, as shown before), it would be very useful for clear statements of what these features are. If not, I don't think it diminishes the work. However, the average reader would benefit from a clear statement that here they demonstrate that STRS works in the mammalian gut and in microbial communities.

- Throughout the manuscript the authors show spatial maps with variable color scales, e.g. Fig 2h, 3e, 3i, 3j, 3k. These color scales are not clearly defined. For these examples, one might guess that these are UMIs for the listed genes/features. However, similar scales are seen when the authors discuss the distribution of cell types, e.g., Fig. S8, in which case it is not at all clear what these color scales mean.

Reviewer #3 (Remarks to the Author):

The manuscript by Ntekas et al. presents the application of their already-published enzymatic in situ polyadenylation approach (STRS; McKellar, D.W., Mantri, M., Hinchman, M.M. et al. Spatial mapping of the total transcriptome by in situ polyadenylation. *Nat Biotechnol* 41, 513–520 (2023). <https://doi.org/10.1038/s41587-022-01517-6>) to the capture of both host and microbial transcripts to perform spatial analysis of the different RNA molecules (host, microbial, ncRNA, lncRNA, microRNA, etc.) in mouse gut. The authors apply their protocol to both the Visium 55-um platform and the Stereo-seq one. By having both host and microbial spatial information, the authors show the distribution of the host spatial gene expression information and of the

microbial information in four different portions of the gut as well as in the lumen. By detecting both host and microbial information, the authors investigate what type of microbes are more present closer to the host tissue and vice versa as well as what microbes are more recurrent. Finally, the authors analyze a portion of ileum tissue with tumors.

The manuscript is well written and presents some potential interesting results related to the spatial architecture and interaction of host and microbial information in the mouse gut.

Major concerns:

1. The authors state that they analyzed multiple mice (line 327). How many? Also, the authors say that they used both male and female mice (line 333) but their numbers are not stated. Please report this information since information on biological replicates and gender effect on the presented biological findings is crucial.
2. Fig. 2c-d: what are the correlation values? It looks like Visium can capture a higher number of host RNA molecules than Visium+PAP as shown in Fig. 2d and also Supp. Fig. 2a. Can the authors explain why?
3. In Supp. Fig. 2a, it almost seems like the microbial reads are competing with host RNA on the surface probes. More analyses are needed to address this aspect. The same trend is observed in STOmics in Supp. Fig. 5a (protein coding). Are the authors sure that Visium+PAP and STOmics(stereo-seq)+PAP can capture comparable host information to Visium and STOmics without PAP? Can the authors use synthetic microbial communities of which they know their microbial proportions and spiked-in RNA to calculate the respective capture proportions?
4. Related to points 2 and 3, how is the enzymatic polyadenylation of naturally polyadenylated transcripts (i.e. double polyadenylation) affect the spatial their capture on the slides? Can the authors quantify how many transcripts are double polyadenylated and if there is a biased in such a double polyadenylation?
5. The authors used a mouse heart to show lack of contamination in the microbial signal. Lines 75-77: were the mouse heart tissue sections analyzed on the same Visium slide to check for contamination? That would be an adequate experimental design even though using a germ-free mouse colon is a more appropriate control.
6. Can the authors expand on why the enrichment of RNA from viruses and archea was greatest in the small intestine (Lines 81-82)?
7. Fig. 2f: profiles between CE and CO are fairly similar but CE captures so much more rRNA. How come?
8. Fig. 3b: non-coding panel has more non-coding RNA molecules on the left side of the tissue section but that pattern is not preserved in the other three panels where the bottom part is more different compared to the rest. How come?
9. Lines 362 and 375: how was the permeabilization conducted?
10. Line 72: 156 M reads/samples. How many reads per spot? It's a more meaningful statistic to understand the sensitivity of the approach.

Minor comments:

- Line 93: missing "0" at ".0683%"
- Fig. 1b: the permeabilization seems done together or right after fixation before histology. In the text is different.

Version 1:

Reviewer comments:

Reviewer #1

(Remarks to the Author)

I commend the authors for substantial work in addressing some of these comments. There are still some outstanding questions below:

1. The authors should include appropriate statistical tests for the responses in Q4 and Q6, the stacked bar plot visualization does not suffice.
2. Can the authors elaborate further on the rationale for fresh frozen + methacarn, which echos R2's question (Q1d) on unexpected bacteria signal in the lumen, likely due to smearing/floating bacteria and fecal contents. Given that the authors previously used methacarn and paraffin embedding versus fresh frozen (PMID: 33268897), this will be important to discuss.

(Remarks on code availability)

Reviewer #2

(Remarks to the Author)

In this revised manuscript, Ntekas and colleagues have performed a series of new experiments and analyses to address the

concerns that I raised in my original review. Overall, I applaud the authors for a thorough response that strengthens their manuscript. In particular, I appreciate the addition of independent cross validations of several of their measurements using metatranscriptomics.

Nonetheless, I have a few issues that I feel were not fully addressed in the previous response. Addressing these issues will provide important performance insights that are essential for readers interested in adopting this method.

First, the agreement between their Visium data and metatranscriptomics of the microbiota in Figure S5a is impressive for cecum and colon, and the addition of these measurements strengthens their work. However, to my eye, there is very little agreement between these measurements in the small intestine, in particular the proximal small intestine. The correlation coefficients seem in both small intestine regions to be driven by a handful of genera. In the text the authors describe this as good agreement across all regions, yet I would say that in the small intestine that is true really only for a very modest number of the measured genera. Readers that would aim to use this technology should be fully aware that the agreement in the small intestine is limited to a very small number of the detected genera. Perhaps there is a technical or biological explanation that mitigates this lack of agreement for the vast majority of detected genera (most of which are detected at low levels in both methods). Nonetheless, I feel the authors need to acknowledge this lack of strong agreement and provide some discussion of why this might be the case.

Second, I raised concerns about false positives, as evidenced by bacteria detected in regions one would next expect to find bacteria. The authors provide a nice discussion of this point in the response. Yet, I cannot find this point acknowledged in the text. It is important for readers to understand that the authors see bacterial reads in locations that they would not be expected to be found, that these reads are not uniformly distributed across bacteria (if this point is true), and to have some explanation from the authors as to why this might be the case. Otherwise, the authors would appear to make the implicit assertion that all of these detected bacterial reads were found in the actual location of those bacteria. If that were true, the presence of such a large number of bacteria in what have previously been believed to be largely sterile regions would be a major discovery.

Third, I thank the authors for the additional data they provide in response to my comments on reproducibility. I feel these additional measurements are indeed very useful for providing some evidence that their measurements are reproducible. Yet, the authors provide limited additional analysis of these data to support any of the larger biological conclusions/patterns they highlight in their discussion. This point is particularly apparent in their discussion of the biological findings within the single small intestine slice presented in Figure 3. As they would appear to have collected a second section of the ileum (Figure S18), could they repeat these same analyses in this slice and discuss which of these patterns is reproduced in that slice? This point is particularly important in the context of my comment about false positives above, as one of the differences in spatial distribution that they describe (Figure 3g) focuses on *Clostridium*, *Klebsiella* (not shown), and *Eggerthella*, is apparent, in part, because of the broad distribution of *Clostridium*. However, by my eye much of the broad *Clostridium* distribution arises because of measurements both within host tissue and on the outside of the host tissue. Is this same pattern seen in a different slice?

This point also applies to their discussion of colony size in Figure 3. Are these patterns present within this slice as well? Again, reproducibility in these biological features is an essential criterion that readers should have to judge whether they would adopt this technique.

Fourth, I appreciate the qualitative demonstration of sensitivity that the authors now provide by showing the distribution of a variety of genes. Indeed, the sensitivity of stereo-seq would appear to be pretty limited, as modestly expressed marker genes such as *Lgr5* or *Cd3* are simply not seen with the frequency one would expect for the cell types they mark. I also appreciate that the authors now provide dot plots (Figure S12b) to provide some support that the cell type labels have been reasonably propagated to their stereo-seq 'cells'. That being said, the overall lack of definition within the UMAP of these stereo-seq features in Figure S12c as well as the broad intermixing of many of these propagated labels would support the assertion that their ability to robustly define fine cell types divisions is questionable. The authors are faced with a challenge that is common to many of the spatial capture methods that have both modest cellular resolution and capture efficiency, so my point here is not to highlight what I think is a serious issue. In particular, Figures S12 and S15 will allow the curious reader to better understand the inherent limitations of the sensitivity of stereo-seq and its ability to robustly identify cells. Nonetheless, so much of the paper makes conclusions draw on firm assignment of cell types/states. Perhaps the authors could acknowledge in the discussion that there is room for improvement.

(Remarks on code availability)

Reviewer #4

(Remarks to the Author)

The author has adequately answered all concerns.

(Remarks on code availability)

N/A

Decision Letter:

3rd December 2025

Dear Iwijn,

Thank you for your patience while your manuscript "High Resolution Spatial Mapping of Microbiome-Host Interactions via in situ Polyadenylation and Spatial RNA Sequencing" was under peer-review at Nature Microbiology. It has now been seen by 3 referees, whose comments you will find at the of this email.

We are very interested in the possibility of publishing your study in Nature Microbiology, but would like to consider your response to these concerns in the form of a revised manuscript before we make a final decision on publication.

Looking at the comments from the R3 (replacement of the original R3), it might feel like all this time was spent waiting for nothing, but we had to be sure before making a decision.

As from the comments I shared before, you will see that the referees have asked for some extra analysis to make sure that the biological findings are reproducible and robust. They have also asked for better statistical analyses and for discussion on aspects like false positives, the use of embedding medium, and cell-type assignment. We feel that these issues should be straightforward to address.

If you have not done so already please begin to revise your manuscript so that it conforms to our Article format instructions at <http://www.nature.com/nmicrobiol/info/final-submission/>

The usual length limit for a Nature Microbiology Article is six display items (figures or tables) and 3,000 words. We have some flexibility, and can allow a revised manuscript at 3,500 words, but please consider this a firm upper limit. There is a trade-off of ~250 words per display item, so if you need more space, you could move a Figure or Table to Supplementary Information.

Some reduction could be achieved by focusing any introductory material and moving it to the start of your opening 'bold' paragraph, whose function is to outline the background to your work, describe in a sentence your new observations, and explain your main conclusions. The discussion should also be limited. Methods should be described in a separate section following the discussion, we do not place a word limit on Methods.

Nature Microbiology titles should give a sense of the main new findings of a manuscript, and should not contain punctuation. Please keep in mind that we strongly discourage active verbs in titles, and that they should ideally fit within 90 characters each (including spaces).

Please include a data availability statement as a separate section after Methods but before references, under the heading "Data Availability". This section should inform readers about the availability of the data used to support the conclusions of your study. This information includes accession codes to public repositories (data banks for protein, DNA or RNA sequences, microarray, proteomics data etc...), references to source data published alongside the paper, unique identifiers such as URLs to data repository entries, or data set DOIs, and any other statement about data availability. At a minimum, you should include the following statement: "The data that support the findings of this study are available from the corresponding author upon request", mentioning any restrictions on availability. If DOIs are provided, we also strongly encourage including these in the Reference list (authors, title, publisher (repository name), identifier, year). For more guidance on how to write this section please see: <http://www.nature.com/authors/policies/data/data-availability-statements-data-citations.pdf>

To improve the accessibility of your paper to readers from other research areas, please pay particular attention to the wording of the paper's opening bold paragraph, which serves both as an introduction and as a brief, non-technical summary in about 150 words. If, however, you require one or two extra sentences to explain your work clearly, please include them even if the paragraph is over-length as a result. The opening paragraph should not contain references. Because scientists from other sub-disciplines will be interested in your results and their implications, it is important to explain essential but specialised terms concisely. We suggest you show your summary paragraph to colleagues in other fields to uncover any problematic concepts.

If your paper is accepted for publication, we will edit your display items electronically so they conform to our house style and will reproduce clearly in print. If necessary, we will re-size figures to fit single or double column width. If your figures contain several parts, the parts should form a neat rectangle when assembled. Choosing the right electronic format at this stage will speed up the processing of your paper and give the best possible results in print. We would like the figures to be supplied as vector files - EPS, PDF, AI or postscript (PS) file formats (not raster or bitmap files), preferably generated with vector-graphics software (Adobe Illustrator for example). Please try to ensure that all figures are non-flattened and fully editable. All images should be at least 300 dpi resolution (when figures are scaled to approximately the size that they are to be printed at) and in RGB colour format. Please do not submit Jpeg or flattened TIFF files. Please see also 'Guidelines for Electronic Submission of Figures' at the end of this letter for further detail.

Figure legends must provide a brief description of the figure and the symbols used, within 350 words, including definitions of any error bars employed in the figures.

EXTENDED DATA FIGURES

Please include a statement before the acknowledgements naming the author to whom correspondence and requests for materials should be addressed.

Finally, we require authors to include a statement of their individual contributions to the paper -- such as experimental work, project planning, data analysis, etc. -- immediately after the acknowledgements. The statement should be short, and refer to authors by their initials. For details please see the Authorship section of our joint Editorial policies at http://www.nature.com/authors/editorial_policies/authorship.html

* include a point-by-point response to any editorial suggestions and to our referees. Please include your response to the editorial suggestions in your cover letter, and please upload your response to the referees as a separate document.

* ensure it complies with our format requirements for Letters as set out in our guide to authors at www.nature.com/nmicrobiol/info/gta/

* state in a cover note the length of the text, methods and legends; the number of references; number and estimated final size of figures and tables

* resubmit electronically if possible using the link below to access your home page:

Link Redacted

*This url links to your confidential homepage and associated information about manuscripts you may have submitted or be reviewing for us. If you wish to forward this e-mail to co-authors, please delete this link to your homepage first.

Please ensure that all correspondence is marked with your Nature Microbiology reference number in the subject line.

Nature Microbiology is committed to improving transparency in authorship. As part of our efforts in this direction, we are now requesting that all authors identified as 'corresponding author' on published papers create and link their Open Researcher and Contributor Identifier (ORCID) with their account on the Manuscript Tracking System (MTS), prior to acceptance. This applies to primary research papers only. ORCID helps the scientific community achieve unambiguous attribution of all scholarly contributions. You can create and link your ORCID from the home page of the MTS by clicking on 'Modify my Springer Nature account'. For more information please visit www.springernature.com/orcid.

We hope to receive your revised paper within three weeks. If you cannot send it within this time, please let us know.

Yours sincerely,

Reviewers Comments:

Reviewer #1 (Remarks to the Author):

I commend the authors for substantial work in addressing some of these comments. There are still some outstanding questions

below:

1. The authors should include appropriate statistical tests for the responses in Q4 and Q6, the stacked bar plot visualization does not suffice.
2. Can the authors elaborate further on the rationale for fresh frozen + methacarn, which echos R2's question (Q1d) on unexpected bacteria signal in the lumen, likely due to smearing/floating bacteria and fecal contents. Given that the authors previously used methacarn and paraffin embedding versus fresh frozen (PMID: 33268897), this will be important to discuss.

Reviewer #2 (Remarks to the Author):

In this revised manuscript, Ntekas and colleagues have performed a series of new experiments and analyses to address the concerns that I raised in my original review. Overall, I applaud the authors for a thorough response that strengthens their manuscript. In particular, I appreciate the addition of independent cross validations of several of their measurements using metatranscriptomics.

Nonetheless, I have a few issues that I feel were not fully addressed in the previous response. Addressing these issues will provide important performance insights that are essential for readers interested in adopting this method.

First, the agreement between their Visium data and metatranscriptomics of the microbiota in Figure S5a is impressive for cecum and colon, and the addition of these measurements strengthens their work. However, to my eye, there is very little agreement between these measurements in the small intestine, in particular the proximal small intestine. The correlation coefficients seem in both small intestine regions to be driven by a handful of genera. In the text the authors describe this as good agreement across all regions, yet I would say that in the small intestine that is true really only for a very modest number of the measured genera. Readers that would aim to use this technology should be fully aware that the agreement in the small intestine is limited to a very small number of the detected genera. Perhaps there is a technical or biological explanation that mitigates this lack of agreement for the vast majority of detected genera (most of which are detected at low levels in both methods). Nonetheless, I feel the authors need to acknowledge this lack of strong agreement and provide some discussion of why this might be the case.

Second, I raised concerns about false positives, as evidenced by bacteria detected in regions one would next expect to find bacteria. The authors provide a nice discussion of this point in the response. Yet, I cannot find this point acknowledged in the text. It is important for readers to understand that the authors see bacterial reads in locations that they would not be expected to be found, that these reads are not uniformly distributed across bacteria (if this point is true), and to have some explanation from the authors as to why this might be the case. Otherwise, the authors would appear to make the implicit assertion that all of these detected bacterial reads were found in the actual location of those bacteria. If that were true, the presence of such a large number of bacteria in what have previously been believed to be largely sterile regions would be a major discovery.

Third, I thank the authors for the additional data they provide in response to my comments on reproducibility. I feel these additional measurements are indeed very useful for providing some evidence that their measurements are reproducible. Yet, the authors provide limited additional analysis of these data to support any of the larger biological conclusions/patterns they highlight in their discussion. This point is particularly apparent in their discussion of the biological findings within the single small intestine slice presented in Figure 3. As they would appear to have collected a second section of the ileum (Figure S18), could they repeat these same analyses in this slice and discuss which of these patterns is reproduced in that slice? This point is particularly important in the context of my comment about false positives above, as one of the differences in spatial distribution that they describe (Figure 3g) focuses on *Clostridium*, *Klebsiella* (not shown), and *Eggerthella*, is apparent, in part, because of the broad distribution of *Clostridium*. However, by my eye much of the broad *Clostridium* distribution arises because of measurements both within host tissue and on the outside of the host tissue. Is this same pattern seen in a different slice?

This point also applies to their discussion of colony size in Figure 3. Are these patterns present within this slice as well? Again, reproducibility in these biological features is an essential criterion that readers should have to judge whether they would adopt this technique.

Fourth, I appreciate the qualitative demonstration of sensitivity that the authors now provide by showing the distribution of a variety of genes. Indeed, the sensitivity of stereo-seq would appear to be pretty limited, as modestly expressed marker genes such as *Lgr5* or *Cd3* are simply not seen with the frequency one would expect for the cell types they mark. I also appreciate that the authors now provide dot plots (Figure S12b) to provide some support that the cell type labels have been reasonably propagated to their stereo-seq 'cells'. That being said, the overall lack of definition within the UMAP of these stereo-seq features in Figure S12c as well as the broad intermixing of many of these propagated labels would support the assertion that their ability to robustly define fine cell type divisions is questionable. The authors are faced with a challenge that is common to many of the spatial capture methods that have both modest cellular resolution and capture efficiency, so my point here is not to highlight what I think is a serious issue. In particular, Figures S12 and S15 will allow the curious reader to better understand the inherent limitations of the sensitivity of stereo-seq and its ability to robustly identify cells. Nonetheless, so much of the paper makes conclusions draw on firm assignment of cell types/states. Perhaps the authors could acknowledge in the discussion that there is room for improvement.

Reviewer #3 (Remarks to the Author):

The author has adequately answered all concerns.

Reviewer #3 (Remarks on code availability):

N/A

Version 2:

Reviewer comments:

Reviewer #1

(Remarks to the Author)

(Remarks on code availability)

Reviewer #2

(Remarks to the Author)

The authors have addressed all of my comments and concerns.

However, I did note one minor typo in a sentence they added in revision: "The lower sensitivity of sequencing platform is important to consider ..." should perhaps be "The lower sensitivity of sequencing platforms is important to consider ..."

(Remarks on code availability)

Decision Letter:

Our ref: NMICROBIOL-25020730B

9th January 2026

Dear Iwijn,

Thank you so much for your patience while your manuscript "High Resolution Spatial Mapping of Microbiome-Host Interactions via Spatial RNA Sequencing" (NMICROBIOL-25020730B) was under consideration.

It has now been seen by the original referees and their comments are below. In light of the referees' comments, we'll be happy to publish it in principle, in Nature Microbiology, pending minor revisions to satisfy the referees' final requests and to comply with our editorial and formatting guidelines.

I would highly recommend that you submit some potential cover images!

Thank you again for your interest in Nature Microbiology. Please do not hesitate to contact me if you have any questions.

Best wishes and Happy New Year,

Reviewer #1 (Remarks to the Author):

The authors' responses are acceptable and my recommendation would be to accept the manuscript!

Reviewer #2 (Remarks to the Author):

The authors have addressed all of my comments and concerns.

However, I did note one minor typo in a sentence they added in revision: "The lower sensitivity of sequencing platform is important to consider ..." should perhaps be "The lower sensitivity of sequencing platforms is important to consider ..."

Version 3:

Decision Letter:

4th February 2026

Dear Iwijn,

I am pleased to accept your Article "Spatial Transcriptomics Maps Host-Gut Microbiome Biogeography at High Resolution" for publication in Nature Microbiology. Thank you for having chosen to submit your work to us and many congratulations.

Authors may need to take specific actions to achieve compliance with funder and institutional open access mandates. If your research is supported by a funder that requires immediate open access (e.g. according to [Plan S principles](https://www.springernature.com/gp/open-science/plan-s-compliance) or the [NIH public access policy](https://www.springernature.com/gp/open-science/us-federal-agency-compliance)) then you should select the gold OA route, and we will direct you to the compliant route where possible. Because authors warrant under our subscription licensing terms that they haven't committed to licensing any version of their article under a licence inconsistent with the terms of our agreement – including the applicable embargo period – publication under the subscription model isn't suitable for authors whose funders require no embargo.

We welcome the submission of potential cover material (including a short caption of around 40 words) related to your manuscript; suggestions should be sent to Nature Microbiology as electronic files (the image should be 300 dpi at 210 x 297 mm in either TIFF or JPEG format). Please note that such pictures should be selected more for their aesthetic appeal than for their scientific content, and that colour images work better than black and white or grayscale images. Please do not try to design a cover with the Nature Microbiology logo etc., and please do not submit composites of images related to your work. I am sure you will understand that we cannot make any promise as to whether any of your suggestions might be selected for the cover of the

journal.

With kind regards,

P.S. Click on the following link if you would like to recommend Nature Microbiology to your librarian
<http://www.nature.com/subscriptions/recommend.html#forms>

** Visit the Springer Nature Editorial and Publishing website at http://editorial-jobs.springernature.com?utm_source=ejP_NMicro_email&utm_medium=ejP_NMicro_email&utm_campaign=ejp_NMicro for more information about our career opportunities. If you have any questions please click [here](mailto:editorial.publishing.jobs@springernature.com).

Response to Reviewers' comments

(Author response in blue)

Overview of the resubmission: To address the comments of the three reviewers we have conducted extensive new experiments and analyses, including: (i) orthogonal bulk metatranscriptomic measurements that corroborate the host and microbiome profiles observed with our spatial sequencing method, (ii) high-resolution profiling of five additional intestinal sites across the gastrointestinal tract of healthy mice, (iii) analysis of two additional tumor sections, including one consecutive to the section discussed in the original submission. In terms of improvements and analysis, we (iv) performed additional sequencing of the high-resolution spatial total RNA-seq libraries presented in the original manuscript (v) revisited the microbial mRNA alignment pipeline to showcase multiple transcripts, and (vi) performed rarefaction analyses demonstrating that the observed total RNA sequencing libraries are highly complex, capture a broad spectrum of molecules, and benefit from deeper sequencing. Together, these new data confirm the reproducibility of host–microbiome interface changes and further validate our observations of microbial colony formation.

Reviewer #1 (Remarks to the Author):

Ntekas, Takayasu, McKellar et al. present an ambitious and timely advance in spatial microbiome profiling, extending the group's prior STRS methodology by combining enzymatic in situ polyadenylation using Poly(A) Polymerase (PAP) with two spatial transcriptomics platforms: Visium and Stereo-seq. The approach enables enhanced capture of both host and microbial RNA, including non-polyadenylated species (which technically also include the virome as previous demonstrated by this group), at high spatial resolution. The study includes method validation in murine intestinal tissue and application to tumor-associated settings, offering detailed microbiome-host maps.

While the methodological framework is compelling and clearly addresses a key technical gap in spatial microbiome profiling, several critical aspects require clarification, benchmarking, and further contextualization to support broader adoption. This is a technically sound and innovative study that bridges spatial transcriptomics and microbiome science in a powerful way. Addressing the comments above will significantly improve the rigor, transparency, and impact of the work.

Author response: We thank reviewer #1 for the positive comments and thorough review. Your comments and suggestions have enabled us to significantly improve our analysis and presentation of our work.

Outlined below are major and minor points that will help for a robust revision.

Major points:

1. Characterization of In Situ PAP Efficiency and Enzymatic Uniformity

The manuscript introduces in situ polyadenylation as a foundational innovation but omits an important discussion of optimization of conditions for PAP, including the enzymatic activity across tissue. Buffer diffusion (e.g., Mg^{2+}) is likely to be spatially heterogeneous, especially in thicker tissue sections or across mucus layers, and within the same tissue across various architecture.

The authors should discuss and ideally quantify the spatial uniformity of PAP activity across the tissue section. Include potential implications of non-uniform polyadenylation on spatial transcript detection and microbial capture bias.

Author response: To address spatially heterogeneous polyadenylation, we computationally divided an ileum section into four approximately equal quadrants and evaluated the polyadenylation of RNA species that are not endogenously polyadenylated. Specifically, we assessed rRNA, snoRNA, and mitochondrial tRNA abundance- classes that require enzymatic A-tailing for detection, and which are constitutively expressed across cell types. We observed no significant differences in the abundances of these RNA biotypes across the four quadrants (Kruskal–Wallis $p > 0.05$ for all), supporting the notion that in situ polyadenylation is uniform at the tissue scale.

2. Optimization Parameters: Time and Concentration of PAP Treatment

Details of how PAP conditions (e.g., enzyme concentration, ATP levels, incubation time) were selected are buried in the methods without clear rationalization for the readers who may wish to adopt this approach.

Given that over- or under-treatment may impact RNA integrity or enzymatic efficiency, the authors should provide a brief optimization study (even $n=1$ technical) or rationale for the selected parameters. Discussions on how further optimization could enhance capture of host non-coding RNAs or low-abundance microbial transcripts will be helpful.

Author response: We used the concentration of Poly(A) Polymerase (PAP) and cofactors as recommended by the manufacturer, which are optimized for RNA concentrations around $0.2 \mu\text{M}$, which is also approximately the RNA content in a typical tissue section area $\approx 0.1 \mu\text{M}$ average ($\sim 10^7$ RNA molecules per cell, $6.5 \text{ mm} \times 6.5 \text{ mm} \times 0.01 \mu\text{m}$ volume). To improve coverage and robustness, we used a longer incubation time than the manufacturer's protocol and supplemented all reactions with RNase inhibitor. This ensures protection of transcripts during the reaction and maximizes labeling efficiency. To investigate this further, we performed a time course experiment to assess the fraction of transcripts which are extended as well as the length of the synthetic A-tails. To this end, we used a uniform library of RNAs, consisting of a 1.8kb in vitro transcription product encoding luciferase. We tested duplicates of five different incubation timepoints with 200 ng of RNA, including "0 minutes" where the reaction was never taken off ice, and then measured the resulting RNA lengths with the Agilent Tapestation. We found that in bulk, well-mixed conditions, the enzyme extends more than 90% of the input RNAs after just 1 minute. Furthermore, we found that the poly(A) reaction saturates within 15 minutes of incubation at 37°C . In conclusion, this experiment demonstrates that the conditions used in our work enable highly efficient polyadenylation of RNAs. The optimization experiment is now outlined in the methods section lines 384-393 and Supplementary Figure 29. In addition, we discuss potential future optimizations in the discussion section (lines 343-346).

3. Sequencing Depth and Data Saturation Metrics (Fig. 2C)

Fig. 2C demonstrates increased microbial capture, but lacks statistical analysis and implications for sequencing depth. The authors should

1. Perform and display a correlation test (e.g., Pearson's R) comparing microbial vs host UMI counts per spot in Fig. 2C.
2. Show the same scales for the X and Y axes
3. Add a sequencing saturation plot or species accumulation curve with and without PAP to contextualize the increase in sensitivity.
4. Discuss what sequencing depth is required to reach saturation for the host side and the microbiome side, especially given lower correlations in host transcript recovery.

Author response: We thank the reviewer for these suggestions. We have now incorporated new analyses to directly address these points. See also our response to reviewer 3, q2 and q3.

Following your request, we performed additional correlation analyses. We compared host mRNA with and without polyA to ensure that the host measurements are comparable and biologically interpretable. We find excellent agreement at the transcript level. In addition to these direct correlations, we have also assessed the agreement in cell type deconvolution, and found excellent agreement (see also our response to q4 below).

We performed similar correlation analysis for the microbial component, where we found lower correlation as expected given the dramatic improvement in microbial RNA recovery following PAP (**Fig. 2c** and **Fig. S3a**). To ensure the validity of the measurement, we compared the relative abundance of detected species to bulk metatranscriptomic sequencing performed on an adjacent tissue section now shown in Supplementary Figure 5 (new experiments). We found excellent agreement (see below), in particular for cecum and colon tissue which have higher microbial biomass. For ileum and proximal small intestine the agreement is lower, which we attribute to the sensitivity of the bulk assay to contamination. Metagenomic sequencing assays are sensitive to contamination, in particular in low biomass settings, a topic our lab has tackled in other contexts^{1,2}.

We also performed direct comparison to bulk RNA measurements performed on adjacent tissue sections and found strong agreement between the spatial assay and the bulk sequencing assay.

To evaluate the effect of PAP on microbial transcript detection, we performed rarefaction analysis by downsampling sequencing data from paired Visium and Visium+PAP experiments across multiple intestinal regions. After aligning to the host genome, unmapped reads were taxonomically classified using Kraken2. As shown in Supplementary Figure 2, Visium libraries recovered few microbial reads, plateauing early. In contrast, PAP-treated samples exhibited a continued increase in microbial signal with deeper sequencing, particularly in large intestine regions, consistent with a more abundant and diverse microbiota.

We performed a saturation analysis using an approach similar to that implemented in STARsolo, extending it beyond host-aligned molecules to include the entire library. The empirical saturation metric was calculated as:

$$\text{Saturation} = 1 - \frac{\text{Modality Unique molecules}}{\text{Modality reads with valid barcode}}$$

where *modality* reflects the source of the molecules: reads aligned to the mouse genome (for host), unaligned reads classified by Kraken2 (for microbiome), or all reads (for total library). This metric reflects the diminishing returns in unique molecular discovery as sequencing depth increases.

We subsampled paired Visium and Visium + PAP libraries from the 4 gut regions (Prox. SI, Ileum, Cecum, Colon) and computed saturation metrics for (i) the total library, (ii) host-aligned molecules, and (iii) the unaligned portion of the library subsequently classified as microbial using Kraken2. We fit Michaelis-Menten-like curves (constrained to a maximum of 1, in line with the definition of saturation) to extrapolate behavior beyond observed sequencing depths. As shown in Supplementary Figure 28, we find that the microbiome component saturates rapidly, typically exceeding 0.9 saturation by 100 million reads. In contrast, host transcriptome recovery, particularly in Visium + PAP libraries, requires deeper sequencing, with saturation levels approaching 0.9 only at ~200–250 million reads. This likely reflects both the added diversity of the host transcriptome and the competition for sequencing space with microbial transcripts. Overall, total library saturation plateaus around 150 million reads. These findings provide practical guidance for optimizing sequencing depth in future studies, especially when balancing host and microbial transcript recovery.

In summary, these new experiments and analyses show that:

- PAP expands the scope of mRNA biotypes that can be detected, at a small cost of a commensurate reduction in host mRNA transcripts detected.
- The assay is not sequencing saturated, and more information can be obtained by sequencing more. Indirectly, this indicates that the reaction is not limited by capture of the RNA by the local probes.
- The host measurements with and without PAP are in very good agreement
- The measurements are in good agreement with bulk analyses performed on sister sections, indicating the validity of the measurement.

4. Impact of Polyadenylation on Downstream Deconvolution

The manuscript uses reference-based deconvolution methods from scRNA-seq data not directly from the paired tissues here, but from prior publications on the same mouse model. However, reference datasets typically represent polyA-enriched libraries, while the spatial data here include polyadenylated total RNA. This mismatch could skew predictions.

1. The authors should discuss or quantify how the addition of non-coding RNA features affect deconvolution accuracy
2. [Optional]: Consider presenting a comparison or benchmarking analysis with and without non-polyA species for cell type prediction.

Author response: To address this, we compared deconvolution results using cell2location for datasets with and without polyadenylation. We found that the predicted cell type abundance and spatial distributions were highly consistent between the two assays, indicating that polyadenylation does not distort the cell type deconvolution, in support of our original conclusions (**Fig. S13**).

5. Biological Interpretation of Low Host UMI Density

In the Stereo-seq dataset, the reported average of 3.77 host UMIs/ μm^2 seems rather low compared to typical single-cell or subcellular spatial datasets. Can the authors help

1. Provide a benchmark or comparative value from prior Stereo-seq or Visium studies without PAP.
2. Discuss whether this capture rate supports reliable downstream analysis (e.g., differential expression, clustering).
3. Discuss or assess if non-host RNA competition for capture sites contribute to lower host UMI recovery

Author response: After additional sequencing, the mean combined host and microbiome UMI capture rate in our healthy ileum dataset reached 11.3 UMIs/ μm^2 . Previous studies in brain tissue have reported capture rates for Stereo-seq, 13.4 UMIs/ μm^2 in the mouse olfactory bulb (Chen et al., 2022, *Cell*) and 10.8 UMIs/ μm^2 in the postnatal mouse brain (Cheng et al., 2022, *Front. Cell Dev. Biol.*). The slightly higher overall capture likely reflects the dense cell packing in the brain. Besides the different architecture and lower cell density, the high density regions in our dataset reached up to ~5,000 UMIs per bin at bin 20 resolution. Of note is that the present spatial transcriptomics data closely match the gene expression profiles of the single-cell reference used for deconvolution, supporting that downstream analyses are possible (**Fig. S12**). Given that the per-spot capture probes are engineered to be in excess and host-derived and microbiome signals are in most cases occupying distinct areas we don't believe that non-host RNA competition is a major contributor to the UMI recovery, instead as discussed in R1 q3 the assay

is not sequencing saturated and additional sequencing can further benefit the observed features recovery (Fig. S8d).

6. Bias in Microbial Detection Efficiency Across Species

The authors present aggregate improvements in microbial RNA capture, but do not evaluate PAP efficiency across different microbial taxa. Will be authors be able to:

1. Test or discuss whether certain bacterial phyla or cell wall types (e.g., Gram-negative vs Gram-positive) are more amenable to PAP
2. Ideally, provide 16S rRNA profiles from adjacent tissue sections or bulk metagenomics as a baseline for microbial composition.

Author response: To evaluate potential taxonomic biases introduced by *in situ* polyadenylation (PAP), we performed bulk metatranscriptomic profiling from sections adjacent to those used for spatial transcriptomics (Methods lines 440-453). This provides an orthogonal measure of microbial composition independent of *in situ* capture or spatial barcoding. We first assessed the overall distribution of Gram-positive and Gram-negative bacteria across platforms (Gram annotations based on the BacDive database, Methods). As shown in Fig. S5b and below, the relative proportions of Gram-positive, Gram-negative, unclassified, and unknown taxa are similar between spatial transcriptomics and bulk metatranscriptomics across all intestinal regions. This suggests that the *in situ* polyadenylation step does not preferentially enrich one Gram class over another.

To evaluate detection at finer taxonomic resolution, we compared genus-level relative abundances across the two modalities (Fig. S5a). In the large intestine, where microbial abundance and diversity are highest, we observe very strong agreement between the spatial and metatranscriptomic datasets. In the small intestine, where microbial biomass is lower and composition more variable, we observe modest divergence between paired datasets, likely reflecting spatial heterogeneity, subtle shifts between sister sections, and contamination due to environmental contamination in the bulk sequencing assays. Despite this, the overall representation of major genera and their Gram-type distributions remain comparable, suggesting that PAP does not introduce systematic taxonomic bias even in more challenging low-biomass environments. Together, these comparisons support the conclusion that *in situ* polyadenylation enables reliable and broadly representative microbial RNA capture.

Minor Comments

1. Clarify Background Signal in Control (Fig. 2H)

The detection of microbial signal in control (non-PAP) samples suggests some level of background or non-specific capture. Can the authors discuss the likely sources of this signal (e.g., A-rich bacterial transcripts, mispriming), and estimate the percentage of background signal attributable to such artifacts.

Author response: Internal priming of A-rich regions is pervasive in spatial sequencing technologies. As few as three consecutive A's are sufficient to prime reverse transcription with poly(dT) primers³. Further, because the resulting cDNA products are shorter than full-length molecules primed at their 3' end, length bias in PCR can cause internal priming artifacts to become enriched in the final libraries.

2. Species Near Host-Microbiome Boundary

Given the spatial resolution and descriptive nature of genus-level enrichment at the host boundary, it would strengthen the interpretation to:

1. Validate taxa localization using FISH or adjacent mucus staining (e.g., Alcian Blue/PAS).
2. Highlight taxa known to inhabit the mucus layer (e.g., Akkermansia, Bacteroides fragilis) and compare observed results to expectations.

Author Response: We thank the reviewer for this insightful comment. To highlight taxa that occupy specific niches, we analyzed the spatial distribution of select families as a function of distance from the host boundary in the proximal colon of a C57BL/6 mouse (Fig. S19). Our results show that *Lachnospiraceae* and *Oscillospiraceae* are enriched closer to the tissue, whereas *Lactobacillaceae* and *Bacteroidaceae* are more prominent toward the lumen. These patterns align with established observations of ascending colon versus digesta communities⁴. We note in the revised manuscript (lines 340-342) that orthogonal validation strategies and appropriate experimental controls remain essential for advancing this type of research.

3. While the authors use Moran's I and Ripley's H to analyze spatial clustering, these are first-order analyses. The authors should consider adding higher-order spatial statistics such as KNN-based clustering, bacterial-bacterial co-occurrence networks, or host-microbe proximity scores to showcase the richness of the data obtained here, and demonstrate future directions of advance analysis of host-microbiome data.

Author response: We thank the reviewer for this thoughtful suggestion. We fully agree that our dataset is rich and amenable to a wide array of higher-order spatial analyses that extend beyond the first-order metrics such as Moran's I and Ripley's H, which we employed to quantify spatial clustering and colony

sizes of bacterial genera. Importantly, we have already implemented bacterial co-occurrence analysis in the current study, as shown in Figure 3i and described in the Results and Methods sections (lines 609-614). This analysis revealed strong spatial correlations between specific genera such as *Turicimonas* and *Sutterella*, highlighting the potential for inferring bacterial-bacterial interactions from high-resolution spatial transcriptomics data. Building on this, we are now extending co-occurrence and proximity-based analyses to additional datasets, including newly acquired high-resolution sections from both healthy and tumor-bearing mice. We also acknowledge the importance of KNN-based approaches and host-microbe proximity scoring, and we agree these represent promising directions for future work. As this is a nascent field, we believe that widespread community engagement will be crucial. In line with this, we have already shared our data with four independent research groups who expressed an interest in developing computational tools to explore host-microbiome interactions using spatial transcriptomics, underscoring the utility and relevance of our dataset for advancing computational method development

4. The biological inference from the tumor dataset (e.g., Fig. 4B–D) seems overextended given the apparent $n=1$. Can the authors help clarify the sample size and biological replicates in the figure legends, main text and material and methods. The authors should additionally temper down the language around biological conclusions, given the limited power (both number of animals and spatial levels), and the relatively descriptive nature of the data.

Author response: We have now generated two additional tumor datasets one from the ileum of the same mouse displayed in Fig. 4 and another individual. The new data from these additional experiments support our initial conclusions. We have clarified the sample size in the main text where applicable. Please see response to R2 (q1b).

5. To support potential adoption of the method, it will be of high interest to the readers for a cost estimation. Can the authors perform a tabulation of the reagent and enzyme costs per sample, time and labor requirements for in situ polyadenylation and library preparation steps. Such an estimation will be very helpful for broad adoption of this approach!

Author response: Thanks for the suggestion! We now provide an approximate cost and time estimate for the protocol per sample. This breakdown excludes bulk reagents (e.g., methanol, buffer solutions) and capital equipment costs (e.g., Qubit, Fragment Analyzer, thermocyclers). We focus on reagent and sequencing costs required for in situ polyadenylation, spatial capture, and library preparation. For the Visium-based protocol, the cost per sample is approximately \$2,048 USD, including the slide (\$1,300), yPAP enzyme (\$44), RNase inhibitor (\$0.30), ATP (\$0.45), and sequencing (\$703). For Stereo-seq, the cost per sample is slightly lower at ~\$1,933 USD, with similar enzymatic costs and a per-tile cost of ~\$1,012 and DNBSEQ sequencing estimated at ~\$875/sample. It is worth noting that the additional cost to perform the *in situ* polyadenylation is only ~\$45/sample. A complete breakdown is shown in Supplementary Table 2. Similarly, we include an overview of the labor and process time required for each step in the protocols in Supplementary Table 3.

Reviewer #1 (Remarks on code availability):

I took a look briefly at the github, and commend the authors for making the code publically available. I would suggest a better cleaning up of the github for readability, including a cleaner figure-specific markdown for each figure/ST method type to allow readers to reproduce the data.

Author response: We have made changes to the GitHub repository as requested by the reviewer.

Reviewer #2 (Remarks to the Author):

In this manuscript, Ntekas, Takayasu and colleagues extend a previous published spatial RNA capture method introduced by the De Vlaminck laboratory (Nature Biotechnology 41, 2023) to a sample type that they had not demonstrated previously: the mammalian gastrointestinal tract. In this approach, previously termed spatial total RNA-sequencing (STRS), the authors treat slices of the mouse GI tract from an APC tumor model with poly-A-polymerase. On the host side, this leads to polyadenylation of a variety of non-mRNA species. On the microbiome side, this leads to the polyadenylation of both ribosomal RNAs and mRNAs. Critically, by marking all RNAs with this sequence feature, the authors are then able to use commercially available spatial transcriptomics platforms that leverage polyA capture (Visium and StereoSeq) to spatially barcode these RNAs, such that sequencing can reveal RNA identity and location.

The authors first demonstrate the ability of this method to characterize the abundance and distribution simultaneously with host side RNA expression at multiple locations along the length of the mouse gut using the visium platform. They highlight an interesting set of observations produced by these measurements, including established variations in the abundance and density of the microbiota along the length of the GI tract, local patches of increased microbial richness within the lumen, and differential enrichment of different microbial taxa along the transverse axis.

They then extend their measurements to the stereo-seq platform which, unlike the ~55-micron pitch between capture features, has a ~0.5-micron pitch between 0.2-micron capture spots. This produces a noticeable increase in the resolution with which host and microbial RNAs can be assigned to locations in space. As validation of this method, the authors demonstrate a general agreement between the stereo-seq measurements of host and microbe and matched slices in measured with Visium. They then highlight a variety of interesting observations made on this singular slice. For example, on the microbial side, these differential distributions of different bacteria can now be resolved with respect to host features such as villi, revealing, as expected, differential abundance of different bacteria in proximity to the host. They also reveal an interesting spatial patchiness to the distribution of bacteria, which may represent local and differential outgrowth of specific bacteria. On the host side, they segment their measurements into cells and then transfer labels to these cells based on published scRNA-seq. They generally show cell types and marker genes thereof are generally found where one would expect in the gut.

Finally, in a single slice that contains a tumor (from a different mouse), they show that there are variations in both the organization of the host tissue, which is unsurprising, and in the distribution of specific bacteria with respect to the tumor-lumen boundary.

Overall, the authors have identified a particularly exciting application space, in my opinion, for their STRS method. Especially when coupled with the stereo-seq platform, I anticipate that this method will be of immediate interest not just to researchers that study the gut microbiome but microbial communities, both commensal and environmental, in general.

Author response: Thank you for the appreciation of this work. Your comments and suggestions below are on point and have enabled us to significantly improve our analysis and presentation of the data.

However, despite this enthusiasm, I have several major concerns that lower my enthusiasm for the work in its current form. I provide these below; however, if the authors can address these concerns, I would be supportive of publication of this method.

1) My first major concern revolves around reproducibility, benchmarking, and quantification of key performance metrics. Given that the central deliverable of this paper is a new method, I feel the authors have not provided sufficient description of these key quantities.

a. The comparison of host-side and microbial-side gene expression between Visium and stereo-seq on matching slices (Figure S6) and of the non-tumor and tumor-containing slice (Figure S14) are the only clear measure of technical reproducibility. It is reassuring that the authors see correlation between these different modalities and two slices. However, so much of the manuscript is dedicated to describing elements of the spatial patterns seen with stereo-seq, particularly in microbial distribution. Yet there is no cross validation of the spatial patterns between these Visium and stereo-seq. Could the authors demonstrate that they see similar patterns (at least coarsely along the transverse axis) between the two measurement types?

Author response: To address this point, we performed additional Stereo-seq experiments with in situ polyadenylation spanning five gut regions: proximal small intestine, ileum, cecum, and proximal and distal colon. Four regions (proximal small intestine, ileum, cecum, distal colon) were profiled in one C57BL/6 mouse, and the proximal colon was profiled in a second C57BL/6 mouse. Spatial spots were divided into five bins by distance from the tissue boundary, enabling direct comparison with the Visium data (**Fig. 2g, Fig. S18**). Microbial distributions along the transverse axis were highly concordant across platforms: in the small intestine, Pseudomonadota were enriched near tissue, Bacteroidota, the dominant phylum in the ileum, were lumen-biased, and Bacillota declined from proximal SI to ileum. In the large intestine, Bacillota remained the most abundant phyla with greater representation in the lumen, while Bacteroidota showed tissue enrichment in the cecum and distal colon. These consistent patterns confirm that the spatial microbial organization we report is robust and platform-independent.

a

b. Similarly, there are only two stereo-seq slice measurements presented in the paper (that of Figures 3 and 4) and they represent a non-tumor and tumor environment. Thus, it is not at all clear if the patterns that are discussed in these figures are reproducible. Could the authors repeat these measurements in a second mouse and show that they see the same distribution of bacteria and the same statistical patterns in bacterial distributions that they highlight in these slices? If the biological variability is such that these patterns do not reproduce, then it would be good to see that the patterns observed in adjacent slices are reproduced.

Author response: To address this concern, we performed additional experiments on an ileal cross-section from the tumor-bearing mouse shown in Figure 4 (Fig. S25), as well as on an additional tumor-bearing section from an independent mouse. These analyses confirm that the presence of a large tumor

mass consistently alters the cell-type composition at the host–microbiome interface and is associated with increased proximity of microbial signal to host tissue. We further demonstrate that the observed colony formation patterns are reproducible across these sections, supporting the robustness of the reported findings (Fig. S27). Additionally, we have now included additional cluster formation analysis across the GI of a second mouse showing preferential colony formation of taxa in defined GI niches. These results are now summarized at lines 238-240 and 295-297.

c. In parallel, while it is great to see reproducibility between stereo-seq and visium. Both methods use the same protocols to prepare samples and generate polyA tails. Thus, it is not clear if there are artifacts shared by both methods. For example, given the diverse cell wall architecture of the microbiota, one might imagine that there is a differential ability of polyA polymerase to tag RNAs from different bacteria or for these tagged RNAs to be released from the sample such that they are equally efficiently captured on the stereo-seq or Visium surface. For these reasons, I feel that it is important that the authors benchmark their measurements against independent techniques. For example, two seemingly straightforward approaches would be 1) to compare host-side RNA expression in stereo-seq to bulk RNA sequencing of the colon and 2) to compare bacterial abundance in the colon to metagenomic or 16S bulk sequencing of either fecal pellets or colonic contents. Without such comparisons, it is not clear, particularly on the microbial side, if the patterns captured accurately reflect what is actually present in these samples.

Author response: We agree with the reviewer on the importance of orthogonal validation. To address this, we performed bulk metatranscriptomic sequencing from intestinal sections adjacent to those profiled with the Visium + PAP protocol. This provides an independent measure of both microbial and host RNA abundance without in situ manipulation or surface capture. As shown in Supplementary Figure 5c, we observed strong concordance between spatial and bulk RNA measurements of host gene expression (Pearson $r = 0.62 - 0.73$). This supports the robustness of host transcript capture by in situ polyadenylation, even when comparing across different sample preparation pipelines. On the microbial side, we compared spatially observed microbial profiles with those obtained from bulk metatranscriptomic sequencing and found excellent agreement (see also response to R1, q6). Together, these comparisons indicate that our method does not introduce major taxonomic or transcriptional biases.

d. The authors provide no discussion of false-positive or false counts in their measurements. Yet, there may be some evidence that these exist. For example, Fig. 3g appears to show *Clostridium* in the lumen as expected but also, surprisingly, inside the host mucosa, submucosa, and muscle layers and lining the outside of the muscle, likely in the serosa. Fig. 3i shows similar features for *Lactobacillus*. Do the authors feel that these intra-host bacteria are real (which would be a provocative conclusion) or do they represent false positives, or perhaps contamination during processing? Are these distributions reproducible between animals?

Author response: We thank the reviewer for raising this important point. Given the high microbial density in the lumen and the 10 μm section thickness, it is possible that microbial RNA originating from adjacent luminal regions is detected deeper within the tissue due to projection artifacts during RNA capture. Other false-positive possibilities include spatial diffusion during enzymatic reactions, small shifts during cryosectioning or mounting, and misclassification of host or library-prep-associated sequences as microbial due to limitations of current databases and classification algorithms. To mitigate the latter, we filtered taxa abundant in non-intestinal tissue (e.g., heart, Fig. S1), as these likely represent misclassified

sequences. We now emphasize the need for orthogonal validation approaches and discuss potential technical refinements to the protocol in the revised Discussion (lines 340-346).

e. The authors make a provocative claim that they can also profile bacterial mRNAs, yet they provide little evidence supporting this claim. First, the methods are not sufficiently detailed to understand how they have handled the ~50% of non-ribosomal RNA reads from microbes using metagenomic sequencing on sister slices. For example, how was the metagenomic sequencing done? These methods should be clarified. Second, of the non-ribosomal reads, they only discuss a single example, EF-Tu, which represents 4% of these reads. What happened to the other 96%? Finally, they provide no evidence that these reads were properly mapped to EF-Tu. Could the authors provide additional evidence that they have properly measured bacterial mRNAs?

Author response: We thank the reviewer for this important remark. Motivated by this comment, we revisited our mRNA alignment pipeline and aligned the unmapped reads to the reference genomes of the six most abundant species. This analysis revealed that the majority of the unmapped signal corresponds to ribosomal regions, as expected, with the remaining signal mapping to other biotypes such as tRNAs and CDS. Importantly, this approach identified multiple bacterial genes beyond EF-Tu, including *atpA*, *gap*, *groL*, *pckA*, and *ppdk*, with spatial distributions shown in Fig. S21. These results support our claim that bacterial mRNAs can indeed be profiled in situ.

We view this as an exciting direction for future work. While this initial analysis represents only a first step as we standardize our computational workflow, it already demonstrates the potential for in situ bacterial mRNA profiling. In follow-up studies, we aim to expand this effort by incorporating dedicated enrichment strategies (e.g., ribodepletion) and improved computational approaches to more comprehensively capture and interpret bacterial mRNA signals.

f. A critical performance metric is the fraction of mRNA that is captured. There are multiple lines of evidence in the paper that suggest that it might be low for both host and microbe. For example, the only two host genes that are highlighted (*Igha* and *Lyz1*) are known to be expressed at massive levels in the cells that make them, plasma cells and Paneth cells. Could the authors provide some estimate of the efficiency of RNA capture? I suspect it is quite modest, which is not necessarily problematic, but potential users need to have a realistic sense of this value so that they can reasonably anticipate what questions this method could be used to answer. In lieu of quantitative estimates, could the authors at least provide examples of more host genes? Examples of the measured expression of more modestly expressed but critical marker genes such as *Lgr5* (stem cells), *Pdgfra* (telocytes), *Pecam* (endothelial cells), *Cd3/Cd4/Cd8* (T cells) would be exceedingly helpful guides for readers as to what they might expect from their own measurements with this technique.

Author response: We thank the reviewer for this thoughtful comment. To provide readers with a realistic sense of detection across expression levels, we have added spatial expression examples of host marker genes, including those suggested by the reviewer (**Fig. S15**). We recognize that capture efficiency of host and microbial transcripts is modest, as expected for a total RNA library; for host alone, detection is spread across ~30,000 unique genes, and the libraries are not yet saturated. Deeper sequencing would likely enhance recovery. This tradeoff reflects the design goal of capturing both host and microbial RNAs simultaneously in an untargeted manner, and is common to sequencing based spatial transcriptomics assays, not only the approach introduced in this work. For applications where detection of specific low-abundance host transcripts is critical, readers may consider complementary approaches such as

MERFISH, which has been optimized for gut tissue applications retrieving tissue morphology and rescuing rare cell type gene marker signatures^{5,6}. Alternatively, the addition of targeted transcript enrichment of our Total RNA libraries could be layered onto our library preparation to enhance detection of select transcripts while preserving the advantages of whole-transcriptome spatial profiling.

g. Finally, the authors repeatedly suggest that the spatial separation of capture features of the stereo-seq array is the actual spatial resolution of their measurements. Yet, they have not provided any evidence of the actual accuracy with which the original distribution of RNAs in the sample is reconstructed. In fact, it would seem highly likely that there is at least some degree of diffusion that spreads RNAs out across multiple stereo-seq capture features. Could the authors provide some quantitative evidence to support their assertion that they reconstruct RNA distributions with 1 micron accuracy (as claimed throughout)?

Author response: We thank the reviewer for this important point. We acknowledge that the spacing of Stereo-seq capture features does not directly equate to the effective resolution at which RNA molecules are localized, as both diffusion during tissue permeabilization and the thickness of the tissue section contribute to signal spread across multiple capture features. We have therefore revised the text throughout to avoid implying literal one-micron positional accuracy. Instead, we now describe our measurements as *high resolution* to emphasize the dense sampling capacity of the Stereo-seq array.

2) My second major concern revolves around the accuracy of the cell type identification that has been performed on the host side

a. The authors claim to segment cells using companion images. Yet I can find no description of how this was done in the methods section. The authors should describe clearly how they have defined cell boundaries and parsed stereo-seq features into cells.

Author response: We thank the reviewer for raising this point. The Stereo-seq assay includes fluorescent nuclear staining, and we followed the recommended analysis workflow and guidelines for image registration and cell segmentation. The figure shows the original fluorescence image (left) alongside the corresponding segmentation mask (right) after processing. We have now included a separate Methods section (lines 477-487), clarifying the approach and added this example panel to the supplement (Fig. S11).

b. There are a wide variety of cell types that are missing in the cell populations the authors list. A particularly glaring omission is essentially all cell types of the stroma. The authors appear to find no fibroblasts, despite ample fibroblasts in the lamina propria and sub-mucosa. Similarly, they do not identify the smooth muscle cells found in the lamina propria, in the sub-mucosa, or that of the inner and outer muscle layers. Finally, they do not identify any endothelial cells, despite multiple such cell types in the

sub-mucosa. Outside of the stroma, they also do not see any of the cells that one might expect for the enteric nervous system. Yet, given the size of their slice one would anticipate at least a few patches of the myenteric plexus visible between the inner and outer muscle layers. The authors should address this apparent issue with their ability to define well established and, in some cases, abundant cell populations.

Author response: We thank the reviewer for highlighting the absence of stromal, endothelial, and ENS classes in our annotation. For deconvolution, we used a published *ApcMin/+* single-cell RNA-seq reference in which dissociation bias enriches epithelium and depletes cells from the lamina propria, submucosa, and muscle layers. In the absence of corresponding cell types, the deconvolution assigns these cells to the transcriptionally most similar reference cluster. In practice, the reference's "Mesothelial cells" label appears in stromal-dense regions. Marker heatmaps comparing the single-cell reference and our Stereo-seq segmented cells (**Fig. S12a**) show robust stromal and endothelial signatures in this cluster, indicating a stromal compartment rather than true mesothelium. We therefore now refer to this category as "Stromal cells," in all relevant analyses and figures. ENS cells may similarly be misclassified as enteroendocrine when explicit ENS classes are absent. Incorporating references with dedicated stromal, endothelial, and ENS populations will further improve spatial cell typing and interpretation.

c. Similarly, some of the cell type labels used are a bit atypical and, at the very least, not clearly defined in the manuscript. For example, what is the meaning of the epithelial population that is not mapped to one of the identified epithelial subtypes. What are glandular epithelial cells? Perhaps the authors could provide a short description of these cell populations in the methods.

Author response: We thank the reviewer for raising this concern. As discussed in Point 2b, we relied on a single-cell reference from a previously published study for the deconvolution analysis⁷. We acknowledge that some of the resulting annotations, like "Epithelial cells" and "Glandular epithelial cells", may appear atypical, particularly in the context of the small intestine. In the original study, the single-cell reference was derived from the entire intestinal epithelium, and these categories were used to describe epithelial populations that expressed general epithelial markers (e.g., *S100a6*) but did not clearly cluster with canonical enterocyte for the "Epithelial cells" or the expected intestinal secretory cell types for the "Glandular epithelial cells". In our dataset, these cells likely represent subsets of the major epithelial lineages with elevated expression of these general epithelial and secretory markers, which is why they were assigned to those categories by the deconvolution algorithm. To avoid confusion and provide a clearer message, we have relabeled these atypical populations as "Other epithelial cells" in all relevant figures and analyses.

d. Along these lines, the authors have a clear problem with their mesothelial population. These cells are known to line the outer muscle layer on the very outside of the gut, yet the cluster identified as mesothelial would appear to fill the entire muscle layer and extend into the lamina propria (e.g. Figure S8). I suspect the authors have misidentified smooth muscle cells and myofibroblasts as mesothelial cells. At the very least they should clarify their interpretation of this distribution.

Author response: We thank the reviewer for raising this concern, please refer to our answer in point 2b.

e. Finally, the authors provide little to no validation that their cells have been segmented properly, that they have been properly identified, and that there is a high degree of certainty in their assignment. Could the authors provide examples of the images they used to drive segmentation of the stereo-seq data along with annotations of the cell types assigned within those images? Similarly, could they provide a UMAP visualization of the expression profiles they see for cells on the host side with the projected

labels colored on this UMAP? This visualization would allow readers to understand the degree to which these cells can be resolved with these measurements. Finally, could the authors provide expression profiles for a variety of canonical markers for the cells that have been assigned to each of their labels? These expression profiles would serve three roles. First, they would allow users to better understand some of the cell type labels. Second, they would provide an independent validation of the quality of the segmentation by revealing if there is any bleed through in expression of canonical markers of one cell type in others. Finally, they would give readers a sense of the actual expression level observed for individual cells for these canonical markers. Again, a sense of RNA capture efficiency and sensitivity of the technique is a critical element for readers to think about how they would use this technique for their own questions.

Author response: We thank the reviewer for this comment. We now include representative segmentation images with cell masks and assigned cell type maps (point 2a; **Fig. S11**). We also provide a UMAP of segmented cells from the ileal cross-section and feature plots for canonical markers (**Fig. S15**). We discuss the consequence of broad feature detection in our assay, which can yield modest per-cell expression for low-abundance markers and therefore reduced separation in UMAP space (point 1f; **Figs. S12b-c**). To validate cell type transcriptional profiles, we added paired heatmaps comparing scaled marker expression between the single-cell reference and our spatially segmented cells, which show strong concordance, and dot plots reporting detection rate and scaled expression for the cell type labels (**Figs. S11–14**).

Finally, I have a series of more minor comments that I think the authors should consider in a potential revision. I provide these below in no specific order.

1) The authors may wish to remove ‘unbiased’ and ‘one micron resolution’ from their abstract. Given that they have not benchmarked their method against any independent technique, I think they have not shown that it is unbiased. Moreover, as I mention above, I do not believe that they have shown that the spatial distributions they reconstruct accurately represent the original spatial distribution in the sample with 1 micron resolution. Similar comments in the abstract should also be removed. The work has the potential to be impressive in its own right without making claims like these that I do not feel are backed up by the experiments presented.

Author response: We agree with the reviewer. No method is completely free of biases, instead we use the term “broad”. The spatial resolution is not only a factor of the physical size of the capture features of the array, but is also limited by diffusion. This is the case for most/all spatial RNA sequencing platforms and needs to be acknowledged. We describe the method as having “high spatial resolution”.

2) The authors cite only their previous image-based microbial mapping method, HiPR-FISH in the introduction and ignore the pioneering work of others. Gary Borisy’s lab is an obvious omission. His CLASI-FISH method was the first to demonstrate the ability for multiplexed mapping of bacteria in microbial communities, and it feels ungenerous to not recognize this work, especially since their method also uses colorimetric barcodes extend via spectral demixing, exactly as CLASI-FISH introduced. This also omits the recent work of Lei Dai. His SEER-FISH method, while not demonstrated in the mammalian gut, clearly provides the ability to map the cellular distribution of microbial communities via imaging.

Author response: We thank the reviewer for pointing out this oversight. We have added citations to CLASI-FISH (Valm et al., *PNAS*, 2011) and SEER-FISH (Dai et al., *Nature communications*, 2023) in the Introduction (lines 28–30 of the revised manuscript).

3) Similarly, the authors criticize the work of other laboratories that have shown that they can use spatial capture platforms, such as Visium, to profile the spatial distribution of bacteria. Specifically, they state that the capture methods used by these approaches will lead to ‘measurement biases and a limited scope of discovery’. Yet, these authors have not shown these limitations with previous methods, nor am I familiar with any published data to support this assertion. Perhaps it would be more accurate (and generous) to the existing methods to highlight what might be benefits of the techniques introduced here.

Author response: We thank the reviewer for this helpful suggestion. In the revised manuscript, we have softened the language to more constructively acknowledge the contributions of existing approaches (lines 33-35). The revised phrasing highlights the value of prior work while motivating the need for a more broadly applicable approach.

4) I feel the authors could do a better job of acknowledging the similarities and differences to their previous STRS method in the introduction. If there are new technical features added to this method to make it compatible with microbes (and not just viruses, as shown before) and with stereo-seq (as opposed to visium, as shown before), it would be very useful for clear statements of what these features are. If not, I don’t think it diminishes the work. However, the average reader would benefit from a clear statement that here they demonstrate that STRS works in the mammalian gut and in microbial communities.

Author response: We appreciate this request, as developing the method for gut microbiome samples required optimization at several key steps. We first tested the previous STRS method and observed that standard methanol fixation led to poor retention of luminal content during sample preparation. Implementing methacarn fixation improved tissue integrity and retention of fecal content within the lumen. We next assessed compatibility of in situ polyadenylation with high-resolution spatial arrays and found that the surface capture probes must end with -VN (data not shown). We also altered the sequencing pipeline to account for unmapped reads. We now clearly acknowledge our previous work (lines 38-39) and summarize key technical differences in lines 66-70 and 75-78.

5) Throughout the manuscript the authors show spatial maps with variable color scales, e.g. Fig 2h, 3e, 3i, 3j, 3k. These color scales are not clearly defined. For these examples, one might guess that these are UMIs for the listed genes/features. However, similar scales are seen when the authors discuss the distribution of cell types, e.g., Fig. S8, in which case it is not at all clear what these color scales mean.

Author response: We have updated the figure legends to clarify the color scales used in each case.

Reviewer #3 (Remarks to the Author):

The manuscript by Ntekas et al. presents the application of their already-published enzymatic in situ polyadenylation approach (STRS; McKellar, D.W., Mantri, M., Hinchman, M.M. et al. Spatial mapping of the total transcriptome by in situ polyadenylation. *Nat Biotechnol* 41, 513–520 (2023). <https://doi.org/10.1038/s41587-022-01517-6>) to the capture of both host and microbial transcripts to perform spatial analysis of the different RNA molecules (host, microbial, ncRNA, lncRNA, microRNA, etc.) in mouse gut. The authors apply their protocol to both the Visium 55-um platform and the Stereo-seq one. By having both host and microbial spatial information, the authors show the distribution of the host spatial gene expression information and of the microbial information in four different portions of the

gut as well as in the lumen. By detecting both host and microbial information, the authors investigate what type of microbes are more present closer to the host tissue and vice versa as well as what microbes are more recurrent. Finally, the authors analyze a portion of ileum tissue with tumors.

The manuscript is well written and presents some potential interesting results related to the spatial architecture and interaction of host and microbial information in the mouse gut.

We thank the reviewer for the supportive comments and appreciation of our work.

Major concerns:

1. The authors state that they analyzed multiple mice (line 327). How many? Also, the authors say that they used both male and female mice (line 333) but their numbers are not stated. Please report this information since information on biological replicates and gender effect on the presented biological findings is crucial.

Author response: We assayed intestinal cross-sections from five mice (2 male, 3 female), all approximately 13 weeks old at the time of collection. One APC female was profiled on both Visium and Stereo-seq, with paired Stereo-seq libraries prepared with and without in situ polyadenylation. All five mice were profiled with Stereo-seq (2 male, 3 female). The neoplasia cohort comprised one APC female and one APC male, each analyzed by Stereo-seq; the female contributed two tumor-bearing sections. The C57BL/6 Gastrointestinal tract profiling dataset comprised one WT male profiled across four gut regions and one WT female profiled at proximal colon only. We have added these numbers to the manuscript at lines 371-373, 396, and 430-432.

2. Fig. 2c-d: what are the correlation values? It looks like Visium can capture a higher number of host RNA molecules than Visium+PAP as shown in Fig. 2d and also Supp. Fig. 2a. Can the authors explain why?

Author response: We now indicate the Pearson correlation values on the correlation plots. The correlation for host mRNAs between the two experiments is very high ($r = 0.96$), as seen below. You are correct: given that polyadenylation broadens the scope of RNA types that can be analyzed, it is expected that the number of host RNAs that are analyzed at the same sequencing depth will be lower as the sequencing budget is distributed across a larger number of RNA types. However, the strong correlation in relative abundance of host mRNAs for Visium and Visium +PAP indicates that the information from the host transcriptome is preserved. This is also corroborated by the excellent agreement in host cell types identified by deconvolution (see response to R1, q4). We find this trade-off of slightly reduced host transcriptome information for broadly increased sensitivity toward microbial RNAs and noncoding RNAs to be highly favorable for studying host-microbiome interactions. If capture of lowly abundant transcripts is important for research questions, we have found that our sequencing libraries are not saturated at standard manufacturer-recommended sequencing depths, and that deeper sequencing increases host UMIs. In summary, polyadenylation does not affect the host transcriptome measurement, and the slight reduction in recovery of host mRNAs (~5-10%) can be compensated by additional sequencing, without increasing the overall cost of the assays by much (sequencing is a small part of the cost of the assay, see response to R1, minor 5).

3. In Supp. Fig. 2a, it almost seems like the microbial reads are competing with host RNA on the surface probes. More analyses are needed to address this aspect. The same trend is observed in STOmics in Supp. Fig. 5a (protein coding). Are the authors sure that Visium+PAP and STOmics(stereo-seq)+PAP can capture comparable host information to Visium and STOmics without PAP? Can the authors use synthetic microbial communities of which they know their microbial proportions and spiked-in RNA to calculate the respective capture proportions?

Author response: The spatially barcoded microarrays used in spatial transcriptomics applications, including Visium and StereoSeq, are manufactured to have high concentration of capture probes on the surface of the slide. These oligonucleotides are in excess in the reverse transcription reaction. Therefore, the hybridization and reverse transcription steps do not lead to a decrease in transcript counts for host genes. Instead, as shown by the rarefaction analysis below, we find that the cause of reduced host transcriptome counts in +PAP samples is the increase in library complexity which arises from incorporating non-coding host RNAs (~2-7% of the UMIs) and microbial RNAs (~15-85% of the UMIs) into the measurement. The rarefaction analysis shows that at a similar depth of sequencing there is a trade-off in the biotypes of molecules covered by the PAP and standard assays, but that deeper sequencing can recover similar numbers of host coding transcripts.

As shown above in our response to q2, the host mRNA levels measured by Visium and Visium + PAP are highly correlated, and we find that the cell type deconvolution is also highly similar. In other words, we are confident the assays +PAP yield comparable host information than the assays without PAP. To address the accuracy of the measurement of microbial composition, we have performed metatranscriptomic analysis of sister sections, and we find excellent agreement between the spatial transcriptomic assay and the metagenomic sequencing, indicating that the microbiome component is similarly reliably measured.

4. Related to points 2 and 3, how is the enzymatic polyadenylation of naturally polyadenylated transcripts (i.e. double polyadenylation) affect the spatial their capture on the slides? Can the authors quantify how many transcripts are double polyadenylated and if there is a biased in such a double polyadenylation?

Author response: This is a great question. Both Visium and Stereo-Seq spatial transcriptomics platforms rely on poly(dT)-VN primers for the capture of polyadenylated molecules, where "V" represents A, C, or G, and "N" can be any base. This anchored design ensures that reverse transcription can only initiate at the very end of the poly(A) tail (avoiding internal priming within the poly(A) stretch and promoting efficient capture of full-length transcripts). Naturally polyadenylated RNAs will undergo "double polyadenylation",

however, due to the nature of VN-anchored oligo(dT) priming, reverse transcription still initiates at the 5' end of the poly(A) tail. For this reason, our workflow does not allow us to directly quantify the proportion of transcripts that are "double polyadenylated," nor to assess poly(A) tail length as a proxy. However, we did not observe systematic biases in transcript capture suggestive of such an effect, as shown by our comparisons to bulk controls, and analyses of transcript and cell type abundance as measured with and without PAP.

5. The authors used a mouse heart to show lack of contamination in the microbial signal. Lines 75-77: were the mouse heart tissue sections analyzed on the same Visium slide to check for contamination? That would be an adequate experimental design even though using a germ-free mouse colon is a more appropriate control.

Author response: To clarify, the heart tissue sections were processed in our laboratory using the same workflow but not on the same Visium slide. Placing multiple samples on the same slide compartment is technically challenging. The purpose of including these samples was not to serve as controls for cross-contamination during slide processing, but rather to identify and filter out sequences that are misclassified as microbial. Reference databases often contain contaminant microbial sequences—human or mouse sequences that have been incorrectly incorporated into microbial genomes during reference construction. These database-level misannotations can lead to false assignments of reads to microbial taxa. In addition, contamination of sequencing libraries with environmental DNA is a frequent issue, specifically for samples with low biomass. By analyzing heart tissue, which lacks genuine microbial signal, we could identify background "microbial" taxa that are artifacts of such misclassification and ensure they were excluded from downstream analyses in our gut datasets. We note that such sequences were detected at only very low levels and that removal of these sequences did not alter our observations. The absence of major sources of contamination, and the excellent agreement we have found with bulk metagenomics provide support for the accuracy of the microbial quantification methods used in this work.

6. Can the authors expand on why the enrichment of RNA from viruses and archaea was greatest in the small intestine (Lines 81-82)?

Author response: We thank the reviewer for this comment. As shown in Supplementary Fig. 2b–c, absolute archaeal and viral counts are highest in the cecum and colon. The 10-fold (viral) and 6-fold (archaeal) increases in the proximal small intestine refer to relative enrichment over the Visium-only baseline. In the ileum, that baseline was near the detection limit; therefore, even a modest absolute gain after in situ polyadenylation (PAP) yields a large fold change. In distal regions, the same or larger absolute increase translates into a smaller fold change because the counts in absence of PAP are already high.

7. Fig. 2f: profiles between CE and CO are fairly similar but CE captures so much more rRNA. How come?

Author response: Figure 2f shows the percentage of RNA molecule types detected across different regions of the gastrointestinal tract using Visium with in situ polyadenylation. The figure shows higher levels of rRNA in the cecum (CE) compared to the colon (CO). To further investigate this finding, we performed additional experiments using the high-resolution Stereo-seq (STOmics) platform with in situ polyadenylation on cross-sections of a control C57BL/6 mouse. These experiments confirmed the same trend: rRNA levels were consistently elevated in the cecum relative to multiple colonic regions, supporting that this difference reflects a genuine biological feature.

8. Fig. 3b: non-coding panel has more non-coding RNA molecules on the left side of the tissue section but that pattern is not preserved in the other three panels where the bottom part is more different compared to the rest. How come?

Author response: We thank the reviewer for the observation. In Figure 3b we show spatial maps of host UMI counts, gene richness, unspliced RNA percentage, and non-coding RNA percentage in an ileal cross-section. These metrics capture distinct facets of the transcriptome and cellular state, so uniform spatial patterns are not expected. Although the sectioning plane can shift local cell-type representation and affect visualization of those metrics, the overall patterns are reproduced in an ileal cross-section of a different mouse.

9. Lines 362 and 375: how was the permeabilization conducted?

Author response: We provide additional detail on the permeabilization procedure in the methods section. We implemented pepsin digestion in an acidic buffer at 37 °C, following the protocol first described in Ref⁸. Pepsin treatment partially digests protein barriers within the tissue section, thereby increasing accessibility of RNA molecules and enabling their hybridization on the array surface.

10. Line 72: 156 M reads/samples. How many reads per spot? It's a more meaningful statistic to understand the sensitivity of the approach.

Author response: We quote this number (31,250 reads per spot) in the new version of the manuscript. We have also included Supplementary Table 1 displaying reads and UMIs per spot for the Visium and Visium + PAP libraries.

Minor comments:

- Line 93: missing “0” at “.0683%”
- Fig. 1b: the permeabilization seems done together or right after fixation before histology. In the text is different.

Author response: We thank the reviewer for pointing this out. We have corrected the typo in Line 93. In Figure 1b, the permeabilization we are referring to is the one achieved during fixation, where the cell membranes become permeable, allowing the PAP enzyme and co-factors to penetrate the cells and polyadenylation to take place. A second level of permeabilization is achieved with the acidic pepsin treatment, resulting in RNA capture on the array surface as described in the methods in lines 405-406.

REFERENCES

1. Burnham, P. *et al.* Separating the signal from the noise in metagenomic cell-free DNA sequencing. *Microbiome* **8**, 18 (2020).
2. Mzava, O. *et al.* A metagenomic DNA sequencing assay that is robust against environmental DNA contamination. *Nat. Commun.* **13**, 4197 (2022).
3. Balázs, Z. *et al.* Template-switching artifacts resemble alternative polyadenylation. *BMC Genomics* **20**, 824 (2019).
4. Nava, G. M., Friedrichsen, H. J. & Stappenbeck, T. S. Spatial organization of intestinal microbiota in the mouse ascending colon. *ISME J.* **5**, 627–638 (2011).
5. Cadinu, P. *et al.* Charting the cellular biogeography in colitis reveals fibroblast trajectories and coordinated spatial remodeling. *Cell* **187**, 2010-2028.e30 (2024).
6. Yang, E. *et al.* In situ profiling of plasma cell clonality with image-based single-cell transcriptomics. Preprint at <https://doi.org/10.1101/2025.05.09.653118> (2025).
7. Jones, J., Shi, Q., Nath, R. R. & Brito, I. L. Keystone pathobionts associated with colorectal cancer promote oncogenic reprogramming. *PLOS ONE* **19**, e0297897 (2024).
8. Ståhl, P. L. *et al.* Visualization and analysis of gene expression in tissue sections by spatial transcriptomics. *Science* **353**, 78–82 (2016).

Response to Reviewers' comments

(Author response in blue)

Reviewer #1 (Remarks to the Author):

I commend the authors for substantial work in addressing some of these comments. There are still some outstanding questions below:

1. The authors should include appropriate statistical tests for the responses in Q4 and Q6, the stacked bar plot visualization does not suffice.

Author response: To address this comment, for Q4, we include correlation analyses, confirming that polyadenylation does not distort deconvolution (Pearson's correlation $r=0.97$, $p=2.33 \times 10^{-10}$, Supplementary Figure 13). For the Gram-category comparisons across different gut regions (Q6), cecum and colon showed very strong agreement between bulk meta-transcriptomics and the spatial RNA-sequencing assay with Jensen–Shannon divergence (JSD) of 0.07 (CE) and 0.09 (CO). The small intestinal sites exhibited modest divergence (SI: JSD = 0.19, IL: JSD = 0.24), consistent with the lower microbial biomass in these sites (Supplementary Figure 5b).

2. Can the authors elaborate further on the rationale for fresh frozen + methacarn, which echos R2's question (Q1d) on unexpected bacteria signal in the lumen, likely due to smearing/floating bacteria and fecal contents. Given that the authors previously used methacarn and paraffin embedding versus fresh frozen (PMID: 33268897), this will be important to discuss.

Author response: Our primary motivation was to develop an assay directly compatible with commercial array-based spatial transcriptomics platforms, which overwhelmingly use fresh-frozen tissue. Within this framework, methacarn fixation (instead of the standard methanol fixation) was essential for retaining fecal content after sectioning and staining, while avoiding the extensive crosslinking and RNA fragmentation associated with FFPE. Although OCT embedding and cryo-sectioning may contribute to some degree of background microbial signal, FFPE processing would also introduce non-trivial and largely uncharacterized effects on microbial RNA recovery and on the capture of host non-polyadenylated RNA, and would require separate optimization. This is an attractive direction for future development, but is beyond the scope of the present study. For these reasons, fresh frozen material combined with methacarn fixation provided the most practical and technically compatible workflow for co-capturing host and microbial RNA.

Reviewer #2 (Remarks to the Author):

In this revised manuscript, Ntekas and colleagues have performed a series of new experiments and analyses to address the concerns that I raised in my original review. Overall, I applaud the authors for a thorough response that strengthens their manuscript. In particular, I appreciate the addition of independent cross validations of several of their measurements using metatranscriptomics. Nonetheless, I have a few issues that I feel were not fully addressed in the previous response. Addressing these issues will provide important performance insights that are essential for readers interested in adopting this method.

First, the agreement between their Visium data and metatranscriptomics of the microbiota in Figure S5a is impressive for cecum and colon, and the addition of these measurements strengthens their work. However, to my eye, there is very little agreement between these measurements in the small intestine, in particular the proximal small intestine. The correlation coefficients seem in both small intestine regions to be driven by a handful of genera. In the text the authors describe this as good agreement across all regions, yet I would say that in the small intestine that is true really only for a very modest number of the measured genera. Readers that would aim to use this technology should be fully aware that the agreement in the small intestine is limited to a very small number of the detected genera. Perhaps there is a technical or biological explanation that mitigates this lack of agreement for the vast majority of detected genera (most of which are detected at low levels in both methods). Nonetheless, I feel the authors need to acknowledge this lack of strong agreement and provide some discussion of why this might be the case.

Author response: We thank the reviewer for this observation. We now acknowledge in the manuscript that agreement with metatranscriptomics is strongest in high-biomass regions (cecum and colon), whereas in the small intestine the agreement is more modest. We attribute this to both technical and biological factors. Technically, low-biomass samples are more susceptible to false positives in bulk metatranscriptomics, while spatial RNA-seq is less sensitive to contamination because barcode incorporation occurs *in situ*. In addition, biologically, small intestine communities can be highly variable over sub-millimeter distances, such that adjacent sections can differ in low-abundance genera.

Second, I raised concerns about false positives, as evidenced by bacteria detected in regions one would next expect to find bacteria. The authors provide a nice discussion of this point in the response. Yet, I cannot find this point acknowledged in the text. It is important for readers to understand that the authors see bacterial reads in locations that they would not be expected to be found, that these reads are not uniformly distributed across bacteria (if this point is true), and to have some explanation from the authors as to why this might be the case. Otherwise, the authors would appear to make the implicit assertion that all of these detected bacterial reads were found in the actual location of those bacteria. If that were true, the presence of such a large number of bacteria in what have previously been believed to be largely sterile regions would be a major discovery.

Author response: We agree with the reviewer that it is important to make this point clear. Therefore, we have now added text to the Discussion (lines: 301-305) describing these possible sources of false positives.

Third, I thank the authors for the additional data they provide in response to my comments on reproducibility. I feel these additional measurements are indeed very useful for providing some evidence that their measurements are reproducible. Yet, the authors provide limited additional analysis of these data to support any of the larger biological conclusions/patterns they highlight in their discussion. This point is particularly apparent in their discussion of the biological findings within the single small intestine slice presented in Figure 3. As they would appear to have collected a second section of the ileum (Figure S18), could they repeat these same analyses in this slice and discuss which of these patterns is reproduced in that slice? This point is particularly important in the context of my comment about false positives above, as one of the differences in spatial distribution that they describe (Figure 3g) focuses on *Clostridium*, *Klebsiella* (not shown), and *Eggerthella*, is apparent, in part, because of the broad distribution of *Clostridium*. However, by my eye much of the broad *Clostridium* distribution arises because of measurements both within host tissue and on the outside of the host tissue. Is this same pattern seen in a different slice?

This point also applies to their discussion of colony size in Figure 3. Are these patterns present within this slice as well? Again, reproducibility in these biological features is an essential criterion that readers should have to judge whether they would adopt this technique.

Author response: We thank the reviewer for raising this important point regarding the reproducibility of spatial patterns in the additional ileum section. As suggested, we examined a second ileum section (Figure S18) and found that the broad spatial distribution of *Clostridium* is indeed reproduced. Although *Eggerthella* was detected at relatively low abundance in this section, the detected reads were predominantly located toward the lumen, consistent with the pattern described in Figure 3 (updated Supplementary Figure 20b and c). Colony analysis for the second ileum slice (*ileum_2*) was previously included in Supplementary Figure 20. In this analysis, colony-like aggregates were again observed for *Lactobacillus* and *Clostridium*, providing independent evidence that the spatial colony patterns described in Figure 3 are reproducible across sections; *Turicibacter* also showed colony-like clustering in this sample. We have now updated Supplementary Figure 20b to include the colony size analysis for the *ileum_2* sample, which again shows smaller colonies for *Lactobacillus* and larger colonies for *Clostridium*, consistent with the patterns observed in Figure 3.

Fourth, I appreciate the qualitative demonstration of sensitivity that the authors now provide by showing the distribution of a variety of genes. Indeed, the sensitivity of stereo-seq would appear to be pretty limited, as modestly expressed marker genes such as *Lgr5* or *Cd3* are simply not seen with the frequency one would expect for the cell types they mark. I also appreciate that the authors now provide dot plots (Figure S12b) to provide some support that the cell type labels have been reasonably propagated to their stereo-seq 'cells'. That being said, the overall lack of definition within the UMAP of these stereo-seq features in Figure S12c as well as the broad intermixing of many of these propagated labels would support the assertion that their ability to robustly define fine cell types divisions is questionable. **The authors are faced with a challenge that is common to many of the spatial capture methods that have both modest cellular resolution and capture efficiency, so my point here is not to highlight what I think is a serious issue.** In particular, Figures S12 and S15 will allow the curious reader to better understand the inherent limitations of the sensitivity of stereo-seq and its ability to robustly identify cells. Nonetheless, so much of the paper makes conclusions draw on firm assignment of cell types/states. Perhaps the authors could acknowledge in the discussion that there is room for improvement.

Author response: The reviewer is correct to point out that these challenges are common to current spatial capture methods, and not a property of the chemistry proposed in this work. The sequencing assay has important advantages (versatility, sequence resolution, breadth of detection) but important limitations too (as noted in the discussion section). We have updated the discussion section further to highlight the strengths and weaknesses of both imaging and sequencing based assays. We believe analyses of spatial interactions and structure for the microbiome offer an important novel dimension to microbiome research. It's unlikely a single technology will be optimal for all scenarios, and we believe imaging and sequencing based assays are highly complementary. We acknowledge in the discussion section that there is room for improvement.

Reviewer #3 (Remarks to the Author):

The author has adequately answered all concerns.

Thank you very much.

Reviewer #3 (Remarks on code availability):

N/A